# TimeMixer: Decomposable Multiscale Mixing for Time Series Forecasting

**Shiyu Wang**[1]*, **Haixu Wu**[2]*, **Xiaoming Shi**[1], **Tengge Hu**[2], **Huakun Luo**[2], **Lintao Ma**[1✉],
**James Y. Zhang**[1], **Jun Zhou**[1✉]

[1]Ant Group, Hangzhou, China [2]Tsinghua University, Beijing, China

`{weiming.wsy,lintao.mlt,peter.sxm,james.z,jun.zhoujun}@antgroup.com,`
`{wuhx23,htg21,luohk19}@mails.tsinghua.edu.cn`

## Abstract

Time series forecasting is widely used in extensive applications, such as traffic planning and weather forecasting. However, real-world time series usually present intricate temporal variations, making forecasting extremely challenging. Going beyond the mainstream paradigms of plain decomposition and multiperiodicity analysis, we analyze temporal variations in a novel view of multiscale-mixing, which is based on an intuitive but important observation that time series present distinct patterns in different sampling scales. The microscopic and the macroscopic information are reflected in fine and coarse scales respectively, and thereby complex variations can be inherently disentangled. Based on this observation, we propose *TimeMixer* as a fully MLP-based architecture with *Past-Decomposable-Mixing* (PDM) and *Future-Multipredictor-Mixing* (FMM) blocks to take full advantage of disentangled multiscale series in both past extraction and future prediction phases. Concretely, PDM applies the decomposition to multiscale series and further mixes the decomposed seasonal and trend components in fine-to-coarse and coarse-to-fine directions separately, which successively aggregates the microscopic seasonal and macroscopic trend information. FMM further ensembles multiple predictors to utilize complementary forecasting capabilities in multiscale observations. Consequently, TimeMixer is able to achieve consistent state-of-the-art performances in both long-term and short-term forecasting tasks with favorable run-time efficiency.

## 1 Introduction

Time series forecasting has been studied with immense interest in extensive applications, such as economics (Granger & Newbold, 2014), energy (Martín et al., 2010; Qian et al., 2019), traffic planning (Chen et al., 2001; Yin et al., 2021) and weather prediction (Wu et al., 2023b), which is to predict future temporal variations based on past observations of time series (Wu et al., 2023a). However, due to the complex and non-stationary nature of the real world or systems, the observed series usually present intricate temporal patterns, where the multitudinous variations, such as increasing, decreasing, and fluctuating, are deeply mixed, bringing severe challenges to the forecasting task.

Recently, deep models have achieved promising progress in time series forecasting. The representative models capture temporal variations with well-designed architectures, which span a wide range of foundation backbones, including CNN (Wang et al., 2023; Wu et al., 2023a; Hewage et al., 2020), RNN (Lai et al., 2018; Qin et al., 2017; Salinas et al., 2020), Transformer (Vaswani et al., 2017; Zhou et al., 2021; Wu et al., 2021; Zhou et al., 2022b; Nie et al., 2023) and MLP (Zeng et al., 2023; Zhang et al., 2022; Oreshkin et al., 2019; Challu et al., 2023). In the development of elaborative model architectures, to tackle intricate temporal patterns, some special designs are also involved in these deep models. The widely-acknowledged paradigms primarily include series decomposition and multiperiodicity analysis. As a classical time series analysis technology, decomposition is introduced to deep models as a basic module by (Wu et al., 2021), which decomposes the complex temporal patterns into more predictable components, such as seasonal and trend, and thereby benefiting the forecasting process (Zeng et al., 2023; Zhou et al., 2022b; Wang et al., 2023). Furthermore,

---

*Equal Contribution. Work was done while Haixu Wu, Tengge Hu, Huakun Luo were interns at Ant Group.

multiperiodicity analysis is also involved in time series forecasting (Wu et al., 2023a; Zhou et al., 2022a) to disentangle mixed temporal variations into multiple components with different period lengths. Empowered with these designs, deep models are able to highlight inherent properties of time series from tanglesome variations and further boost the forecasting performance.

Going beyond the above mentioned designs, we further observe that time series present distinct temporal variations in different sampling scales, e.g., the hourly recorded traffic flow presents traffic changes at different times of the day, while for the daily sampled series, these fine-grained variations disappear but fluctuations associated with holidays emerge. On the other hand, the trend of macro-economics dominates the yearly averaged patterns. These observations naturally call for a multiscale analysis paradigm to disentangle complex temporal variations, where fine and coarse scales can reflect the micro- and the macro-scopic information respectively. Especially for the time series forecasting task, it is also notable that the future variation is jointly determined by the variations in multiple scales. Therefore, in this paper, we attempt to design the forecasting model from a novel view of multiscale-mixing, which is able to *take advantage of both disentangled variations and complementary forecasting capabilities from multiscale series simultaneously*.

Technically, we propose *TimeMixer* with a multiscale mixing architecture that is able to extract essential information from past variations by *Past-Decomposable-Mixing* (PDM) blocks and then predicts the future series by the *Future-Multipredictor-Mixing* (FMM) block. Concretely, TimeMixer first generates multiscale observations through average downsampling. Next, PDM adopts a decom-posable design to better cope with distinct properties of seasonal and trend variations, by mixing decomposed multiscale seasonal and trend components in fine-to-coarse and coarse-to-fine directions separately. With our novel design, PDM is able to successfully aggregate the detailed seasonal information starting from the finest series and dive into macroscopic trend components along with the knowledge from coarser scales. In the forecasting phase, FMM ensembles multiple predictors to utilize complementary forecasting capabilities from multiscale observations. With our meticulous architecture, TimeMixer achieves the consistent state-of-the-art performances in both long-term and short-term forecasting tasks with superior efficiency across all of our experiments, covering extensive well-established benchmarks. Our contributions are summarized as follows:

- Going beyond previous methods, we tackle intricate temporal variations in series forecasting from a novel view of multiscale mixing, taking advantage of disentangled variations and complementary forecasting capabilities from multiscale series simultaneously.
- We propose TimeMixer as a simple but effective forecasting model, which enables the combination of the multiscale information in both history extraction and future prediction phases, empowered by our tailored decomposable and multiple-predictor mixing technique.
- TimeMixer achieves consistent state-of-the-art in performances in both long-term and short-term forecasting tasks with superior efficiency on a wide range of benchmarks.

## 2 RELATED WORK

### 2.1 TEMPORAL MODELING IN DEEP TIME SERIES FORECASTING

As the key problem in time series analysis (Wu et al., 2023a), temporal modeling has been widely explored. According to foundation backbones, deep models can be roughly categorized into the following four paradigms: RNN-, CNN-, Transformer- and MLP-based methods. Typically, CNN-based models employ the convolution kernels along the time dimension to capture temporal patterns (Wang et al., 2023; Hewage et al., 2020). And RNN-based methods adopt the recurrent structure to model the temporal state transition (Lai et al., 2018; Zhao et al., 2017). However, both RNN-and CNN-based methods suffer from the limited receptive field, limiting the long-term forecasting capability. Recently, benefiting from the global modeling capacity, Transformer-based models have been widely-acknowledged in long-term series forecasting (Zhou et al., 2021; Wu et al., 2021; Liu et al., 2022b; Kitaev et al., 2020; Nie et al., 2023), which can capture the long-term temporal dependencies adaptively with attention mechanism. Furthermore, multiple layer projection (MLP) is also introduced to time series forecasting (Oreshkin et al., 2019; Challu et al., 2023; Zeng et al., 2023), which achieves favourable performance in both forecasting performance and efficiency.

Additionally, several specific designs are proposed to better capture intricate temporal patterns, including series decomposition and multi-periodicity analysis. Firstly, for the series decomposition,

Autoformer (Wu et al., 2021) presents the series decomposition block based on moving average to decompose complex temporal variations into seasonal and trend components. Afterwards, FEDformer (Zhou et al., 2022b) enhances the series decomposition block with multiple kernels moving average. DLinear (Zeng et al., 2023) utilizes the series decomposition as the pre-processing before linear regression. MICN (Wang et al., 2023) also decomposes input series into seasonal and trend terms, and then integrates the global and local context for forecasting. As for the multi-periodicity analysis, N-BEATS (Oreshkin et al., 2019) fits the time series with multiple trigonometric basis functions. FiLM (Zhou et al., 2022a) projects time series into Legendre Polynomials space, where different basis functions correspond to different period components in the original series. Recently, TimesNet (Wu et al., 2023a) adopts Fourier Transform to map time series into multiple components with different period lengths and presents a modular architecture to process decomposed components.

Unlike the designs mentioned above, this paper explores the multiscale mixing architecture in time series forecasting. Although there exist some models with temporal multiscale designs, such as Pyraformer (Liu et al., 2021) with pyramidal attention and SCINet (Liu et al., 2022a) with a bifurcate downsampling tree, their future predictions do not make use of the information at different scales extracted from the past observations simultaneously. In TimeMixer, we present a new multiscale mixing architecture with Past-Decomposable-Mixing to utilize the disentangled series for multiscale representation learning and Future-Multipredictor-Mixing to ensemble the complementary forecasting skills of multiscale series for better prediction.

## 2.2 Mixing Networks

Mixing is an effective way of information integration and has been applied to computer vision and natural language processing. For instance, MLP-Mixer (Tolstikhin et al., 2021) designs a two-stage mixing structure for image recognition, which mixes the channel information and patch information successively with linear layers. FNet (Lee-Thorp et al., 2022) replaces attention layers in Transformer with simple Fourier Transform, achieving the efficient token mixing of a sentence. In this paper, we further explore the mixing structure in time series forecasting. Unlike previous designs, TimeMixer presents a decomposable multi-scale mixing architecture and distinguishes the mixing methods in both past information extraction and future prediction phases.

## 3 TimeMixer

Given a series $\mathbf{x}$ with one or multiple observed variates, the main objective of time series forecasting is to utilize past observations (length-$P$) to obtain the most probable future prediction (length-$F$). As mentioned above, the key challenge of accurate forecasting is to tackle intricate temporal variations. In this paper, we propose *TimeMixer* of multiscale-mixing, benefiting from disentangled variations and complementary forecasting capabilities from multiscale series. Technically, TimeMixer consists of a *multiscale mixing architecture* with *Past-Decomposable-Mixing* and *Future-Multipredictor-Mixing* for past information extraction and future prediction respectively.

### 3.1 Multiscale Mixing Architecture

Time series of different scales naturally exhibit distinct properties, where fine scales mainly depict detailed patterns and coarse scales highlight macroscopic variations (Mozer, 1991). This multiscale view can inherently disentangle intricate variations in multiple components, thereby benefiting temporal variation modeling. It is also notable that, especially for the forecasting task, multiscale time series present different forecasting capabilities, due to their distinct dominating temporal patterns (Ferreira et al., 2006). Therefore, we present TimeMixer in a *multiscale mixing architecture* to utilize multiscale series with distinguishing designs for past extraction and future prediction phases.

As shown in Figure 1, to disentangle complex variations, we first downsample the past observations $\mathbf{x} \in \mathbb{R}^{P \times C}$ into $M$ scales by average pooling and finally obtain a set of multiscale time series $\mathcal{X} = \{\mathbf{x}_0, \cdots, \mathbf{x}_M\}$, where $\mathbf{x}_m \in \mathbb{R}^{\lfloor \frac{P}{2^m} \rfloor \times C}, m \in \{0, \cdots, M\}$, $C$ denotes the variate number. The lowest level series $\mathbf{x}_0 = \mathbf{x}$ is the input series, which contains the finest temporal variations, while the highest-level series $\mathbf{x}_M$ is for the macroscopic variations. Then we project these multiscale series into deep features $\mathcal{X}^0$ by the embedding layer, which can be formalized as $\mathcal{X}^0 = \mathrm{Embed}(\mathcal{X})$. With the above designs, we obtain the multiscale representations of input series.

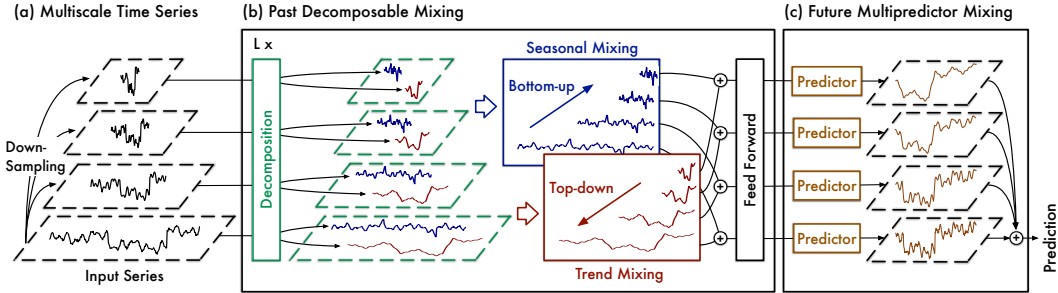

Figure 1: Overall architecture of TimeMixer, which consists of Past-Decomposable Mixing and Future-Multipredictor-Mixing for past observations and future predictions respectively.

Next, we utilize stacked Past-Decomposable-Mixing (PDM) blocks to mix past information across different scales. For the $l$-th layer, the input is $\mathcal{X}^{l-1}$ and the process of PDM can be formalized as:

$$\mathcal{X}^l = \text{PDM}(\mathcal{X}^{l-1}), \;\; l \in \{0, \cdots, L\}, \tag{1}$$

where $L$ is the total layer and $\mathcal{X}^l = \{\mathbf{x}_0^l, \cdots, \mathbf{x}_M^l\}, \mathbf{x}_m^l \in \mathbb{R}^{\lfloor \frac{P}{2^m} \rfloor \times d_{\text{model}}}$ denotes the mixed past representations with $d_{\text{model}}$ channels. More details of PDM are described in the next section.

As for the future prediction phase, we adopt the Future-Multipredictor-Mixing (FMM) block to ensemble extracted multiscale past information $\mathcal{X}^L$ and generate future predictions, which is:

$$\widehat{\mathbf{x}} = \text{FMM}(\mathcal{X}^L), \tag{2}$$

where $\widehat{\mathbf{x}} \in \mathbb{R}^{F \times C}$ represents the final prediction. With the above designs, TimeMixer can successfully capture essential past information from disentangled multiscale observations and predict the future with benefits from multiscale past information.

## 3.2 PAST DECOMPOSABLE MIXING

We observe that for past observations, due to the complex nature of real-world series, even the coarsest scale series present mixed variations. As shown in Figure 1, the series in the top layer still present clear seasonality and trend simultaneously. It is notable that the seasonal and trend components hold distinct properties in time series analysis (Cleveland et al., 1990), which corresponds to short-term and long-term variations or stationary and non-stationary dynamics respectively. Therefore, instead of directly mixing multiscale series as a whole, we propose the Past-Decomposable-Mixing (PDM) block to mix the decomposed seasonal and trend components in multiple scales separately.

Concretely, for the $l$-th PDM block, we first decompose the multiscale time series $\mathcal{X}_l$ into seasonal parts $\mathcal{S}^l = \{\mathbf{s}_0^l, \cdots, \mathbf{s}_M^l\}$ and trend parts $\mathcal{T}^l = \{\mathbf{t}_0^l, \cdots, \mathbf{t}_M^l\}$ by series decomposition block from Autoformer (Wu et al., 2021). As the above analyzed, taking the distinct properties of seasonal-trend parts into account, we apply the mixing operation to seasonal and trend terms separately to interact information from multiple scales. Overall, the $l$-th PDM block can be formalized as:

$$\mathbf{s}_m^l, \mathbf{t}_m^l = \text{SeriesDecomp}(\mathbf{x}_m^l), m \in \{0, \cdots, M\},$$
$$\mathcal{X}^l = \mathcal{X}^{l-1} + \text{FeedForward}\left(\text{S-Mix}\left(\{\mathbf{s}_m^l\}_{m=0}^M\right) + \text{T-Mix}\left(\{\mathbf{t}_m^l\}_{m=0}^M\right)\right), \tag{3}$$

where $\text{FeedForward}(\cdot)$ contains two linear layers with intermediate GELU activation function for information interaction among channels, $\text{S-Mix}(\cdot), \text{T-Mix}(\cdot)$ denote seasonal and trend mixing.

**Seasonal Mixing** In seasonality analysis (Box & Jenkins, 1970), larger periods can be seen as the aggregation of smaller periods, such as the weekly period of traffic flow formed by seven daily changes, addressing the importance of detailed information in predicting future seasonal variations.

Therefore, in seasonal mixing, we adopt the bottom-up approach to incorporate information from the lower-level fine-scale time series upwards, which can supplement detailed information to the seasonality modeling of coarser scales. Technically, for the set of multiscale seasonal parts $\mathcal{S}^l =$

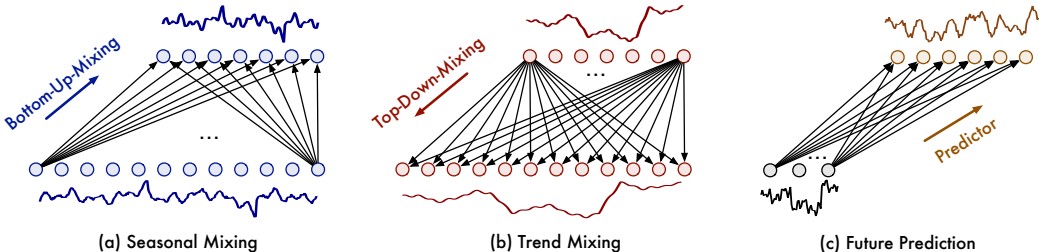

Figure 2: The temporal linear layer in seasonal mixing (a), trend mixing (b) and future prediction (c).

$\{\mathbf{s}_0^l, \cdots, \mathbf{s}_M^l\}$, we use the Bottom-Up-Mixing layer for the $m$-th scale in a residual way to achieve bottom-up seasonal information interaction, which can be formalized as:

$$\text{for } m\colon 1 \to M \text{ do:} \quad \mathbf{s}_m^l = \mathbf{s}_m^l + \text{Bottom-Up-Mixing}(\mathbf{s}_{m-1}^l). \tag{4}$$

where Bottom-Up-Mixing$(\cdot)$ is instantiated as two linear layers with an intermediate GELU activation function along the temporal dimension, whose input dimension is $\lfloor \frac{P}{2^{m-1}} \rfloor$ and output dimension is $\lfloor \frac{P}{2^m} \rfloor$. See Figure 2 for an intuitive understanding.

**Trend Mixing**  Contrary to seasonal parts, for trend items, the detailed variations can introduce noise in capturing macroscopic trend. Note that the upper coarse scale time series can easily provide clear macro information than the lower level. Therefore, we adopt a top-down mixing method to utilize the macro knowledge from coarser scales to guide the trend modeling of finer scales.

Technically, for multiscale trend components $\mathcal{T}^l = \{\mathbf{t}_0^l, \cdots, \mathbf{t}_M^l\}$, we adopt the Top-Down-Mixing layer for the $m$-th scale in a residual way to achieve top-down trend information interaction:

$$\text{for } m\colon (M-1) \to 0 \text{ do:} \quad \mathbf{t}_m^l = \mathbf{t}_m^l + \text{Top-Down-Mixing}(\mathbf{t}_{m+1}^l), \tag{5}$$

where Top-Down-Mixing$(\cdot)$ is two linear layers with an intermediate GELU activation function, whose input dimension is $\lfloor \frac{P}{2^{m+1}} \rfloor$ and output dimension is $\lfloor \frac{P}{2^m} \rfloor$ as shown in Figure 2.

Empowered by seasonal and trend mixing, PDM progressively aggregates the detailed seasonal information from fine to coarse and dive into the macroscopic trend information with prior knowledge from coarser scales, eventually achieving the multiscale mixing in past information extraction.

### 3.3  FUTURE MULTIPREDICTOR MIXING

After $L$ PDM blocks, we obtain the multiscale past information as $\mathcal{X}^L = \{\mathbf{x}_0^L, \cdots, \mathbf{x}_M^L\}, \mathbf{x}_m^L \in \mathbb{R}^{\lfloor \frac{P}{2^m} \rfloor \times d_{\text{model}}}$. Since the series in different scales presents different dominating variations, their predictions also present different capabilities. To fully utilize the multiscale information, we propose to aggregate predictions from multiscale series and present Future-Multipredictor-Mixing block as:

$$\widehat{\mathbf{x}}_m = \text{Predictor}_m(\mathbf{x}_m^L), \ m \in \{0, \cdots, M\}, \ \widehat{\mathbf{x}} = \sum_{m=0}^{M} \widehat{\mathbf{x}}_m, \tag{6}$$

where $\widehat{\mathbf{x}}_m \in \mathbb{R}^{F \times C}$ represents the future prediction from the $m$-th scale series and the final output is $\widehat{\mathbf{x}} \in \mathbb{R}^{F \times C}$. $\text{Predictor}_m(\cdot)$ denotes the predictor of the $m$-th scale series, which firstly adopts one single linear layer to directly regress length-$F$ future from length-$\lfloor \frac{P}{2^m} \rfloor$ extracted past information (Figure 2) and then projects deep representations into $C$ variates. Note that FMM is an ensemble of multiple predictors, where different predictors are based on past information from different scales, enabling FMM to integrate complementary forecasting capabilities of mixed multiscale series.

## 4  EXPERIMENTS

We conduct extensive experiments to evaluate the performance and efficiency of TimeMixer, covering long-term and short-term forecasting, including 18 real-world benchmarks and 15 baselines. The detailed model and experiment configurations are summarized in Appendix A.

Table 1: Summary of benchmarks. Forecastability is one minus the entropy of Fourier domain.

| Tasks | Dataset | Variate | Predict Length | Frequency | Forecastability | Information |
|---|---|---|---|---|---|---|
| Long-term forecasting | ETT (4 subsets) | 7 | 96∼720 | 15 mins | 0.46 | Temperature |
| | Weather | 21 | 96∼720 | 10 mins | 0.75 | Weather |
| | Solar-Energy | 137 | 96∼720 | 10min | 0.33 | Electricity |
| | Electricity | 321 | 96∼720 | Hourly | 0.77 | Electricity |
| | Traffic | 862 | 96∼720 | Hourly | 0.68 | Transportation |
| Short-term forecasting | PEMS (4 subsets) | 170∼883 | 12 | 5min | 0.55 | Traffic network |
| | M4 (6 subsets) | 1 | 6∼48 | Hourly∼Yearly | 0.47 | Database |

**Benchmarks** For long-term forecasting, we experiment on 8 well-established benchmarks: ETT datasets (including 4 subsets: ETTh1, ETTh2, ETTm1, ETTm2), Weather, Solar-Energy, Electricity, and Traffic following (Zhou et al., 2021; Wu et al., 2021; Liu et al., 2022a). For short-term forecasting, we adopt the PeMS (Chen et al., 2001) which contains four public traffic network datasets (PEMS03, PEMS04, PEMS07, PEMS08), and M4 dataset which involves 100,000 different time series collected in different frequencies. Furthermore, we measure the forecastability (Goerg, 2013) of all datasets. It is observed that ETT, M4, and Solar-Energy exhibit relatively low forecastability, indicating the challenges in these benchmarks. More information is summarized in Table 1.

**Baselines** We compare TimeMixer with 15 baselines, which comprise the state-of-the-art long-term forecasting model PatchTST (2023) and advanced short-term forecasting models TimesNet (2023a) and SCINet (2022a), as well as other competitive models including Crossformer (2023), MICN (2023), FiLM (2022a), DLinear (2023), LightTS (2022), FEDformer (2022b), Stationary (2022b), Pyraformer (2021), Autoformer (2021), Informer (2021), N-HiTS (2023) and N-BEATS (2019).

**Unified experiment settings** Note that experimental results reported by the above mentioned baselines cannot be compared directly due to different choices of input length and hyper-parameter searching strategy. For fairness, we make a great effort to provide two types of experiments. In the main text, we align the input length of all baselines and report results averaged from three repeats (see Appendix C for error bars). In Appendix, to compare the upper bound of models, we also conduct a comprehensive hyperparameter searching and report the best results in Table 14 of Appendix.

**Implementation details** All the experiments are implemented in PyTorch (Paszke et al., 2019) and conducted on a single NVIDIA A100 80GB GPU. We utilize the L2 loss for model training. The number of scales $M$ is set according to the time series length to trade off performance and efficiency.

## 4.1 MAIN RESULTS

**Long-term forecasting** As shown in Table 2, TimeMixer achieves consistent state-of-the-art performance in all benchmarks, covering a large variety of series with different frequencies, variate numbers and real-world scenarios. Especially, TimeMixer outperforms PatchTST by a considerable margin, with a 9.4% MSE reduction in Weather and a 24.7% MSE reduction in Solar-Energy. It is worth noting that TimeMixer exhibits good performance even for datasets with low forecastability, such as Solar-Energy and ETT, further proving the generality and effectiveness of TimeMixer.

Table 2: Long-term forecasting results. All the results are averaged from 4 different prediction lengths, that is {96, 192, 336, 720}. A lower MSE or MAE indicates a better prediction. We fix the input length as 96 for all experiments. See Table 13 in Appendix for the full results.

| Models | TimeMixer (Ours) | | PatchTST (2023) | | TimesNet (2023a) | | Crossformer (2023) | | MICN (2023) | | FiLM (2022a) | | DLinear (2023) | | FEDformer (2022b) | | Stationary (2022b) | | Autoformer (2021) | | Informer (2021) | |
|---|---|---|---|---|---|---|---|---|---|---|---|---|---|---|---|---|---|---|---|---|---|---|
| Metric | MSE | MAE | MSE | MAE | MSE | MAE | MSE | MAE | MSE | MAE | MSE | MAE | MSE | MAE | MSE | MAE | MSE | MAE | MSE | MAE | MSE | MAE |
| Weather | **0.240** | **0.271** | 0.265 | 0.285 | 0.251 | 0.294 | 0.264 | 0.320 | 0.268 | 0.321 | 0.271 | 0.291 | 0.265 | 0.315 | 0.309 | 0.360 | 0.288 | 0.314 | 0.338 | 0.382 | 0.634 | 0.548 |
| Solar-Energy | **0.216** | **0.280** | 0.287 | 0.333 | 0.403 | 0.374 | 0.406 | 0.442 | 0.283 | 0.358 | 0.380 | 0.371 | 0.330 | 0.401 | 0.328 | 0.383 | 0.350 | 0.390 | 0.586 | 0.557 | 0.331 | 0.381 |
| Electricity | **0.182** | **0.272** | 0.216 | 0.318 | 0.193 | 0.304 | 0.244 | 0.334 | 0.196 | 0.309 | 0.223 | 0.302 | 0.225 | 0.319 | 0.214 | 0.327 | 0.193 | 0.296 | 0.227 | 0.338 | 0.311 | 0.397 |
| Traffic | **0.484** | **0.297** | 0.529 | 0.341 | 0.620 | 0.336 | 0.667 | 0.426 | 0.593 | 0.356 | 0.637 | 0.384 | 0.625 | 0.383 | 0.610 | 0.376 | 0.624 | 0.340 | 0.628 | 0.379 | 0.764 | 0.416 |
| ETTh1 | **0.447** | **0.440** | 0.516 | 0.484 | 0.495 | 0.450 | 0.529 | 0.522 | 0.475 | 0.480 | 0.516 | 0.483 | 0.461 | 0.457 | 0.498 | 0.484 | 0.570 | 0.537 | 0.496 | 0.487 | 1.040 | 0.795 |
| ETTh2 | **0.364** | **0.395** | 0.391 | 0.411 | 0.414 | 0.427 | 0.942 | 0.684 | 0.574 | 0.531 | 0.402 | 0.420 | 0.563 | 0.519 | 0.437 | 0.449 | 0.526 | 0.516 | 0.450 | 0.459 | 4.431 | 1.729 |
| ETTm1 | **0.381** | **0.395** | 0.406 | 0.407 | 0.400 | 0.406 | 0.513 | 0.495 | 0.423 | 0.422 | 0.411 | 0.402 | 0.404 | 0.408 | 0.448 | 0.452 | 0.481 | 0.456 | 0.588 | 0.517 | 0.961 | 0.734 |
| ETTm2 | **0.275** | **0.323** | 0.290 | 0.334 | 0.291 | 0.333 | 0.757 | 0.610 | 0.353 | 0.402 | 0.287 | 0.329 | 0.354 | 0.402 | 0.305 | 0.349 | 0.306 | 0.347 | 0.327 | 0.371 | 1.410 | 0.810 |

Table 3: Short-term forecasting results in the PEMS datasets with multiple variates. All input lengths are 96 and prediction lengths are 12. A lower MAE, MAPE or RMSE indicates a better prediction.

| Models | | TimeMixer (Ours) | SCINet (2022a) | Crossformer (2023) | PatchTST (2023) | TimesNet (2023a) | MICN (2023) | FiLM (2022a) | DLinear (2023) | FEDformer (2022b) | Stationary (2022b) | Autoformer (2021) | Informer (2021) |
|---|---|---|---|---|---|---|---|---|---|---|---|---|---|
| PEMS03 | MAE | **14.63** | 15.97 | 15.64 | 18.95 | 16.41 | 15.71 | 21.36 | 19.70 | 19.00 | 17.64 | 18.08 | 19.19 |
| | MAPE | **14.54** | 15.89 | 15.74 | 17.29 | 15.17 | 15.67 | 18.35 | 18.35 | 18.57 | 17.56 | 18.75 | 19.58 |
| | RMSE | **23.28** | 25.20 | 25.56 | 30.15 | 26.72 | 24.55 | 35.07 | 32.35 | 30.05 | 28.37 | 27.82 | 32.70 |
| PEMS04 | MAE | **19.21** | 20.35 | 20.38 | 24.86 | 21.63 | 21.62 | 26.74 | 24.62 | 26.51 | 22.34 | 25.00 | 22.05 |
| | MAPE | **12.53** | 12.84 | 12.84 | 16.65 | 13.15 | 13.53 | 16.46 | 16.12 | 16.76 | 14.85 | 16.70 | 14.88 |
| | RMSE | **30.92** | 32.31 | 32.41 | 40.46 | 34.90 | 34.39 | 42.86 | 39.51 | 41.81 | 35.47 | 38.02 | 36.20 |
| PEMS07 | MAE | **20.57** | 22.79 | 22.54 | 27.87 | 25.12 | 22.28 | 28.76 | 28.65 | 27.92 | 26.02 | 26.92 | 27.26 |
| | MAPE | **8.62** | 9.41 | 9.38 | 12.69 | 10.60 | 9.57 | 11.21 | 12.15 | 12.29 | 11.75 | 11.83 | 11.63 |
| | RMSE | **33.59** | 35.61 | 35.49 | 42.56 | 40.71 | 35.40 | 45.85 | 45.02 | 42.29 | 42.34 | 40.60 | 45.81 |
| PEMS08 | MAE | **15.22** | 17.38 | 17.56 | 20.35 | 19.01 | 17.76 | 22.11 | 20.26 | 20.56 | 19.29 | 20.47 | 20.96 |
| | MAPE | **9.67** | 10.80 | 10.92 | 13.15 | 11.83 | 10.76 | 12.81 | 12.09 | 12.41 | 12.21 | 12.27 | 13.20 |
| | RMSE | **24.26** | 27.34 | 27.21 | 31.04 | 30.65 | 27.26 | 35.13 | 32.38 | 32.97 | 38.62 | 31.52 | 30.61 |

Table 4: Short-term forecasting results in the M4 dataset with a single variate. All prediction lengths are in [6, 48]. A lower SMAPE, MASE or OWA indicates a better prediction. *. in the Transformers indicates the name of *former. *Stationary* means the Non-stationary Transformer.

| Models | | TimeMixer (Ours) | TimesNet (2023a) | N-HiTS (2023) | N-BEATS* (2019) | SCINet (2022a) | PatchTST (2023) | MICN (2023) | FiLM (2022a) | LightTS (2022) | DLinear (2023) | FED. (2022b) | Stationary (2022b) | Auto. (2021) | Pyra. (2021) | In. (2021) |
|---|---|---|---|---|---|---|---|---|---|---|---|---|---|---|---|---|
| Yearly | SMAPE | **13.206** | 13.387 | 13.418 | 13.436 | 18.605 | 16.463 | 25.022 | 17.431 | 14.247 | 16.965 | 13.728 | 13.717 | 13.974 | 15.530 | 14.727 |
| | MASE | **2.916** | 2.996 | 3.045 | 3.043 | 4.471 | 3.967 | 7.162 | 4.043 | 3.109 | 4.283 | 3.048 | 3.078 | 3.134 | 3.711 | 3.418 |
| | OWA | **0.776** | 0.786 | 0.793 | 0.794 | 1.132 | 1.003 | 1.667 | 1.042 | 0.827 | 1.058 | 0.803 | 0.807 | 0.822 | 0.942 | 0.881 |
| Quarterly | SMAPE | **9.996** | 10.100 | 10.202 | 10.124 | 14.871 | 10.644 | 15.214 | 12.925 | 11.364 | 12.145 | 10.792 | 10.958 | 11.338 | 15.449 | 11.360 |
| | MASE | **1.166** | 1.182 | 1.194 | 1.169 | 2.054 | 1.278 | 1.963 | 1.664 | 1.328 | 1.520 | 1.283 | 1.325 | 1.365 | 2.350 | 1.401 |
| | OWA | **0.825** | 0.890 | 0.899 | 0.886 | 1.424 | 0.949 | 1.407 | 1.193 | 1.000 | 1.106 | 0.958 | 0.981 | 1.012 | 1.558 | 1.027 |
| Monthly | SMAPE | **12.605** | 12.670 | 12.791 | 12.677 | 14.925 | 13.399 | 16.943 | 15.407 | 14.014 | 13.514 | 14.260 | 13.917 | 13.958 | 17.642 | 14.062 |
| | MASE | **0.919** | 0.933 | 0.969 | 0.937 | 1.131 | 1.031 | 1.442 | 1.298 | 1.053 | 1.037 | 1.102 | 1.097 | 1.103 | 1.913 | 1.141 |
| | OWA | **0.869** | 0.878 | 0.899 | 0.880 | 1.027 | 0.949 | 1.265 | 1.144 | 0.981 | 0.956 | 1.012 | 0.998 | 1.002 | 1.511 | 1.024 |
| Others | SMAPE | **4.564** | 4.891 | 5.061 | 4.925 | 16.655 | 6.558 | 41.985 | 7.134 | 15.880 | 6.709 | 4.954 | 6.302 | 5.485 | 24.786 | 24.460 |
| | MASE | **3.115** | 3.302 | 3.216 | 3.391 | 15.034 | 4.511 | 62.734 | 5.09 | 11.434 | 4.953 | 3.264 | 4.064 | 3.865 | 18.581 | 20.960 |
| | OWA | **0.982** | 1.035 | 1.040 | 1.053 | 4.123 | 1.401 | 14.313 | 1.553 | 3.474 | 1.487 | 1.036 | 1.304 | 1.187 | 5.538 | 5.879 |
| Weighted Average | SMAPE | **11.723** | 11.829 | 11.927 | 11.851 | 15.542 | 13.152 | 19.638 | 14.863 | 13.525 | 13.639 | 12.840 | 12.780 | 12.909 | 16.987 | 14.086 |
| | MASE | **1.559** | 1.585 | 1.613 | 1.559 | 2.816 | 1.945 | 5.947 | 2.207 | 2.111 | 2.095 | 1.701 | 1.756 | 1.771 | 3.265 | 2.718 |
| | OWA | **0.840** | 0.851 | 0.861 | 0.855 | 1.309 | 0.998 | 2.279 | 1.125 | 1.051 | 1.051 | 0.918 | 0.930 | 0.939 | 1.480 | 1.230 |

* The original paper of N-BEATS (2019) adopts a special ensemble method to promote the performance. For fair comparisons, we remove the ensemble and only compare the pure forecasting models.

**Short-term forecasting**  TimeMixer also shows great performance in short-term forecasting under both multivariate and univariate settings (Table 3-4). For PeMS benchmarks that record multiple time series of citywide traffic networks, due to the complex spatiotemporal correlations among multiple variates, many advanced models degenerate a lot in this task, such as PatchTST (2023) and DLinear (2023), which adopt the channel independence design. In contrast, TimeMixer still performs favourablely in this challenging problem, verifying its effectiveness in handling complex multivariate time series forecasting. As for the M4 dataset for univariate forecasting, it contains various temporal variations under different sampling frequencies, including hourly, daily, weekly, monthly, quarterly, and yearly, which exhibits low predictability and distinctive characteristics across different frequencies. Remarkably, Timemixer consistently performs best across all frequencies, affirming the multiscale mixing architecture's capacity in modeling complex temporal variations.

## 4.2  MODEL ANALYSIS

**Ablations**  To verify the effectiveness of each component of TimeMixer, we provide detailed ablation study on every possible design in both Past-Decomposable-Mixing and Future-Multipredictor-Mixing blocks on all 18 experiment benchmarks. From Table 5, we have the following observations.

The exclusion of Future-Multipredictor-Mixing in ablation ② results in a significant decrease in the model's forecasting accuracy for both short and long-term predictions. This demonstrates that mixing future predictions from multiscale series can effectively boost the model performance.

For the past mixing, we verify the effectiveness by removing or replacing components gradually. In ablations ③ and ④ that remove seasonal mixing and trend mixing respectively, also cause a decline

Table 5: Ablations on both PDM (*Decompose*, *Season Mixing*, *Trend Mixing*) and FMM blocks in M4, PEMS04 and predict-336 setting of ETTm1. ↗ indicates the bottom-up mixing while ↙ indicates top-down. A check mark ✓ and a wrong mark × indicate with and without certain components respectively. ① is the official design in TimeMixer (See Appendix F for complete ablation results).

| Case | Decompose | Past mixing | | Future mixing | M4 | | | PEMS04 | | | ETTm1 | |
|------|-----------|-------------|-------|---------------|-------|------|------|--------|-------|-------|-------|-------|
| | | Seasonal | Trend | Multipredictor | SMAPE | MASE | OWA | MAE | MAPE | RMSE | MSE | MAE |
| ① | ✓ | ↗ | ↙ | ✓ | **11.723** | **1.559** | **0.840** | **19.21** | **12.53** | **30.92** | **0.390** | **0.404** |
| ② | ✓ | ↗ | ↙ | × | 12.503 | 1.634 | 0.925 | 21.67 | 13.45 | 34.89 | 0.402 | 0.415 |
| ③ | ✓ | × | ↙ | ✓ | 13.051 | 1.676 | 0.962 | 24.49 | 16.28 | 38.79 | 0.411 | 0.427 |
| ④ | ✓ | ↗ | × | ✓ | 12.911 | 1.655 | 0.941 | 22.91 | 15.02 | 37.04 | 0.405 | 0.414 |
| ⑤ | ✓ | ↙ | ↙ | ✓ | 12.008 | 1.628 | 0.871 | 20.78 | 13.02 | 32.47 | 0.392 | 0.413 |
| ⑥ | ✓ | ↗ | ↗ | ✓ | 11.978 | 1.626 | 0.859 | 21.09 | 13.78 | 33.11 | 0.396 | 0.415 |
| ⑦ | ✓ | ↙ | ↗ | ✓ | 13.012 | 1.657 | 0.954 | 22.27 | 15.14 | 34.67 | 0.412 | 0.429 |
| ⑧ | × | ↗ | | ✓ | 11.975 | 1.617 | 0.851 | 21.51 | 13.47 | 34.81 | 0.395 | 0.408 |
| ⑨ | × | ↙ | | ✓ | 11.973 | 1.622 | 0.850 | 21.79 | 14.03 | 35.23 | 0.393 | 0.406 |
| ⑩ | × | × | | ✓ | 12.468 | 1.671 | 0.916 | 24.87 | 16.66 | 39.48 | 0.405 | 0.412 |

of performance. This illustrates that solely relying on seasonal or trend information interaction is insufficient for accurate predictions. Furthermore, in both ablations ⑤ and ⑥, we employed the same mixing approach for both seasons and trends. However, it cannot bring better predictive performance. Similar situation occurs in ⑦ that adopts opposite mixing strategies to our design. These results demonstrate the effectiveness of our design in both bottom-up seasonal mixing and top-down trend mixing. Concurrently, in ablations ⑧ and ⑨, we opted to eliminate the decomposition architecture and mix the multiscale series directly. However, without decomposition, neither bottom-up nor top-down mixing method can achieve a good performance, indicating the necessity of season-trend separate mixing. Furthermore, in ablations ⑩, eliminating the entire Past-Decomposable-Mixing block causes a serious drop in the model's predictive performance. The above findings highlight the substantial influence of an appropriate past mixing method on the final performance of the model. Starting from the insights in time series, TimeMixer presents the best mixing method in past information extraction.

**Seasonal and trend mixing visualization** To provide an intuitive understanding of PDM, we visualize temporal linear weights for seasonal mixing and trend mixing in Figure 3(a)∼(b). We find that the seasonal and trend items present distinct mixing properties, where the seasonal mixing layer presents periodic changes (repeated blue lines in (a)) and the trend mixing layer is dominated by local aggregations (the dominating diagonal yellow line in (b)). This also verifies the necessity of adopting separate mixing techniques for seasonal and trend terms. Furthermore, Figure 3(c) shows the predictions of season and trend terms in fine (scale 0) and coarse (scale 3) scales. We can observe that the seasonal terms of fine-scale and trend parts of coarse-scale are crucial for accurate predictions. This observation provides insights for our design in utilizing bottom-up mixing for seasonal terms and top-down mixing for trend components.

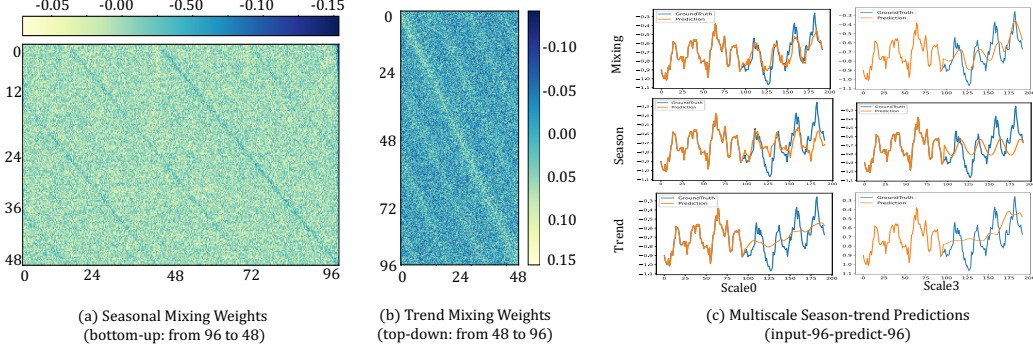

(a) Seasonal Mixing Weights (bottom-up: from 96 to 48)  (b) Trend Mixing Weights (top-down: from 48 to 96)  (c) Multiscale Season-trend Predictions (input-96-predict-96)

Figure 3: Visualization of temporal linear weights in seasonal mixing (Eq. 4), trend mixing (Eq. 5), and predictions from multiscale season-trend items. All the experiments are on the ETTh1 dataset under the input-96-predict-96 setting.

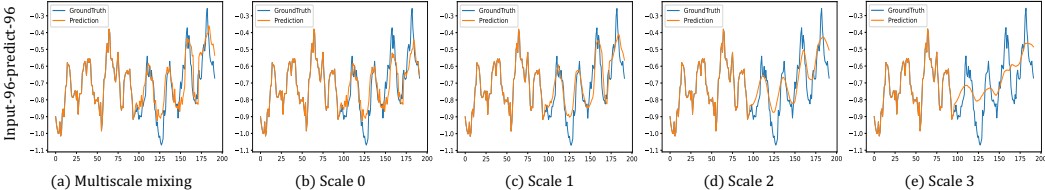

| (a) Multiscale mixing | (b) Scale 0 | (c) Scale 1 | (d) Scale 2 | (e) Scale 3 |

Figure 4: Visualization of predictions from different scales ($\widehat{\mathbf{x}}_m^L$ in Eq. 6) on the input-96-predict-96 settings of the ETTh1 dataset. The implementation details are included in Appendix A.

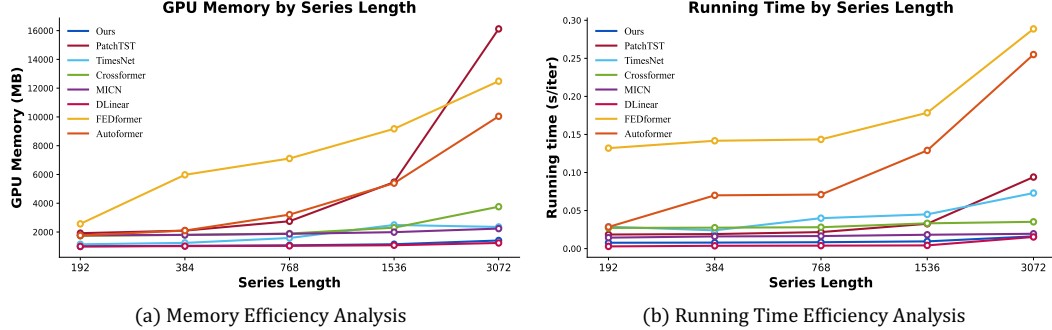

| (a) Memory Efficiency Analysis | (b) Running Time Efficiency Analysis |

Figure 5: Efficiency analysis in both GPU memory and running time. The results are recorded on the ETTh1 dataset with batch size as 16. The running time is averaged from $10^2$ iterations.

**Multipredictor visualization**  To provide an intuitive understanding of the forecasting skills of multiscale series, we plot the forecasting results from different scales for qualitative comparison. Figure 4(a) presents the overall prediction of our model with Future-Multipredictor-Mixing, which indicates accurate prediction according to the future variations using mixed scales. To study the component of each individual scale, we demonstrate the prediction results for each scale in Figure 4(b)~(e). Specifically, prediction results from fine-scale time series concentrate more on the detailed variations of time series and capture seasonal patterns with greater precision. In contrast, as shown in Figure 4(c)~(e), with multiple downsampling, the predictions from coarse-scale series focus more on macro trends. The above results also highlight the benefits of Future-Multipredictor-Mixing in utilizing complementary forecasting skills from multiscale series.

**Efficiency analysis**  We compare the running memory and time against the latest state-of-the-art models in Figure 5 under the training phase, where TimeMixer consistently demonstrates favorable efficiency, in terms of both GPU memory and running time, for various series lengths (ranging from 192 to 3072), in addition to the consistent state-of-the-art performances for both long-term and short-term forecasting tasks.

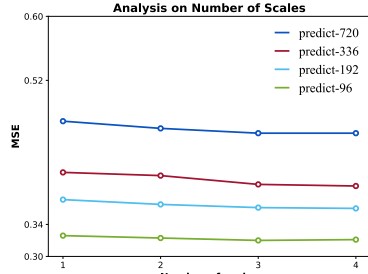

Figure 6: Analysis on number of scales on ETTm1 dataset.

**Analysis on number of scales**  We explore the impact from the number of scales ($M$) in Figure 6 under different series lengths. Specifically, when $M$ increases, the performance gain declines for shorter prediction lengths. In contrast, for longer prediction lengths, the performance improves more as $M$ increases. Therefore, we set $M$ as 3 for long-term forecast and 1 for short-term forecast to trade off performance and efficiency.

## 5 CONCLUSION

We presented TimeMixer with a multiscale mixing architecture to tackle the intricate temporal variations in time series forecasting. Empowered by Past-Decomposable-Mixing and Future-Multipredictor-Mixing blocks, TimeMixer took advantage of both disentangled variations and complementary forecasting capabilities. In all of our experiments, TimeMixer achieved consistent state-of-the-art performances in both long-term and short-term forecasting tasks. Moreover, benefiting from the fully MLP-based architecture, TimeMixer demonstrated favorable run-time efficiency. Detailed visualizations and ablations are included to provide insights for our design.

## 6 ETHICS STATEMENT

Our work only focuses on the scientific problem, so there is no potential ethical risk.

## 7 REPRODUCIBILITY STATEMENT

We involve the implementation details in Appendix A, including dataset descriptions, metric calculation and experiment configuration. The source code is provided in supplementary materials and public in GitHub (https://github.com/kwuking/TimeMixer) for reproducibility.

## ACKNOWLEDGMENTS

This work was supported by Ant Group through CCF-Ant Research Fund.

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

## A  IMPLEMENTATION DETAILS

We summarized details of datasets, evaluation metrics, experiments and visualizations in this section.

**Datasets details**  We evaluate the performance of different models for long-term forecasting on 8 well-established datasets, including Weather, Traffic, Electricity, Solar-Energy, and ETT datasets (ETTh1, ETTh2, ETTm1, ETTm2). Furthermore, we adopt PeMS and M4 datasets for short-term forecasting. We detail the descriptions of the dataset in Table 6.

Table 6: Dataset detailed descriptions. The dataset size is organized in (Train, Validation, Test).

| Tasks | Dataset | Dim | Series Length | Dataset Size | Frequency | Forecastability* | Information |
|---|---|---|---|---|---|---|---|
| Long-term Forecasting | ETTm1 | 7 | {96, 192, 336, 720} | (34465, 11521, 11521) | 15min | 0.46 | Temperature |
| | ETTm2 | 7 | {96, 192, 336, 720} | (34465, 11521, 11521) | 15min | 0.55 | Temperature |
| | ETTh1 | 7 | {96, 192, 336, 720} | (8545, 2881, 2881) | 15 min | 0.38 | Temperature |
| | ETTh2 | 7 | {96, 192, 336, 720} | (8545, 2881, 2881) | 15 min | 0.45 | Temperature |
| | Electricity | 321 | {96, 192, 336, 720} | (18317, 2633, 5261) | Hourly | 0.77 | Electricity |
| | Traffic | 862 | {96, 192, 336, 720} | (12185, 1757, 3509) | Hourly | 0.68 | Transportation |
| | Weather | 21 | {96, 192, 336, 720} | (36792, 5271, 10540) | 10 min | 0.75 | Weather |
| | Solar-Energy | 137 | {96, 192, 336, 720} | (36601, 5161, 10417) | 10min | 0.33 | Electricity |
| Short-term Forecasting | PEMS03 | 358 | 12 | (15701,5216,434) | 5min | 0.65 | Transportation |
| | PEMS04 | 307 | 12 | (10172,3375,281) | 5min | 0.45 | Transportation |
| | PEMS07 | 883 | 12 | (16911,5622,468) | 5min | 0.58 | Transportation |
| | PEMS08 | 170 | 12 | (10690,3548,265) | 5min | 0.52 | Transportation |
| | M4-Yearly | 1 | 6 | (23000, 0, 23000) | Yearly | 0.43 | Demographic |
| | M4-Quarterly | 1 | 8 | (24000, 0, 24000) | Quarterly | 0.47 | Finance |
| | M4-Monthly | 1 | 18 | (48000, 0, 48000) | Monthly | 0.44 | Industry |
| | M4-Weakly | 1 | 13 | (359, 0, 359) | Weakly | 0.43 | Macro |
| | M4-Daily | 1 | 14 | (4227, 0, 4227) | Daily | 0.44 | Micro |
| | M4-Hourly | 1 | 48 | (414, 0, 414) | Hourly | 0.46 | Other |

∗ The forecastability is calculated by one minus the entropy of Fourier decomposition of time series (Goerg, 2013). A larger value indicates better predictability.

**Metric details**  Regarding metrics, we utilize the mean square error (MSE) and mean absolute error (MAE) for long-term forecasting. In the case of short-term forecasting, we follow the metrics of SCINet (Liu et al., 2022a) on the PeMS datasets, including mean absolute error (MAE), mean absolute percentage error (MAPE), root mean squared error (RMSE). As for the M4 datasets, we follow the methodology of N-BEATS (Oreshkin et al., 2019) and implement the symmetric mean absolute percentage error (SMAPE), mean absolute scaled error (MASE), and overall weighted

average (OWA) as metrics. It is worth noting that OWA is a specific metric utilized in the M4 competition. The calculations of these metrics are:

$$\text{RMSE} = \left(\sum_{i=1}^{F}(\mathbf{X}_i - \widehat{\mathbf{X}}_i)^2\right)^{\frac{1}{2}}, \qquad \text{MAE} = \sum_{i=1}^{F}|\mathbf{X}_i - \widehat{\mathbf{X}}_i|,$$

$$\text{SMAPE} = \frac{200}{F}\sum_{i=1}^{F}\frac{|\mathbf{X}_i - \widehat{\mathbf{X}}_i|}{|\mathbf{X}_i| + |\widehat{\mathbf{X}}_i|}, \qquad \text{MAPE} = \frac{100}{F}\sum_{i=1}^{F}\frac{|\mathbf{X}_i - \widehat{\mathbf{X}}_i|}{|\mathbf{X}_i|},$$

$$\text{MASE} = \frac{1}{F}\sum_{i=1}^{F}\frac{|\mathbf{X}_i - \widehat{\mathbf{X}}_i|}{\frac{1}{F-s}\sum_{j=s+1}^{F}|\mathbf{X}_j - \mathbf{X}_{j-s}|}, \qquad \text{OWA} = \frac{1}{2}\left[\frac{\text{SMAPE}}{\text{SMAPE}_{\text{Naïve2}}} + \frac{\text{MASE}}{\text{MASE}_{\text{Naïve2}}}\right],$$

where $s$ is the periodicity of the data. $\mathbf{X}, \widehat{\mathbf{X}} \in \mathbb{R}^{F \times C}$ are the ground truth and prediction results of the future with $F$ time pints and $C$ dimensions. $\mathbf{X}_i$ means the $i$-th future time point.

**Experiment details**   All experiments were run three times, implemented in Pytorch (Paszke et al., 2019), and conducted on a single NVIDIA A100 80GB GPU. We set the initial learning rate as $10^{-2}$ or $10^{-3}$ and used the ADAM optimizer (Kingma & Ba, 2015) with L2 loss for model optimization. And the batch size was set to be 8 between 128. By default, TimeMixer contains 2 Past Decomposable Mixing blocks. We choose the number of scales $M$ according to the length of the time series to achieve a balance between performance and efficiency. To handle longer series in long-term forecasting, we set $M$ to 3. As for short-term forecasting with limited series length, we set $M$ to 1. Detailed model configuration information is presented in Table 7.

**Visualization details**   To verify complementary forecasting capabilities of multiscale series, we fix the PDM and train a new predictor for the feature at each scale with the ground truth future as supervision in Figure 4; Figure 3(c) also utilizes the same operations. Especially for Figure 4, we also provide the visualization of directly plotting the output of each predictor, i.e. $\widehat{\mathbf{x}}_m$, $m \in \{0, \cdots, M\}$ in Eq. 6. Note that in FMM, we adopt the sum ensemble $\widehat{\mathbf{x}} = \sum_{m=0}^{M} \widehat{\mathbf{x}}_m$ as the final output, the scale of each plotted cure is around $\frac{1}{M+1}$ of ground truth, while we can still observe the distinct forecasting capability of series in different scales. For clearness, we also plot the $(M+1) \times \widehat{\mathbf{x}}_m$ in the second row of Figure 7, where the visualizations are similar to Figure 4.

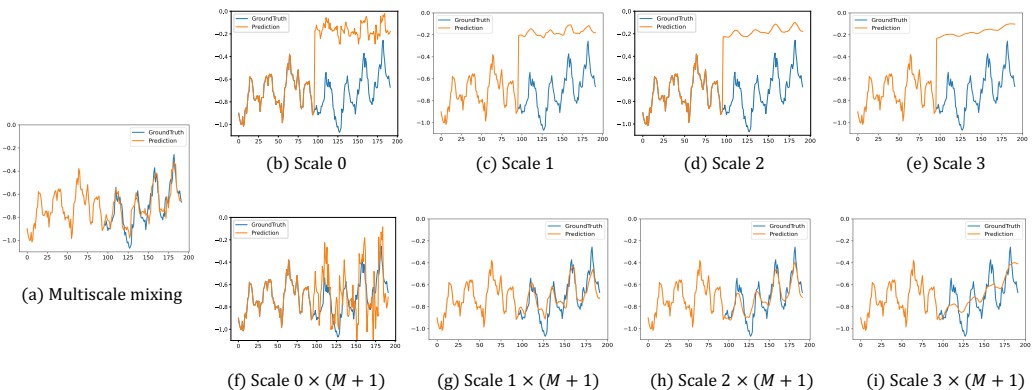

Figure 7: Direct visualization of predictions from different scales ($\widehat{\mathbf{x}}_m^L$ in Eq. 6) on the input-96-predict-96 settings of the ETTh1 dataset. We multiply the $(M+1)$ by the predictions of each scale in the second row.

# B   EFFICIENCY ANALYSIS

In the main text, we have ploted the curve of efficiency in Figure 5. Here we present the quantitive results in Table 8. It should be noted that TimeMixer's outstanding efficiency advantage over Transformer-based models, such as PatchTST, FEDformer, and Autoformer, is attributed to its fully MLP-based network architecture.

Table 7: Experiment configuration of TimeMixer. All the experiments use the ADAM (2015) optimizer with the default hyperparameter configuration for $(\beta_1, \beta_2)$ as (0.9, 0.999).

| Dataset / Configurations | Model Hyper-parameter | | | Training Process | | | |
|---|---|---|---|---|---|---|---|
| | $M$ (Equ. 1) | Layers | $d_{\mathrm{model}}$ | LR* | Loss | Batch Size | Epochs |
| ETTh1 | 3 | 2 | 16 | $10^{-2}$ | MSE | 128 | 10 |
| ETTh2 | 3 | 2 | 16 | $10^{-2}$ | MSE | 128 | 10 |
| ETTm1 | 3 | 2 | 16 | $10^{-2}$ | MSE | 128 | 10 |
| ETTm2 | 3 | 2 | 32 | $10^{-2}$ | MSE | 128 | 10 |
| Weather | 3 | 2 | 16 | $10^{-2}$ | MSE | 128 | 20 |
| Electricity | 3 | 2 | 16 | $10^{-2}$ | MSE | 32 | 20 |
| Solar-Energy | 3 | 2 | 128 | $10^{-2}$ | MSE | 32 | 20 |
| Traffic | 3 | 2 | 32 | $10^{-2}$ | MSE | 8 | 20 |
| PEMS | 1 | 5 | 128 | $10^{-3}$ | MSE | 32 | 10 |
| M4 | 1 | 4 | 32 | $10^{-2}$ | SMAPE | 128 | 50 |

∗ LR means the initial learning rate.

Table 8: The GPU memory (MiB) and speed (running time, s/iter) of each model.

| Series Length | 192 | | 384 | | 768 | | 1536 | | 3072 | |
|---|---|---|---|---|---|---|---|---|---|---|
| Models | Mem | Speed | Mem | Speed | Mem | Speed | Mem | Speed | Mem | Speed |
| TimeMixer (**Ours**) | 1003 | 0.007 | 1043 | 0.007 | 1075 | 0.008 | 1151 | 0.009 | 1411 | 0.016 |
| PatchTST (2023) | 1919 | 0.018 | 2097 | 0.019 | 2749 | 0.021 | 5465 | 0.032 | 16119 | 0.094 |
| TimesNet (2023a) | 1148 | 0.028 | 1245 | 0.024 | 1585 | 0.042 | 2491 | 0.045 | 2353 | 0.073 |
| Crossformer (2023) | 1737 | 0.027 | 1799 | 0.027 | 1895 | 0.028 | 2303 | 0.033 | 3759 | 0.035 |
| MICN (2023) | 1771 | 0.014 | 1801 | 0.016 | 1873 | 0.017 | 1991 | 0.018 | 2239 | 0.020 |
| DLinear (2023) | 1001 | 0.002 | 1021 | 0.003 | 1041 | 0.003 | 1081 | 0.0004 | 1239 | 0.015 |
| FEDFormer (2022b) | 2567 | 0.132 | 5977 | 0.141 | 7111 | 0.143 | 9173 | 0.178 | 12485 | 0.288 |
| Autoformer (2021) | 1761 | 0.028 | 2101 | 0.070 | 3209 | 0.071 | 5395 | 0.129 | 10043 | 0.255 |

## C  ERROR BARS

In this paper, we repeat all the experiments three times. Here we report the standard deviation of our model and the second best model, as well as the statistical significance test in Table 9, 10, 11.

Table 9: Standard deviation and statistical tests for our TimeMixer method and second-best method (PatchTST) on ETT, Weather, Solar-Energy, Electricity and Traffic datasets.

| Model | TimeMixer | | PatchTST (2023) | | Confidence |
|---|---|---|---|---|---|
| Dataset | MSE | MAE | MSE | MAE | Interval |
| Weather | $0.240 \pm 0.010$ | $0.271 \pm 0.009$ | $0.265 \pm 0.012$ | $0.285 \pm 0.011$ | 99% |
| Solar-Energy | $0.216 \pm 0.002$ | $0.280 \pm 0.022$ | $0.287 \pm 0.020$ | $0.333 \pm 0.018$ | 99% |
| Electricity | $0.182 \pm 0.017$ | $0.272 \pm 0.006$ | $0.216 \pm 0.012$ | $0.318 \pm 0.015$ | 99% |
| Traffic | $0.484 \pm 0.015$ | $0.297 \pm 0.013$ | $0.529 \pm 0.008$ | $0.341 \pm 0.002$ | 99% |
| ETTh1 | $0.447 \pm 0.002$ | $0.440 \pm 0.005$ | $0.516 \pm 0.003$ | $0.484 \pm 0.005$ | 99% |
| ETTh2 | $0.364 \pm 0.008$ | $0.395 \pm 0.010$ | $0.391 \pm 0.005$ | $0.411 \pm 0.003$ | 99% |
| ETTm1 | $0.381 \pm 0.003$ | $0.395 \pm 0.006$ | $0.400 \pm 0.002$ | $0.407 \pm 0.005$ | 99% |
| ETTm2 | $0.275 \pm 0.001$ | $0.323 \pm 0.003$ | $0.290 \pm 0.002$ | $0.334 \pm 0.002$ | 99% |

Table 10: Standard deviation and statistical tests for our TimeMixer method and second-best method (SCINet) on PEMS dataset.

| Model | TimeMixer | | | SCINet (2022a) | | | Confidence |
|---|---|---|---|---|---|---|---|
| Dataset | MAE | MAPE | RMSE | MAE | MAPE | RMSE | Interval |
| PEMS03 | $14.63 \pm 0.112$ | $14.54 \pm 0.105$ | $23.28 \pm 0.128$ | $15.97 \pm 0.153$ | $15.89 \pm 0.122$ | $25.20 \pm 0.137$ | 99% |
| PEMS04 | $19.21 \pm 0.217$ | $12.53 \pm 0.154$ | $30.92 \pm 0.143$ | $20.35 \pm 0.201$ | $12.84 \pm 0.213$ | $32.31 \pm 0.178$ | 95% |
| PEMS07 | $20.57 \pm 0.158$ | $8.62 \pm 0.112$ | $33.59 \pm 0.273$ | $22.79 \pm 0.179$ | $9.41 \pm 0.105$ | $35.61 \pm 0.112$ | 99% |
| PEMS08 | $15.22 \pm 0.311$ | $9.67 \pm 0.101$ | $24.26 \pm 0.212$ | $17.38 \pm 0.332$ | $10.80 \pm 0.219$ | $27.34 \pm 0.178$ | 99% |

Table 11: Standard deviation and statistical tests for our TimeMixer method and second-best method (TimesNet) on M4 dataset.

| Model | TimeMixer | | | TimesNet (2023a) | | | Confidence |
|---|---|---|---|---|---|---|---|
| Dataset | SMAPE | MAPE | OWA | SMAPE | MAPE | OWA | Interval |
| Yearly | $13.206 \pm 0.121$ | $2.916 \pm 0.022$ | $0.776 \pm 0.002$ | $13.387 \pm 0.112$ | $2.996 \pm 0.017$ | $0.786 \pm 0.010$ | 95% |
| Quarterly | $9.996 \pm 0.101$ | $1.166 \pm 0.015$ | $0.825 \pm 0.008$ | $10.100 \pm 0.105$ | $1.182 \pm 0.012$ | $0.890 \pm 0.006$ | 95% |
| Monthly | $12.605 \pm 0.115$ | $0.919 \pm 0.011$ | $0.869 \pm 0.003$ | $12.670 \pm 0.106$ | $0.933 \pm 0.008$ | $0.878 \pm 0.001$ | 95% |
| Others | $4.564 \pm 0.114$ | $3.115 \pm 0.027$ | $0.982 \pm 0.011$ | $4.891 \pm 0.120$ | $3.302 \pm 0.023$ | $1.035 \pm 0.017$ | 99% |
| Averaged | $11.723 \pm 0.011$ | $1.559 \pm 0.022$ | $0.840 \pm 0.001$ | $11.829 \pm 0.120$ | $1.585 \pm 0.017$ | $0.851 \pm 0.003$ | 95% |

## D  HYPERPARAMTER SENSITIVITY

In the main text, we have explored the effect of number of scales $M$. Here, we further evaluate the number of layers $L$. As shown in Table 12, we can find that in general, increasing the number of layers ($L$) will bring improvements across different prediction lengths. Therefore, we set to 2 to trade off efficiency and performance.

Table 12: The MSE results of different number of scales ($M$) and layers ($L$) on the ETTm1 dataset.

| Predict Length / Num. of Scales | 96 | 192 | 336 | 720 | Predict Length / Num. of Layers | 96 | 192 | 336 | 720 |
|---|---|---|---|---|---|---|---|---|---|
| 1 | 0.326 | 0.371 | 0.405 | 0.469 | 1 | 0.328 | 0.369 | 0.405 | 0.467 |
| 2 | 0.323 | 0.365 | 0.401 | 0.460 | 2 | 0.320 | 0.361 | 0.390 | 0.454 |
| 3 | 0.320 | 0.361 | 0.390 | 0.454 | 3 | 0.321 | 0.360 | 0.389 | 0.451 |
| 4 | 0.321 | 0.360 | 0.388 | 0.454 | 4 | 0.318 | 0.361 | 0.385 | 0.452 |
| 5 | 0.321 | 0.362 | 0.389 | 0.461 | 5 | 0.322 | 0.359 | 0.390 | 0.457 |

## E  FULL RESULTS

To ensure a fair comparison between models, we conducted experiments using unified parameters and reported results in the main text, including aligning all the input lengths, batch sizes, and training epochs in all experiments. Here, we provide the full results for each forecasting setting in Table 13.

In addition, considering that the reported results in different papers are mostly obtained through hyperparameter search, we provide the experiment results with the full version of the parameter search. We searched for input length among 96, 192, 336, and 512, learning rate from $10^{-5}$ to 0.05, encoder layers from 1 to 5, the $d_{\text{model}}$ from 16 to 512, training epochs from 10 to 100. The results are included in Table 14, which can be used to compare the upper bound of each forecasting model.

We can find that the relative promotion of TimesMixer over PatchTST is smaller under comprehensive hyperparameter search than the unified hyperparameter setting. It is worth noticing that TimeMixer runs much faster than PatchTST according to the efficiency comparison in Table 8. Therefore, considering perfromance, hyperparameter-search cost and efficiency, we believe TimeMixer is a practical model in real-world applications and is valuable to deep time series forecasting community.

Table 13: Unified hyperparameter results for the long-term forecasting task. We compare extensive competitive models under different prediction lengths. *Avg* is averaged from all four prediction lengths, that is 96, 192, 336, 720.

| Models | | TimeMixer (Ours) | | PatchTST 2023 | | TimesNet 2023a | | Crossformer 2023 | | MICN 2023 | | FiLM 2022a | | DLinear 2023 | | FEDformer 2022b | | Stationary 2022b | | Autoformer 2021 | | Informer 2021 | |
|---|---|---|---|---|---|---|---|---|---|---|---|---|---|---|---|---|---|---|---|---|---|---|---|
| Metric | | MSE | MAE | MSE | MAE | MSE | MAE | MSE | MAE | MSE | MAE | MSE | MAE | MSE | MAE | MSE | MAE | MSE | MAE | MSE | MAE | MSE | MAE |
| Weather | 96 | 0.163 | 0.209 | 0.186 | 0.227 | 0.172 | 0.220 | 0.195 | 0.271 | 0.198 | 0.261 | 0.195 | 0.236 | 0.195 | 0.252 | 0.217 | 0.296 | 0.173 | 0.223 | 0.266 | 0.336 | 0.300 | 0.384 |
| | 192 | 0.208 | 0.250 | 0.234 | 0.265 | 0.219 | 0.261 | 0.209 | 0.277 | 0.239 | 0.299 | 0.239 | 0.271 | 0.237 | 0.295 | 0.276 | 0.336 | 0.245 | 0.285 | 0.307 | 0.367 | 0.598 | 0.544 |
| | 336 | 0.251 | 0.287 | 0.284 | 0.301 | 0.246 | 0.337 | 0.273 | 0.332 | 0.285 | 0.336 | 0.289 | 0.306 | 0.282 | 0.331 | 0.339 | 0.380 | 0.321 | 0.338 | 0.359 | 0.395 | 0.578 | 0.523 |
| | 720 | 0.339 | 0.341 | 0.356 | 0.349 | 0.365 | 0.359 | 0.379 | 0.401 | 0.351 | 0.388 | 0.361 | 0.351 | 0.345 | 0.382 | 0.403 | 0.428 | 0.414 | 0.410 | 0.419 | 0.428 | 1.059 | 0.741 |
| | Avg | 0.240 | 0.271 | 0.265 | 0.285 | 0.251 | 0.294 | 0.264 | 0.320 | 0.268 | 0.321 | 0.271 | 0.291 | 0.265 | 0.315 | 0.309 | 0.360 | 0.288 | 0.314 | 0.338 | 0.382 | 0.634 | 0.548 |
| Solar-Energy | 96 | 0.189 | 0.259 | 0.265 | 0.323 | 0.373 | 0.358 | 0.232 | 0.302 | 0.257 | 0.325 | 0.333 | 0.350 | 0.290 | 0.378 | 0.286 | 0.341 | 0.321 | 0.380 | 0.456 | 0.446 | 0.287 | 0.323 |
| | 192 | 0.222 | 0.283 | 0.288 | 0.332 | 0.397 | 0.376 | 0.371 | 0.410 | 0.278 | 0.354 | 0.371 | 0.372 | 0.320 | 0.398 | 0.291 | 0.337 | 0.346 | 0.369 | 0.588 | 0.561 | 0.297 | 0.341 |
| | 336 | 0.231 | 0.292 | 0.301 | 0.339 | 0.420 | 0.380 | 0.495 | 0.515 | 0.298 | 0.375 | 0.408 | 0.385 | 0.353 | 0.415 | 0.354 | 0.416 | 0.357 | 0.387 | 0.595 | 0.588 | 0.367 | 0.429 |
| | 720 | 0.223 | 0.285 | 0.295 | 0.336 | 0.420 | 0.381 | 0.526 | 0.542 | 0.299 | 0.379 | 0.406 | 0.377 | 0.357 | 0.413 | 0.380 | 0.437 | 0.375 | 0.424 | 0.733 | 0.633 | 0.374 | 0.431 |
| | Avg | 0.216 | 0.280 | 0.287 | 0.333 | 0.403 | 0.374 | 0.406 | 0.442 | 0.283 | 0.358 | 0.380 | 0.371 | 0.330 | 0.401 | 0.328 | 0.383 | 0.350 | 0.390 | 0.586 | 0.557 | 0.331 | 0.381 |
| Electricity | 96 | 0.153 | 0.247 | 0.190 | 0.296 | 0.168 | 0.272 | 0.219 | 0.314 | 0.180 | 0.293 | 0.198 | 0.274 | 0.210 | 0.302 | 0.193 | 0.308 | 0.169 | 0.273 | 0.201 | 0.317 | 0.274 | 0.368 |
| | 192 | 0.166 | 0.256 | 0.199 | 0.304 | 0.184 | 0.322 | 0.231 | 0.322 | 0.189 | 0.302 | 0.198 | 0.278 | 0.210 | 0.305 | 0.201 | 0.315 | 0.182 | 0.286 | 0.222 | 0.334 | 0.296 | 0.386 |
| | 336 | 0.185 | 0.277 | 0.217 | 0.319 | 0.198 | 0.300 | 0.246 | 0.337 | 0.198 | 0.312 | 0.217 | 0.300 | 0.223 | 0.319 | 0.214 | 0.329 | 0.200 | 0.304 | 0.231 | 0.443 | 0.300 | 0.394 |
| | 720 | 0.225 | 0.310 | 0.258 | 0.352 | 0.220 | 0.320 | 0.280 | 0.363 | 0.217 | 0.330 | 0.278 | 0.356 | 0.258 | 0.350 | 0.246 | 0.355 | 0.222 | 0.321 | 0.254 | 0.361 | 0.373 | 0.439 |
| | Avg | 0.182 | 0.272 | 0.216 | 0.318 | 0.193 | 0.304 | 0.244 | 0.334 | 0.196 | 0.309 | 0.223 | 0.302 | 0.225 | 0.319 | 0.214 | 0.327 | 0.193 | 0.296 | 0.227 | 0.338 | 0.311 | 0.397 |
| Traffic | 96 | 0.462 | 0.285 | 0.526 | 0.347 | 0.593 | 0.321 | 0.644 | 0.429 | 0.577 | 0.350 | 0.647 | 0.384 | 0.650 | 0.396 | 0.587 | 0.366 | 0.612 | 0.338 | 0.613 | 0.388 | 0.719 | 0.391 |
| | 192 | 0.473 | 0.296 | 0.522 | 0.332 | 0.617 | 0.336 | 0.665 | 0.431 | 0.589 | 0.356 | 0.600 | 0.361 | 0.598 | 0.370 | 0.604 | 0.373 | 0.613 | 0.340 | 0.616 | 0.382 | 0.696 | 0.379 |
| | 336 | 0.498 | 0.296 | 0.517 | 0.334 | 0.629 | 0.336 | 0.674 | 0.420 | 0.594 | 0.358 | 0.610 | 0.367 | 0.605 | 0.373 | 0.621 | 0.383 | 0.618 | 0.328 | 0.622 | 0.337 | 0.777 | 0.420 |
| | 720 | 0.506 | 0.313 | 0.552 | 0.352 | 0.640 | 0.350 | 0.683 | 0.424 | 0.613 | 0.361 | 0.691 | 0.425 | 0.645 | 0.394 | 0.626 | 0.382 | 0.653 | 0.355 | 0.660 | 0.408 | 0.864 | 0.472 |
| | Avg | 0.484 | 0.297 | 0.529 | 0.341 | 0.620 | 0.336 | 0.667 | 0.426 | 0.593 | 0.356 | 0.637 | 0.384 | 0.625 | 0.383 | 0.610 | 0.376 | 0.624 | 0.340 | 0.628 | 0.379 | 0.764 | 0.416 |
| ETTh1 | 96 | 0.375 | 0.400 | 0.460 | 0.447 | 0.384 | 0.402 | 0.423 | 0.448 | 0.426 | 0.446 | 0.438 | 0.433 | 0.397 | 0.412 | 0.395 | 0.424 | 0.513 | 0.491 | 0.449 | 0.459 | 0.865 | 0.713 |
| | 192 | 0.429 | 0.421 | 0.512 | 0.477 | 0.436 | 0.429 | 0.471 | 0.474 | 0.454 | 0.464 | 0.493 | 0.466 | 0.446 | 0.441 | 0.469 | 0.470 | 0.534 | 0.504 | 0.500 | 0.482 | 1.008 | 0.792 |
| | 336 | 0.484 | 0.458 | 0.546 | 0.496 | 0.638 | 0.469 | 0.570 | 0.546 | 0.493 | 0.487 | 0.547 | 0.495 | 0.489 | 0.467 | 0.530 | 0.499 | 0.588 | 0.535 | 0.521 | 0.496 | 1.107 | 0.809 |
| | 720 | 0.498 | 0.482 | 0.544 | 0.517 | 0.521 | 0.500 | 0.653 | 0.621 | 0.526 | 0.526 | 0.586 | 0.538 | 0.513 | 0.510 | 0.598 | 0.544 | 0.643 | 0.616 | 0.514 | 0.512 | 1.181 | 0.865 |
| | Avg | 0.447 | 0.440 | 0.516 | 0.484 | 0.495 | 0.450 | 0.529 | 0.522 | 0.475 | 0.480 | 0.516 | 0.483 | 0.461 | 0.457 | 0.498 | 0.484 | 0.570 | 0.537 | 0.496 | 0.487 | 1.040 | 0.795 |
| ETTh2 | 96 | 0.289 | 0.341 | 0.308 | 0.355 | 0.340 | 0.374 | 0.745 | 0.584 | 0.372 | 0.424 | 0.322 | 0.364 | 0.340 | 0.394 | 0.358 | 0.397 | 0.476 | 0.458 | 0.346 | 0.388 | 3.755 | 1.525 |
| | 192 | 0.372 | 0.392 | 0.393 | 0.405 | 0.402 | 0.414 | 0.877 | 0.656 | 0.492 | 0.492 | 0.404 | 0.414 | 0.482 | 0.479 | 0.429 | 0.439 | 0.512 | 0.493 | 0.456 | 0.452 | 5.602 | 1.931 |
| | 336 | 0.386 | 0.414 | 0.427 | 0.436 | 0.452 | 0.452 | 1.043 | 0.731 | 0.607 | 0.555 | 0.435 | 0.445 | 0.591 | 0.541 | 0.496 | 0.487 | 0.552 | 0.551 | 0.482 | 0.486 | 4.721 | 1.835 |
| | 720 | 0.412 | 0.434 | 0.436 | 0.450 | 0.462 | 0.468 | 1.104 | 0.763 | 0.824 | 0.655 | 0.447 | 0.458 | 0.839 | 0.661 | 0.463 | 0.474 | 0.562 | 0.560 | 0.515 | 0.511 | 3.647 | 1.625 |
| | Avg | 0.364 | 0.395 | 0.391 | 0.411 | 0.414 | 0.427 | 0.942 | 0.684 | 0.574 | 0.531 | 0.402 | 0.420 | 0.563 | 0.519 | 0.437 | 0.449 | 0.526 | 0.516 | 0.450 | 0.459 | 4.431 | 1.729 |
| ETTm1 | 96 | 0.320 | 0.357 | 0.352 | 0.374 | 0.338 | 0.375 | 0.404 | 0.426 | 0.365 | 0.387 | 0.353 | 0.370 | 0.346 | 0.374 | 0.379 | 0.419 | 0.386 | 0.398 | 0.505 | 0.475 | 0.672 | 0.571 |
| | 192 | 0.361 | 0.381 | 0.390 | 0.393 | 0.374 | 0.387 | 0.450 | 0.451 | 0.403 | 0.408 | 0.389 | 0.387 | 0.382 | 0.391 | 0.426 | 0.441 | 0.459 | 0.444 | 0.553 | 0.496 | 0.795 | 0.669 |
| | 336 | 0.390 | 0.404 | 0.421 | 0.414 | 0.410 | 0.411 | 0.532 | 0.515 | 0.436 | 0.431 | 0.421 | 0.408 | 0.415 | 0.415 | 0.445 | 0.459 | 0.495 | 0.464 | 0.621 | 0.537 | 1.212 | 0.871 |
| | 720 | 0.454 | 0.441 | 0.462 | 0.449 | 0.478 | 0.450 | 0.666 | 0.589 | 0.489 | 0.462 | 0.481 | 0.441 | 0.473 | 0.451 | 0.543 | 0.490 | 0.585 | 0.516 | 0.671 | 0.561 | 1.166 | 0.823 |
| | Avg | 0.381 | 0.395 | 0.406 | 0.407 | 0.400 | 0.406 | 0.513 | 0.495 | 0.423 | 0.422 | 0.411 | 0.402 | 0.404 | 0.408 | 0.448 | 0.452 | 0.481 | 0.456 | 0.588 | 0.517 | 0.961 | 0.734 |
| ETTm2 | 96 | 0.175 | 0.258 | 0.183 | 0.270 | 0.187 | 0.267 | 0.287 | 0.366 | 0.197 | 0.296 | 0.183 | 0.266 | 0.193 | 0.293 | 0.203 | 0.287 | 0.192 | 0.274 | 0.255 | 0.339 | 0.365 | 0.453 |
| | 192 | 0.237 | 0.299 | 0.255 | 0.314 | 0.249 | 0.309 | 0.414 | 0.492 | 0.284 | 0.361 | 0.248 | 0.305 | 0.284 | 0.361 | 0.269 | 0.328 | 0.280 | 0.339 | 0.281 | 0.340 | 0.533 | 0.563 |
| | 336 | 0.298 | 0.340 | 0.309 | 0.347 | 0.321 | 0.351 | 0.597 | 0.542 | 0.381 | 0.429 | 0.309 | 0.343 | 0.382 | 0.429 | 0.325 | 0.366 | 0.334 | 0.361 | 0.339 | 0.372 | 1.363 | 0.887 |
| | 720 | 0.391 | 0.396 | 0.412 | 0.404 | 0.408 | 0.403 | 1.730 | 1.042 | 0.549 | 0.522 | 0.410 | 0.400 | 0.558 | 0.525 | 0.421 | 0.415 | 0.417 | 0.413 | 0.433 | 0.432 | 3.379 | 1.338 |
| | Avg | 0.275 | 0.323 | 0.290 | 0.334 | 0.291 | 0.333 | 0.757 | 0.610 | 0.353 | 0.402 | 0.287 | 0.329 | 0.354 | 0.402 | 0.305 | 0.349 | 0.306 | 0.347 | 0.327 | 0.371 | 1.410 | 0.810 |

Table 14: Experiment results under hyperparameter searching for the long-term forecasting task. *Avg* is averaged from all four prediction lengths.

| Models | | TimeMixer (Ours) | | PatchTST 2023 | | TimesNet 2023a | | Crossformer 2023 | | MICN 2023 | | FiLM 2022a | | DLinear 2023 | | FEDformer 2022b | | Stationary 2022b | | Autoformer 2021 | | Informer 2021 | |
|---|---|---|---|---|---|---|---|---|---|---|---|---|---|---|---|---|---|---|---|---|---|---|---|
| Metric | | MSE | MAE | MSE | MAE | MSE | MAE | MSE | MAE | MSE | MAE | MSE | MAE | MSE | MAE | MSE | MAE | MSE | MAE | MSE | MAE | MSE | MAE |
| Weather | 96 | 0.147 | 0.197 | 0.149 | 0.198 | 0.172 | 0.220 | 0.232 | 0.302 | 0.161 | 0.229 | 0.199 | 0.262 | 0.176 | 0.237 | 0.217 | 0.296 | 0.173 | 0.223 | 0.266 | 0.336 | 0.300 | 0.384 |
| | 192 | 0.189 | 0.239 | 0.194 | 0.241 | 0.219 | 0.261 | 0.371 | 0.410 | 0.220 | 0.281 | 0.228 | 0.288 | 0.220 | 0.282 | 0.276 | 0.336 | 0.245 | 0.285 | 0.307 | 0.367 | 0.598 | 0.544 |
| | 336 | 0.241 | 0.280 | 0.306 | 0.282 | 0.246 | 0.337 | 0.495 | 0.515 | 0.278 | 0.331 | 0.267 | 0.323 | 0.265 | 0.319 | 0.339 | 0.380 | 0.321 | 0.338 | 0.359 | 0.395 | 0.578 | 0.523 |
| | 720 | 0.310 | 0.330 | 0.314 | 0.334 | 0.365 | 0.359 | 0.526 | 0.542 | 0.311 | 0.356 | 0.319 | 0.361 | 0.323 | 0.362 | 0.403 | 0.428 | 0.414 | 0.410 | 0.419 | 0.428 | 1.059 | 0.741 |
| | Avg | 0.222 | 0.262 | 0.241 | 0.264 | 0.251 | 0.294 | 0.406 | 0.442 | 0.242 | 0.299 | 0.253 | 0.309 | 0.246 | 0.300 | 0.309 | 0.360 | 0.288 | 0.314 | 0.338 | 0.382 | 0.634 | 0.548 |
| Solar-Energy | 96 | 0.167 | 0.220 | 0.224 | 0.278 | 0.219 | 0.314 | 0.181 | 0.240 | 0.188 | 0.252 | 0.320 | 0.339 | 0.289 | 0.377 | 0.201 | 0.304 | 0.321 | 0.380 | 0.456 | 0.446 | 0.200 | 0.247 |
| | 192 | 0.187 | 0.249 | 0.253 | 0.298 | 0.231 | 0.322 | 0.196 | 0.252 | 0.215 | 0.280 | 0.360 | 0.362 | 0.319 | 0.397 | 0.237 | 0.337 | 0.346 | 0.369 | 0.588 | 0.561 | 0.220 | 0.251 |
| | 336 | 0.200 | 0.258 | 0.273 | 0.306 | 0.246 | 0.337 | 0.216 | 0.243 | 0.222 | 0.267 | 0.398 | 0.375 | 0.352 | 0.415 | 0.254 | 0.362 | 0.357 | 0.387 | 0.595 | 0.588 | 0.260 | 0.287 |
| | 720 | 0.215 | 0.250 | 0.272 | 0.308 | 0.280 | 0.363 | 0.220 | 0.256 | 0.226 | 0.264 | 0.399 | 0.368 | 0.356 | 0.412 | 0.280 | 0.397 | 0.335 | 0.384 | 0.733 | 0.633 | 0.244 | 0.301 |
| | Avg | 0.192 | 0.244 | 0.256 | 0.298 | 0.244 | 0.334 | 0.204 | 0.248 | 0.213 | 0.266 | 0.369 | 0.361 | 0.329 | 0.400 | 0.243 | 0.350 | 0.340 | 0.380 | 0.593 | 0.557 | 0.231 | 0.272 |
| Electricity | 96 | 0.129 | 0.224 | 0.129 | 0.222 | 0.168 | 0.272 | 0.150 | 0.251 | 0.164 | 0.269 | 0.154 | 0.267 | 0.140 | 0.237 | 0.193 | 0.308 | 0.169 | 0.273 | 0.201 | 0.317 | 0.274 | 0.368 |
| | 192 | 0.140 | 0.220 | 0.147 | 0.240 | 0.184 | 0.322 | 0.161 | 0.260 | 0.177 | 0.285 | 0.164 | 0.258 | 0.153 | 0.249 | 0.201 | 0.315 | 0.182 | 0.286 | 0.222 | 0.334 | 0.296 | 0.386 |
| | 336 | 0.161 | 0.255 | 0.163 | 0.259 | 0.198 | 0.300 | 0.182 | 0.281 | 0.193 | 0.304 | 0.188 | 0.283 | 0.169 | 0.267 | 0.214 | 0.329 | 0.200 | 0.304 | 0.231 | 0.338 | 0.300 | 0.394 |
| | 720 | 0.194 | 0.287 | 0.197 | 0.290 | 0.220 | 0.320 | 0.251 | 0.339 | 0.212 | 0.321 | 0.236 | 0.332 | 0.203 | 0.301 | 0.246 | 0.355 | 0.222 | 0.321 | 0.254 | 0.361 | 0.373 | 0.439 |
| | Avg | 0.156 | 0.246 | 0.159 | 0.253 | 0.192 | 0.295 | 0.186 | 0.283 | 0.186 | 0.295 | 0.186 | 0.285 | 0.166 | 0.264 | 0.214 | 0.321 | 0.213 | 0.296 | 0.227 | 0.338 | 0.311 | 0.397 |
| Traffic | 96 | 0.360 | 0.249 | 0.360 | 0.249 | 0.593 | 0.321 | 0.514 | 0.267 | 0.519 | 0.309 | 0.416 | 0.294 | 0.410 | 0.282 | 0.587 | 0.366 | 0.612 | 0.338 | 0.613 | 0.388 | 0.719 | 0.391 |
| | 192 | 0.375 | 0.250 | 0.379 | 0.256 | 0.617 | 0.336 | 0.549 | 0.252 | 0.537 | 0.315 | 0.408 | 0.288 | 0.423 | 0.287 | 0.604 | 0.373 | 0.613 | 0.340 | 0.616 | 0.382 | 0.696 | 0.379 |
| | 336 | 0.385 | 0.270 | 0.392 | 0.264 | 0.629 | 0.336 | 0.530 | 0.300 | 0.534 | 0.313 | 0.425 | 0.298 | 0.436 | 0.296 | 0.621 | 0.383 | 0.618 | 0.328 | 0.622 | 0.337 | 0.777 | 0.420 |
| | 720 | 0.430 | 0.281 | 0.432 | 0.286 | 0.640 | 0.350 | 0.573 | 0.313 | 0.577 | 0.325 | 0.520 | 0.353 | 0.466 | 0.315 | 0.626 | 0.382 | 0.653 | 0.355 | 0.660 | 0.408 | 0.864 | 0.472 |
| | Avg | 0.387 | 0.262 | 0.391 | 0.264 | 0.620 | 0.336 | 0.542 | 0.283 | 0.541 | 0.315 | 0.442 | 0.308 | 0.434 | 0.295 | 0.609 | 0.376 | 0.624 | 0.340 | 0.628 | 0.379 | 0.764 | 0.415 |
| ETTh1 | 96 | 0.361 | 0.390 | 0.370 | 0.400 | 0.384 | 0.402 | 0.418 | 0.438 | 0.421 | 0.431 | 0.422 | 0.432 | 0.375 | 0.399 | 0.376 | 0.419 | 0.513 | 0.491 | 0.449 | 0.459 | 0.865 | 0.713 |
| | 192 | 0.409 | 0.414 | 0.413 | 0.429 | 0.436 | 0.429 | 0.539 | 0.517 | 0.474 | 0.487 | 0.462 | 0.458 | 0.405 | 0.416 | 0.420 | 0.448 | 0.534 | 0.504 | 0.500 | 0.482 | 1.008 | 0.792 |
| | 336 | 0.430 | 0.429 | 0.422 | 0.440 | 0.638 | 0.469 | 0.709 | 0.638 | 0.569 | 0.551 | 0.501 | 0.483 | 0.439 | 0.443 | 0.459 | 0.465 | 0.588 | 0.535 | 0.521 | 0.496 | 1.107 | 0.809 |
| | 720 | 0.445 | 0.460 | 0.447 | 0.468 | 0.521 | 0.500 | 0.733 | 0.636 | 0.770 | 0.672 | 0.544 | 0.526 | 0.472 | 0.490 | 0.506 | 0.507 | 0.643 | 0.616 | 0.514 | 0.512 | 1.181 | 0.865 |
| | Avg | 0.411 | 0.423 | 0.413 | 0.434 | 0.458 | 0.450 | 0.600 | 0.557 | 0.558 | 0.535 | 0.482 | 0.475 | 0.423 | 0.437 | 0.440 | 0.460 | 0.57 | 0.536 | 0.496 | 0.487 | 1.040 | 0.795 |
| ETTh2 | 96 | 0.271 | 0.330 | 0.274 | 0.337 | 0.340 | 0.374 | 0.425 | 0.463 | 0.299 | 0.364 | 0.323 | 0.370 | 0.289 | 0.353 | 0.346 | 0.388 | 0.476 | 0.458 | 0.358 | 0.397 | 3.755 | 1.525 |
| | 192 | 0.317 | 0.402 | 0.314 | 0.382 | 0.231 | 0.322 | 0.473 | 0.500 | 0.441 | 0.454 | 0.391 | 0.415 | 0.383 | 0.418 | 0.429 | 0.439 | 0.512 | 0.493 | 0.456 | 0.452 | 5.602 | 1.931 |
| | 336 | 0.332 | 0.396 | 0.329 | 0.384 | 0.452 | 0.452 | 0.581 | 0.562 | 0.654 | 0.567 | 0.415 | 0.440 | 0.448 | 0.465 | 0.496 | 0.487 | 0.552 | 0.551 | 0.482 | 0.486 | 4.721 | 1.835 |
| | 720 | 0.342 | 0.408 | 0.379 | 0.422 | 0.462 | 0.468 | 0.775 | 0.665 | 0.956 | 0.716 | 0.441 | 0.459 | 0.605 | 0.551 | 0.463 | 0.474 | 0.562 | 0.560 | 0.515 | 0.511 | 3.647 | 1.625 |
| | Avg | 0.316 | 0.384 | 0.324 | 0.381 | 0.371 | 0.404 | 0.564 | 0.548 | 0.588 | 0.525 | 0.393 | 0.421 | 0.431 | 0.447 | 0.433 | 0.447 | 0.526 | 0.516 | 0.453 | 0.462 | 4.431 | 1.729 |
| ETTm1 | 96 | 0.291 | 0.340 | 0.293 | 0.346 | 0.338 | 0.375 | 0.361 | 0.403 | 0.316 | 0.362 | 0.302 | 0.345 | 0.299 | 0.343 | 0.379 | 0.419 | 0.386 | 0.398 | 0.505 | 0.475 | 0.672 | 0.571 |
| | 192 | 0.327 | 0.365 | 0.333 | 0.370 | 0.374 | 0.387 | 0.387 | 0.422 | 0.363 | 0.390 | 0.338 | 0.368 | 0.335 | 0.365 | 0.426 | 0.441 | 0.459 | 0.444 | 0.553 | 0.496 | 0.795 | 0.669 |
| | 336 | 0.360 | 0.381 | 0.369 | 0.392 | 0.410 | 0.411 | 0.605 | 0.572 | 0.408 | 0.426 | 0.373 | 0.388 | 0.369 | 0.386 | 0.445 | 0.459 | 0.495 | 0.464 | 0.621 | 0.537 | 1.212 | 0.871 |
| | 720 | 0.415 | 0.417 | 0.416 | 0.420 | 0.478 | 0.450 | 0.703 | 0.645 | 0.481 | 0.476 | 0.420 | 0.420 | 0.425 | 0.421 | 0.543 | 0.490 | 0.585 | 0.516 | 0.671 | 0.561 | 1.166 | 0.823 |
| | Avg | 0.348 | 0.375 | 0.353 | 0.382 | 0.353 | 0.382 | 0.514 | 0.510 | 0.392 | 0.413 | 0.358 | 0.38 | 0.357 | 0.379 | 0.448 | 0.452 | 0.481 | 0.456 | 0.588 | 0.517 | 0.961 | 0.733 |
| ETTm2 | 96 | 0.164 | 0.254 | 0.166 | 0.256 | 0.187 | 0.267 | 0.275 | 0.358 | 0.179 | 0.275 | 0.165 | 0.256 | 0.167 | 0.260 | 0.203 | 0.287 | 0.192 | 0.274 | 0.255 | 0.339 | 0.365 | 0.453 |
| | 192 | 0.223 | 0.295 | 0.223 | 0.296 | 0.249 | 0.309 | 0.345 | 0.400 | 0.307 | 0.376 | 0.222 | 0.296 | 0.224 | 0.303 | 0.269 | 0.328 | 0.280 | 0.339 | 0.281 | 0.340 | 0.533 | 0.563 |
| | 336 | 0.279 | 0.330 | 0.274 | 0.329 | 0.321 | 0.351 | 0.657 | 0.528 | 0.325 | 0.388 | 0.277 | 0.333 | 0.281 | 0.342 | 0.325 | 0.366 | 0.334 | 0.361 | 0.339 | 0.372 | 1.363 | 0.887 |
| | 720 | 0.359 | 0.383 | 0.362 | 0.385 | 0.408 | 0.403 | 1.208 | 0.753 | 0.502 | 0.490 | 0.371 | 0.389 | 0.397 | 0.421 | 0.421 | 0.415 | 0.417 | 0.413 | 0.422 | 0.419 | 3.379 | 1.388 |
| | Avg | 0.256 | 0.315 | 0.256 | 0.317 | 0.291 | 0.333 | 0.621 | 0.510 | 0.328 | 0.382 | 0.259 | 0.319 | 0.267 | 0.332 | 0.304 | 0.349 | 0.306 | 0.347 | 0.324 | 0.368 | 1.410 | 0.823 |

# F FULL ABLATIONS

Here we provide the complete results of ablations and alternative designs for TimeMixer.

## F.1 ABLATIONS OF EACH DESIGN IN TIMEMIXER

To verify the effectiveness of our design in TimeMixer, we conduct comprehensive ablations for all benchmarks. All the results are provided in Table 15, 16, 17 as a supplement to Table 5 of main text.

Table 15: Ablations on both Past-Decompose-Mixing (*Decompose*, *Season Mixing*, and *Trend Mixing*) and Future-Multipredictor-Mixing blocks in predict-336 setting for all long-term benchmarks. ↗ indicates the bottom-up mixing while ↙ indicates top-down. A check mark ✓ and a wrong mark × indicate with and without certain components respectively. ① is the official design in TimeMixer.

| Case | Decompose | Past mixing Seasonal | Trend | Future mixing Multipredictor | ETTh1 MSE | MAE | ETTh2 MSE | MAE | ETTm1 MSE | MAE | ETTm2 MSE | MAE | Weather MSE | MAE | Solar MSE | MAE | Electricity MSE | MAE | Traffic MSE | MAE |
|---|---|---|---|---|---|---|---|---|---|---|---|---|---|---|---|---|---|---|---|---|
| ① | ✓ | ↗ | ↙ | ✓ | **0.484** | **0.458** | **0.386** | **0.414** | **0.390** | **0.404** | **0.298** | **0.340** | **0.251** | **0.287** | **0.231** | **0.292** | **0.185** | **0.277** | **0.498** | **0.296** |
| ② | ✓ | ↗ | ↙ | × | 0.493 | 0.472 | 0.399 | 0.426 | 0.402 | 0.415 | 0.311 | 0.357 | 0.262 | 0.308 | 0.267 | 0.339 | 0.198 | 0.301 | 0.518 | 0.337 |
| ③ | ✓ | × | ↙ | ✓ | 0.507 | 0.490 | 0.408 | 0.437 | 0.411 | 0.427 | 0.322 | 0.366 | 0.273 | 0.321 | 0.274 | 0.355 | 0.207 | 0.304 | 0.532 | 0.348 |
| ④ | ✓ | ↗ | × | ✓ | 0.491 | 0.483 | 0.397 | 0.424 | 0.405 | 0.414 | 0.317 | 0.351 | 0.269 | 0.311 | 0.268 | 0.341 | 0.200 | 0.299 | 0.525 | 0.339 |
| ⑤ | ✓ | ↙ | ↙ | ✓ | 0.488 | 0.466 | 0.393 | 0.426 | 0.392 | 0.413 | 0.309 | 0.349 | 0.257 | 0.293 | 0.252 | 0.330 | 0.191 | 0.293 | 0.520 | 0.331 |
| ⑥ | ✓ | ↗ | ↗ | ✓ | 0.493 | 0.484 | 0.401 | 0.432 | 0.396 | 0.415 | 0.319 | 0.361 | 0.271 | 0.322 | 0.281 | 0.363 | 0.214 | 0.307 | 0.541 | 0.351 |
| ⑦ | ✓ | ↙ | ↗ | ✓ | 0.498 | 0.491 | 0.421 | 0.436 | 0.412 | 0.429 | 0.321 | 0.369 | 0.277 | 0.332 | 0.298 | 0.375 | 0.221 | 0.319 | 0.564 | 0.357 |
| ⑧ | × | ↗ | | ✓ | 0.494 | 0.488 | 0.396 | 0.421 | 0.395 | 0.408 | 0.313 | 0.360 | 0.259 | 0.308 | 0.260 | 0.321 | 0.199 | 0.303 | 0.522 | 0.340 |
| ⑨ | × | ↙ | | ✓ | 0.487 | 0.462 | 0.394 | 0.419 | 0.393 | 0.406 | 0.307 | 0.354 | 0.261 | 0.327 | 0.257 | 0.334 | 0.196 | 0.310 | 0.526 | 0.339 |
| ⑩ | × | × | | ✓ | 0.502 | 0.489 | 0.411 | 0.427 | 0.405 | 0.412 | 0.319 | 0.358 | 0.273 | 0.331 | 0.295 | 0.336 | 0.217 | 0.318 | 0.558 | 0.347 |

Table 16: Ablations on both Past-Decompose-Mixing (*Decompose*, *Season Mixing*, and *Trend Mixing*) and Future-Multipredictor-Mixing blocks in the M4 short-term forecasting benchmark. Case notations are same as Table 5 and 15. ① is the official design in TimeMixer.

| Case | Decompose | Past mixing Seasonal | Trend | Future mixing Multipredictor | M4 SMAPE | MASE | OWA |
|---|---|---|---|---|---|---|---|
| ① | ✓ | ↗ | ↙ | ✓ | **11.723** | **1.559** | **0.840** |
| ② | ✓ | ↗ | ↙ | × | 12.503 | 1.634 | 0.925 |
| ③ | ✓ | × | ↙ | ✓ | 13.051 | 1.676 | 0.962 |
| ④ | ✓ | ↗ | × | ✓ | 12.911 | 1.655 | 0.941 |
| ⑤ | ✓ | ↙ | ↙ | ✓ | 12.008 | 1.628 | 0.871 |
| ⑥ | ✓ | ↗ | ↗ | ✓ | 11.978 | 1.626 | 0.859 |
| ⑦ | ✓ | ↙ | ↗ | ✓ | 13.012 | 1.657 | 0.954 |
| ⑧ | × | ↗ | | ✓ | 11.975 | 1.617 | 0.851 |
| ⑨ | × | ↙ | | ✓ | 11.973 | 1.622 | 0.850 |
| ⑩ | × | × | | ✓ | 12.468 | 1.671 | 0.916 |

Table 17: Ablations on both Past-Decompose-Mixing (*Decompose*, *Season Mixing*, and *Trend Mixing*) and Future-Multipredictor-Mixing blocks in the PEMS short-term forecasting benchmarks. Case notations are same as Table 5 and 15. ① is the official design in TimeMixer.

| Case | Decompose | Past mixing Seasonal | Trend | Future mixing Multipredictor | PEMS03 MAE | MAPE | RMSE | PEMS04 MAE | MAPE | RMSE | PEMS07 MAE | MAPE | RMSE | PEMS08 MAE | MAPE | RMSE |
|---|---|---|---|---|---|---|---|---|---|---|---|---|---|---|---|---|
| ① | ✓ | ↗ | ↙ | ✓ | **14.63** | **14.54** | **23.28** | **19.21** | **12.53** | **30.92** | **20.57** | **8.62** | **33.59** | **15.22** | **9.67** | **24.26** |
| ② | ✓ | ↗ | ↙ | × | 15.66 | 15.81 | 25.77 | 21.67 | 13.45 | 34.89 | 22.78 | 9.52 | 35.57 | 17.48 | 10.91 | 27.84 |
| ③ | ✓ | × | ↙ | ✓ | 18.90 | 17.33 | 30.75 | 24.49 | 16.28 | 38.79 | 25.27 | 10.74 | 40.06 | 19.02 | 11.71 | 30.05 |
| ④ | ✓ | ↗ | × | ✓ | 17.67 | 17.58 | 28.48 | 22.91 | 15.02 | 37.14 | 24.81 | 10.02 | 38.68 | 18.29 | 12.21 | 28.62 |
| ⑤ | ✓ | ↙ | ↙ | ✓ | 15.46 | 15.73 | 24.91 | 20.78 | 13.02 | 32.47 | 22.57 | 9.33 | 35.87 | 16.54 | 9.97 | 26.88 |
| ⑥ | ✓ | ↗ | ↗ | ✓ | 15.32 | 15.41 | 24.83 | 21.09 | 13.78 | 33.11 | 21.94 | 9.41 | 35.40 | 17.01 | 10.82 | 26.93 |
| ⑦ | ✓ | ↙ | ↗ | ✓ | 18.81 | 17.29 | 29.78 | 22.27 | 15.14 | 34.67 | 25.11 | 10.60 | 39.74 | 18.74 | 12.09 | 28.67 |
| ⑧ | × | ↗ | | ✓ | 15.57 | 15.62 | 24.98 | 21.51 | 13.47 | 34.81 | 22.94 | 9.81 | 35.49 | 18.17 | 11.02 | 28.14 |
| ⑨ | × | ↙ | | ✓ | 15.48 | 15.55 | 24.83 | 21.79 | 14.03 | 35.23 | 21.93 | 9.91 | 36.02 | 17.71 | 10.88 | 27.91 |
| ⑩ | × | × | | ✓ | 19.01 | 18.58 | 30.06 | 24.87 | 16.66 | 39.48 | 24.72 | 9.97 | 37.18 | 19.18 | 12.21 | 30.79 |

**Implementations**  We implement the following 10 types of ablations:

- Offical design in TimeMixer (case ①).

- Ablations on Future Mixing (case ②): In this case, we only adopt a single predictor to the finest scale features, that is $\hat{\mathbf{x}} = \text{Predictor}_0(\mathbf{x}_0^L)$.

- Ablations on Past Mixing (case ③-⑦): Firstly, we remove the mixing operation of TimeMixer in seasonal and trend parts respectively (case ③-④), that is removing Bottom-Up-Mixing or Top-Down-Mixing layer. Then, we reverse the mixing directions for seasonal and trend parts (case ⑤-⑦), which means adopting Bottom-Up-Mixing layer to trend and Top-Down-Mixing layer to seasonal part.

- Ablations on Decomposition (case ⑧-⑩): In these cases, we do not adopt the decomposition, which means that there is only one single feature for each scale. Thus, we can only try one single mixing direction for these features, that is bottom-up mixing in case ⑧, top-down mixing in case ⑨. Besides, we also test the case that is without mixing in ⑩, where the interactions among multiscale features are removed.

**Analysis**  In all ablations, we can find that the official design in TimeMixer performs best, which provides solid support to our insights in special mixing approaches. Notably, it is observed that completely reversing mixing directions for seasonal and trend parts (case ⑦) leads to a seriously performance drop. This may come from that the essential microscopic information in finer-scale seasons and macroscopic information in coarser-scale trends are ruined by unsuitable mixing approaches.

### F.2  ALTERNATIVE DECOMPOSITION METHODS

In this paper, we adopt the moving-average-based season-trend decomposition, which is widely used in previous work, such as Autoformer (Wu et al., 2021), FEDformer (Zhou et al., 2022b) and DLinear (Zeng et al., 2023). It is notable that Discrete Fourier Transformer (DFT) have been widely recognized in time series analysis. Thus, we also try the DFT-based decomposition as a substitute. Here we present two types of experiments.

The first one is *DFT-based high- and low-frequency decomposition*. We treat the high-frequency part like the seasonal part in TimeMixer and the low-frequency part like the trend part. The results are shown in Table 18. It observed that DFT-based decomposition performs worse than our design in TimeMixer. Since we only explore the proper mixing approach for decomposed seasonal and trend parts in the paper, the bottom-up and top-down mixing strategies may be not suitable for high- and low-frequency parts. New visualizations like Figure 3 and 4 are expected to provide insights to the model design. Thus, we would like to leave the exploration of DFT-based high- and low-frequency decomposition methods as the future work.

The second one is to enhance season-trend decomposition with DFT. Here we present the *DFT-based season-trend decomposition*. Firstly, we transform the raw series into a frequency domain by DFT and then extract the most significant frequencies. After transforming the selected frequencies by inverse DFT, we obtain the seasonal part of the time series. Then the trend part is the raw series minus the seasonal part. We can find that this superior decomposition method surpasses the moving-average design. However, since moving average is quite simple and easy to implement with PyTorch, we eventually chose the moving-average-based season-trend decomposition in TimeMixer, which can also achieve a favorable balance between performance and efficiency.

Table 18: Alternative decomposition methods in M4, PEMS04 and predict-336 setting of ETTm1.

| Decomposition methods | ETTm1 | | M4 | | | PEMS04 | | |
|---|---|---|---|---|---|---|---|---|
| | MSE | MAE | SMAPE | MASE | OWA | MAE | MAPE | RMSE |
| DFT-based high- and low-frequency decomposition | 0.392 | 0.404 | 12.054 | 1.632 | 0.862 | 19.83 | 12.74 | 31.48 |
| DFT-based season-trend decomposition | 0.383 | 0.399 | 11.673 | 1.536 | 0.824 | 18.91 | 12.27 | 29.47 |
| Moving-average-based season-trend decomposition (TimeMixer) | 0.390 | 0.404 | 11.723 | 1.559 | 0.840 | 19.21 | 12.53 | 30.92 |

### F.3 Alternative Downsampling Methods

As we stated in Section 3.1, we adopt the average pooling to obtain the multiscale series. Here we replace this operation with 1D convolutions. From Table 19, we can find that the complicated 1D-convolution-based outperforms average pooling slightly. But considering both performance and efficiency, we eventually use average pooling in TimeMixer.

Table 19: Alternative downsampling methods in M4, PEMS04 and predict-336 setting of ETTm1.

| Downsampling methods | ETTm1 | | M4 | | | PEMS04 | | |
|---|---|---|---|---|---|---|---|---|
| | MSE | MAE | SMAPE | MASE | OWA | MAE | MAPE | RMSE |
| Moving average | 0.390 | 0.404 | 11.723 | 1.559 | 0.840 | 19.21 | 12.53 | 30.92 |
| 1D convolutions with stride as 2 | 0.387 | 0.401 | 11.682 | 1.542 | 0.831 | 19.04 | 12.17 | 29.88 |

### F.4 Alternative Ensemble Strategies

In the main text, we sum the outputs from multiple predictors towards the final result (Eq. 6). Here we also try the average strategy. Note that in TimeMixer, the loss is calculated based on the ensemble results, not for each predictor, that is $\|\mathbf{x} - \hat{\mathbf{x}}\| = \|\mathbf{x} - \sum_{m=0}^{M} \hat{\mathbf{x}}_m\|$. When we change the ensemble strategy as average, the loss will be $\|\mathbf{x} - \hat{\mathbf{x}}\| = \|\mathbf{x} - \frac{1}{M+1}\sum_{m=0}^{M} \hat{\mathbf{x}}_m\|$. Obviously, the difference between average and mean strategies is only a constant multiple.

It is common sense in the deep learning community that deep models can easily fit constant multiple. For example, if we replace the "sum" with "average", under the same supervision, the deep model can easily fit this change by learning the parameters of each predictor equal to the $\frac{1}{M+1}$ of the "sum" case, which means these two designs are equivalent in learning the final prediction under the deep model aspect. Besides, we also provide the experiment results in Table 20, where we can find that the performances of these two strategies are almost the same.

Table 20: Alternative ensemble strategies in M4, PEMS04 and predict-336 setting of ETTm1.

| Ensemble strategies | ETTm1 | | M4 | | | PEMS04 | | |
|---|---|---|---|---|---|---|---|---|
| | MSE | MAE | SMAPE | MASE | OWA | MAE | MAPE | RMSE |
| Sum ensemble | 0.390 | 0.404 | 11.723 | 1.559 | 0.840 | 19.21 | 12.53 | 30.92 |
| Average ensemble | 0.391 | 0.407 | 11.742 | 1.573 | 0.851 | 19.17 | 12.45 | 30.88 |

### F.5 Ablations on Larger Scales and Larger Input Length Settings

In the previous section (Table 15, 16, 17), we have conducted comprehensive ablations under the unified configuration presented in Table 7, which $M$ is set to 1 for short-term forecasting and input length is set to 96 for short-term forecasting. To further evaluate the effectiveness of our proposed module, we also provide additional ablations on larger scales for short-term forecasting and larger input length settings in Table 21 as a supplement to Table 5 of the main text. Besides, we also provide a detailed analysis of relative promotion (Table 22), where we can find the following observations:

- All the designs of TimeMixer are effective in both hyperparameter settings. Especially, the seasonal mixing (case ③) and proper mixing directions (case ⑦) are essential.
- As shown in Table 22, in large $M$ and longer input length situations, the relative promotions brought by seasonal and trend mixing in PDM and FMM are more significant in most cases, which further verifies the effectiveness of our design.
- It is observed that the seasonal mixing direction contribution (case ⑤) is much more significant in the longer-input setting on the ETTm1 dataset. This may come from that input-96 only corresponds to one day in 15-minutely sampled ETTm1, while input-336 maintains 3.5 days of information (around 3.5 periods). Thus, the bottom-up mixing direction will benefit from sufficient microscopic seasonal information under the longer-input setting.

Table 21: A supplement to Table 5 of the main text with ablations on both PDM (*Decompose*, *Season Mixing*, *Trend Mixing*) and FMM blocks in PEMS04 with $M = 3$ and predict-336 setting of ETTm1 with input-336. Since the input length of M4 is fixed to a small value, larger $M$ may result in a meaningless configuration. We only experiment on PEMS04 here.

| Case | Decompose | Past mixing | | Future mixing | PEMS04 with $M = 3$ | | | ETTm1 with input 336 | |
|------|-----------|-------------|------|---------------|-----|------|------|-----|-----|
| | | Seasonal | Trend | Multipredictor | MAE | MAPE | RMSE | MSE | MAE |
| ① | ✓ | ↗ | ↙ | ✓ | **18.10** | **11.73** | **28.51** | **0.360** | **0.381** |
| ② | ✓ | ↗ | ↙ | ✗ | 21.49 | 13.12 | 33.48 | 0.375 | 0.398 |
| ③ | ✓ | ✗ | ↙ | ✓ | 23.68 | 16.01 | 37.42 | 0.390 | 0.415 |
| ④ | ✓ | ↗ | ✗ | ✓ | 22.44 | 14.81 | 36.54 | 0.386 | 0.410 |
| ⑤ | ✓ | ↙ | ↙ | ✓ | 20.41 | 13.08 | 31.92 | 0.371 | 0.389 |
| ⑥ | ✓ | ↗ | ↗ | ✓ | 21.28 | 13.19 | 32.84 | 0.370 | 0.388 |
| ⑦ | ✓ | ↙ | ↗ | ✓ | 22.16 | 14.60 | 35.42 | 0.384 | 0.409 |
| ⑧ | ✗ | ↗ | | ✓ | 20.98 | 13.12 | 33.94 | 0.372 | 0.396 |
| ⑨ | ✗ | ↙ | | ✓ | 20.66 | 13.06 | 32.74 | 0.374 | 0.398 |
| ⑩ | ✗ | ✗ | | ✓ | 24.16 | 16.21 | 38.04 | 0.401 | 0.414 |

Table 22: Relative promotion analysis on ablations under different hyperparameter configurations. For example, the relative promotion is calculated by $(1-①/②)$ in case ②.

| Case | Decompose | Past mixing | | Future mixing | PEMS04 with $M = 3$ | | | ETTm1 with input 336 | |
|------|-----------|-------------|------|---------------|-----|------|------|-----|-----|
| | | Seasonal | Trend | Multipredictor | MAE | MAPE | RMSE | MSE | MAE |
| ① | ✓ | ↗ | ↙ | ✓ | - | - | - | - | - |
| ② | ✓ | ↗ | ↙ | ✗ | 15.8% | 10.6% | 14.9% | 4.1% | 4.2% |
| ③ | ✓ | ✗ | ↙ | ✓ | 23.6% | 26.7% | 23.8% | 7.7% | 8.2% |
| ④ | ✓ | ↗ | ✗ | ✓ | 19.2% | 20.1% | 22.0% | 6.8% | 7.1% |
| ⑤ | ✓ | ↙ | ↙ | ✓ | 11.4% | 10.4% | 10.7% | 3.0% | 2.1% |
| ⑥ | ✓ | ↗ | ↗ | ✓ | 15.0% | 11.1% | 13.2% | 2.7% | 1.8% |
| ⑦ | ✓ | ↙ | ↗ | ✓ | 18.4% | 19.7% | 19.5% | 6.2% | 6.9% |
| ⑧ | ✗ | ↗ | | ✓ | 13.7% | 10.6% | 15.9% | 3.2% | 3.8% |
| ⑨ | ✗ | ↙ | | ✓ | 12.4% | 10.2% | 13.0% | 3.7% | 4.2% |
| ⑩ | ✗ | ✗ | | ✓ | 25.1% | 27.6% | 25.0% | 10.2% | 8.0% |

| Case | Decompose | Past mixing | | Future mixing | PEMS04 with $M = 1$ | | | ETTm1 with input 96 | |
|------|-----------|-------------|------|---------------|-----|------|------|-----|-----|
| | | Seasonal | Trend | Multipredictor | MAE | MAPE | RMSE | MSE | MAE |
| ① | ✓ | ↗ | ↙ | ✓ | - | - | - | - | - |
| ② | ✓ | ↗ | ↙ | ✗ | 11.4% | 6.8% | 11.2% | 2.9% | 2.6% |
| ③ | ✓ | ✗ | ↙ | ✓ | 21.6% | 23.1% | 20.2% | 5.1% | 5.4% |
| ④ | ✓ | ↗ | ✗ | ✓ | 16.2% | 16.6% | 16.7% | 3.7% | 2.4% |
| ⑤ | ✓ | ↙ | ↙ | ✓ | 7.6% | 3.7% | 4.7% | 0.5% | 2.1% |
| ⑥ | ✓ | ↗ | ↗ | ✓ | 8.9% | 9.0% | 6.6% | 1.5% | 2.6% |
| ⑦ | ✓ | ↙ | ↗ | ✓ | 13.7% | 17.2% | 10.8% | 5.3% | 5.8% |
| ⑧ | ✗ | ↗ | | ✓ | 10.6% | 6.9% | 11.7% | 1.2% | 1.0% |
| ⑨ | ✗ | ↙ | | ✓ | 11.8% | 10.6% | 12.2% | 0.7% | 0.5% |
| ⑩ | ✗ | ✗ | | ✓ | 22.7% | 24.7% | 21.6% | 3.9% | 2.1% |

## G    ADDITIONAL BASELINES

Due to the limitation of the main text, we also include three advanced baselines here: the general multi-scale framework Scaleformer (Amin Shabani & Sylvain., 2023), two concurrent MLP-based model MTSMixer (Li et al., 2023) and TSMixer (Chen et al., 2023). Since the latter two baselines were not officially published during our submission, we adopted their public code and reproduced them with both unified hyperparameter setting and the hyperparameter searching settings. As presented in Table 23, 24, 25, TimeMixer still performs best in comparison with these baselines. Showcases of these additional baselines are also provided in Appendix I for an intuitive comparison.

## H    SPECTRAL ANALYSIS OF MODEL PREDICTIONS

To demonstrate the advancement of TimeMixer, we plot the spectrum of ground truth and model predictions. It is observed that TimeMixer captures different frequency parts precisely.

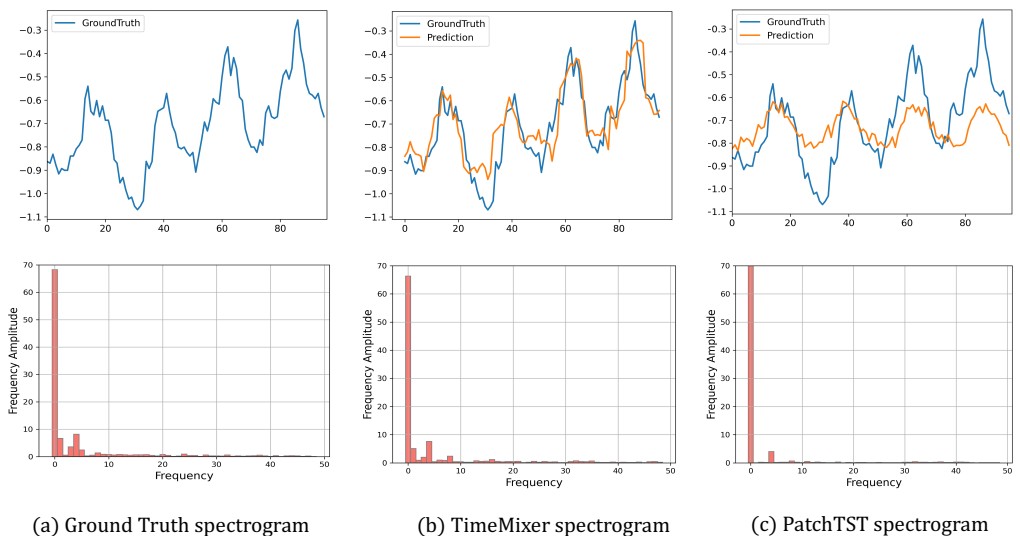

| (a) Ground Truth spectrogram | (b) TimeMixer spectrogram | (c) PatchTST spectrogram |

Figure 8: Prediction spectrogram cases from ETTh1 by ground truth and different models under the input-96-predict-96 settings.

## I    SHOWCASES

In order to evaluate the performance of different models, we conduct the qualitative comparison by plotting the final dimension of forecasting results from the test set of each dataset (Figures 9, 10, 11, 12, 13, 17, 18). Among the various models, TimeMixer exhibits superior performance.

## J    LIMITATIONS AND FUTURE WORK

TimeMixer has shown favorable efficiency in GPU memory and running time as we presented in the main text. However, it should be noted that as the input length increases, the linear mixing layer may result in a larger number of model parameters, which is inefficient for mobile applications. To address this issue and improve TimeMixer's parameter efficiency, we plan to investigate alternative mixing designs, such as attention-based or CNN-based in future research. In addition, we only focus on the temporal dimension mixing in this paper, and also plan to incorporate the variate dimension mixing into model design in our future work. Furthermore, as an application-oriented model, we made a great effort to verify the effectiveness of our design with experiments and ablations. The theoretical analysis to verify the optimality and completeness of our design is also a promising direction.

Table 23: Unified and searched hyperparameter results for additional baselines in the long-term forecasting task. We compare extensive competitive models under different prediction lengths. *Avg* is averaged from all four prediction lengths, that is 96, 192, 336, 720. All these baselines are reproduced by their official code. For the searched hyperparameter setting, we follow the searching strategy described in Appendix E. Especially for TSMixer, we reproduced it in Pytorch (Paszke et al., 2019).

| Models | | **Unified Hyperparameter** | | | | | | | | **Searched Hyperparameter** | | | | | | | |
| --- | --- | --- | --- | --- | --- | --- | --- | --- | --- | --- | --- | --- | --- | --- | --- | --- | --- |
| | | **TimeMixer (Ours)** | | Scaleformer (2023) | | MTSMixer (2023) | | TSMixer (2023) | | **TimeMixer (Ours)** | | Scaleformer (2023) | | MTSMixer (2023) | | TSMixer (2023) | |
| Metric | | MSE | MAE | MSE | MAE | MSE | MAE | MSE | MAE | MSE | MAE | MSE | MAE | MSE | MAE | MSE | MAE |
| **Weather** | 96 | **0.163** | **0.209** | 0.220 | 0.289 | 0.173 | 0.224 | 0.175 | 0.247 | **0.147** | **0.197** | 0.192 | 0.241 | 0.167 | 0.221 | 0.149 | 0.198 |
| | 192 | **0.208** | **0.250** | 0.341 | 0.385 | 0.219 | 0.261 | 0.224 | 0.294 | **0.189** | **0.239** | 0.220 | 0.288 | 0.208 | 0.250 | 0.201 | 0.251 |
| | 336 | **0.251** | **0.287** | 0.463 | 0.455 | 0.274 | 0.300 | 0.262 | 0.326 | **0.241** | **0.280** | 0.288 | 0.324 | 0.298 | 0.302 | 0.287 | 0.291 |
| | 720 | **0.339** | **0.341** | 0.640 | 0.565 | 0.365 | 0.359 | 0.349 | 0.348 | **0.310** | **0.330** | 0.321 | 0.360 | 0.344 | 0.339 | 0.320 | 0.336 |
| | Avg | **0.240** | **0.271** | 0.416 | 0.423 | 0.258 | 0.286 | 0.253 | 0.304 | **0.222** | **0.262** | 0.248 | 0.304 | 0.254 | 0.278 | 0.240 | 0.269 |
| **Solar-Energy** | 96 | **0.189** | **0.259** | 0.271 | 0.331 | 0.217 | 0.272 | 0.216 | 0.294 | **0.167** | **0.220** | 0.224 | 0.328 | 0.199 | 0.251 | 0.190 | 0.272 |
| | 192 | **0.222** | **0.283** | 0.288 | 0.332 | 0.258 | 0.299 | 0.294 | 0.359 | **0.187** | **0.249** | 0.247 | 0.312 | 0.221 | 0.275 | 0.207 | 0.278 |
| | 336 | **0.231** | **0.292** | 0.358 | 0.412 | 0.278 | 0.310 | 0.302 | 0.367 | **0.200** | **0.258** | 0.274 | 0.308 | 0.231 | 0.281 | 0.246 | 0.304 |
| | 720 | **0.223** | **0.285** | 0.377 | 0.437 | 0.293 | 0.321 | 0.311 | 0.372 | **0.215** | **0.250** | 0.335 | 0.384 | 0.270 | 0.301 | 0.274 | 0.308 |
| | Avg | **0.216** | **0.280** | 0.323 | 0.378 | 0.261 | 0.300 | 0.280 | 0.348 | **0.192** | **0.244** | 0.270 | 0.333 | 0.230 | 0.277 | 0.229 | 0.283 |
| **Electricity** | 96 | **0.153** | **0.247** | 0.182 | 0.297 | 0.173 | 0.270 | 0.190 | 0.299 | **0.129** | **0.224** | 0.162 | 0.274 | 0.154 | 0.267 | 0.142 | 0.234 |
| | 192 | **0.166** | **0.256** | 0.188 | 0.300 | 0.186 | 0.280 | 0.216 | 0.323 | **0.140** | **0.220** | 0.171 | 0.284 | 0.168 | 0.272 | 0.154 | 0.248 |
| | 336 | **0.185** | **0.277** | 0.210 | 0.324 | 0.204 | 0.297 | 0.226 | 0.334 | **0.161** | **0.255** | 0.192 | 0.304 | 0.182 | 0.281 | **0.161** | 0.262 |
| | 720 | **0.225** | **0.310** | 0.232 | 0.339 | 0.241 | 0.326 | 0.250 | 0.353 | **0.194** | **0.287** | 0.238 | 0.332 | 0.212 | 0.321 | 0.209 | 0.304 |
| | Avg | **0.182** | **0.272** | 0.203 | 0.315 | 0.201 | 0.293 | 0.220 | 0.327 | **0.156** | **0.246** | 0.191 | 0.298 | 0.179 | 0.286 | 0.167 | 0.262 |
| **Traffic** | 96 | **0.462** | **0.285** | 0.564 | 0.351 | 0.523 | 0.357 | 0.499 | 0.344 | **0.360** | **0.249** | 0.409 | 0.281 | 0.514 | 0.338 | 0.398 | 0.272 |
| | 192 | **0.473** | **0.296** | 0.570 | 0.349 | 0.535 | 0.367 | 0.540 | 0.370 | **0.375** | **0.250** | 0.418 | 0.294 | 0.519 | 0.351 | 0.402 | 0.281 |
| | 336 | **0.498** | **0.296** | 0.576 | 0.349 | 0.566 | 0.379 | 0.557 | 0.378 | **0.385** | **0.270** | 0.427 | 0.294 | 0.557 | 0.361 | 0.412 | 0.294 |
| | 720 | **0.506** | **0.313** | 0.602 | 0.360 | 0.608 | 0.397 | 0.586 | 0.397 | **0.430** | **0.281** | 0.518 | 0.356 | 0.569 | 0.362 | 0.448 | 0.311 |
| | Avg | **0.484** | **0.297** | 0.578 | 0.352 | 0.558 | 0.375 | 0.546 | 0.372 | **0.387** | **0.262** | 0.443 | 0.307 | 0.539 | 0.354 | 0.415 | 0.290 |
| **ETTh1** | 96 | **0.375** | **0.400** | 0.401 | 0.428 | 0.418 | 0.437 | 0.387 | 0.411 | **0.361** | **0.390** | 0.381 | 0.412 | 0.397 | 0.428 | 0.370 | 0.402 |
| | 192 | **0.429** | **0.421** | 0.471 | 0.478 | 0.463 | 0.460 | 0.441 | 0.437 | 0.409 | **0.414** | 0.445 | 0.441 | 0.452 | 0.466 | **0.406** | **0.414** |
| | 336 | **0.484** | **0.458** | 0.527 | 0.498 | 0.516 | 0.478 | 0.507 | 0.467 | 0.430 | **0.429** | 0.501 | 0.484 | 0.487 | 0.462 | **0.424** | 0.434 |
| | 720 | **0.498** | **0.482** | 0.578 | 0.547 | 0.532 | 0.549 | 0.527 | 0.548 | **0.445** | **0.460** | 0.544 | 0.528 | 0.510 | 0.506 | 0.471 | 0.479 |
| | Avg | **0.447** | **0.440** | 0.495 | 0.488 | 0.482 | 0.481 | 0.466 | 0.467 | **0.411** | **0.423** | 0.468 | 0.466 | 0.461 | 0.464 | 0.418 | 0.432 |
| **ETTh2** | 96 | **0.289** | **0.341** | 0.368 | 0.398 | 0.343 | 0.378 | 0.308 | 0.357 | **0.271** | **0.330** | 0.340 | 0.394 | 0.328 | 0.367 | 0.271 | 0.339 |
| | 192 | **0.372** | **0.392** | 0.431 | 0.446 | 0.422 | 0.425 | 0.395 | 0.404 | **0.317** | **0.402** | 0.401 | 0.414 | 0.404 | 0.426 | 0.344 | 0.397 |
| | 336 | **0.386** | **0.414** | 0.486 | 0.474 | 0.462 | 0.460 | 0.428 | 0.434 | **0.332** | **0.396** | 0.437 | 0.448 | 0.406 | 0.434 | 0.360 | 0.400 |
| | 720 | **0.412** | **0.434** | 0.517 | 0.522 | 0.476 | 0.475 | 0.443 | 0.451 | **0.342** | **0.408** | 0.469 | 0.471 | 0.448 | 0.463 | 0.428 | 0.461 |
| | Avg | **0.364** | **0.395** | 0.451 | 0.460 | 0.426 | 0.435 | 0.394 | 0.412 | **0.316** | **0.384** | 0.412 | 0.432 | 0.397 | 0.422 | 0.350 | 0.399 |
| **ETTm1** | 96 | **0.320** | **0.357** | 0.383 | 0.408 | 0.344 | 0.378 | 0.331 | 0.378 | 0.291 | 0.340 | 0.338 | 0.375 | 0.316 | 0.362 | **0.288** | **0.336** |
| | 192 | **0.361** | **0.381** | 0.417 | 0.421 | 0.397 | 0.408 | 0.386 | 0.399 | **0.327** | **0.365** | 0.392 | 0.406 | 0.374 | 0.391 | 0.332 | 0.374 |
| | 336 | **0.390** | **0.404** | 0.437 | 0.448 | 0.429 | 0.430 | 0.426 | 0.421 | 0.360 | **0.381** | 0.410 | 0.426 | 0.408 | 0.411 | **0.358** | 0.381 |
| | 720 | **0.454** | **0.441** | 0.512 | 0.481 | 0.489 | 0.460 | 0.489 | 0.465 | **0.415** | **0.417** | 0.481 | 0.476 | 0.472 | 0.454 | 0.420 | **0.417** |
| | Avg | **0.381** | **0.395** | 0.438 | 0.440 | 0.415 | 0.419 | 0.408 | 0.416 | **0.348** | **0.375** | 0.406 | 0.421 | 0.393 | 0.405 | 0.350 | 0.377 |
| **ETTm2** | 96 | **0.175** | **0.258** | 0.201 | 0.297 | 0.191 | 0.278 | 0.179 | 0.282 | 0.164 | 0.254 | 0.192 | 0.274 | 0.187 | 0.268 | **0.160** | **0.249** |
| | 192 | **0.237** | **0.299** | 0.261 | 0.324 | 0.258 | 0.320 | 0.244 | 0.305 | **0.223** | **0.295** | 0.248 | 0.322 | 0.237 | 0.301 | 0.228 | 0.299 |
| | 336 | **0.298** | **0.340** | 0.328 | 0.366 | 0.319 | 0.357 | 0.320 | 0.357 | 0.279 | 0.330 | 0.301 | 0.348 | 0.299 | 0.352 | **0.269** | **0.328** |
| | 720 | **0.391** | **0.396** | 0.424 | 0.417 | 0.417 | 0.412 | 0.419 | 0.432 | **0.359** | **0.383** | 0.411 | 0.398 | 0.413 | 0.419 | 0.421 | 0.426 |
| | Avg | **0.275** | **0.323** | 0.303 | 0.351 | 0.296 | 0.342 | 0.290 | 0.344 | **0.256** | **0.315** | 0.288 | 0.336 | 0.284 | 0.335 | 0.270 | 0.326 |

Table 24: Short-term forecasting results in the PEMS datasets with multiple variates. All input lengths are 96 and prediction lengths are 12. A lower MAE, MAPE or RMSE indicates a better prediction.

| Models | | TimeMixer (Ours) | Scaleformer (2023) | MTSMixer (2023) | TSMixer (2023) |
|---|---|---|---|---|---|
| PEMS03 | MAE | **14.63** | 17.66 | 18.63 | 15.71 |
| | MAPE | **14.54** | 17.58 | 19.35 | 15.28 |
| | RMSE | **23.28** | 27.51 | 28.85 | 25.88 |
| PEMS04 | MAE | **19.21** | 22.68 | 25.57 | 20.86 |
| | MAPE | **12.53** | 14.81 | 17.79 | 12.97 |
| | RMSE | **30.92** | 35.61 | 39.92 | 32.68 |
| PEMS07 | MAE | **20.57** | 27.62 | 25.69 | 22.97 |
| | MAPE | **8.62** | 12.68 | 11.57 | 9.93 |
| | RMSE | **33.59** | 42.27 | 39.82 | 35.68 |
| PEMS08 | MAE | **15.22** | 20.74 | 24.22 | 18.79 |
| | MAPE | **9.67** | 12.81 | 14.98 | 10.69 |
| | RMSE | **24.26** | 32.77 | 37.21 | 26.74 |

Table 25: Short-term forecasting results in the M4 dataset with a single variate. All prediction lengths are in $[6, 48]$. A lower SMAPE, MASE or OWA indicates a better prediction.

| Models | | TimeMixer (Ours) | Scaleformer (2023) | MTSMixer (2023) | TSMixer (2023) |
|---|---|---|---|---|---|
| Yearly | SMAPE | **13.206** | 13.778 | 20.071 | 19.845 |
| | MASE | **2.916** | 3.176 | 4.537 | 4.439 |
| | OWA | **0.776** | 0.871 | 1.185 | 1.166 |
| Quarterly | SMAPE | **9.996** | 10.727 | 16.371 | 16.322 |
| | MASE | **1.166** | 1.291 | 2.216 | 2.21 |
| | OWA | **0.825** | 0.954 | 1.551 | 1.543 |
| Monthly | SMAPE | **12.605** | 13.378 | 18.947 | 19.248 |
| | MASE | **0.919** | 1.104 | 1.725 | 1.774 |
| | OWA | **0.869** | 0.972 | 1.468 | 1.501 |
| Others | SMAPE | **4.564** | 4.972 | 7.493 | 7.494 |
| | MASE | **3.115** | 3.311 | 5.457 | 5.463 |
| | OWA | **0.982** | 1.112 | 1.649 | 1.651 |
| Weighted Average | SMAPE | **11.723** | 12.978 | 18.041 | 18.095 |
| | MASE | **1.559** | 1.764 | 2.677 | 2.674 |
| | OWA | **0.840** | 0.921 | 1.364 | 1.336 |

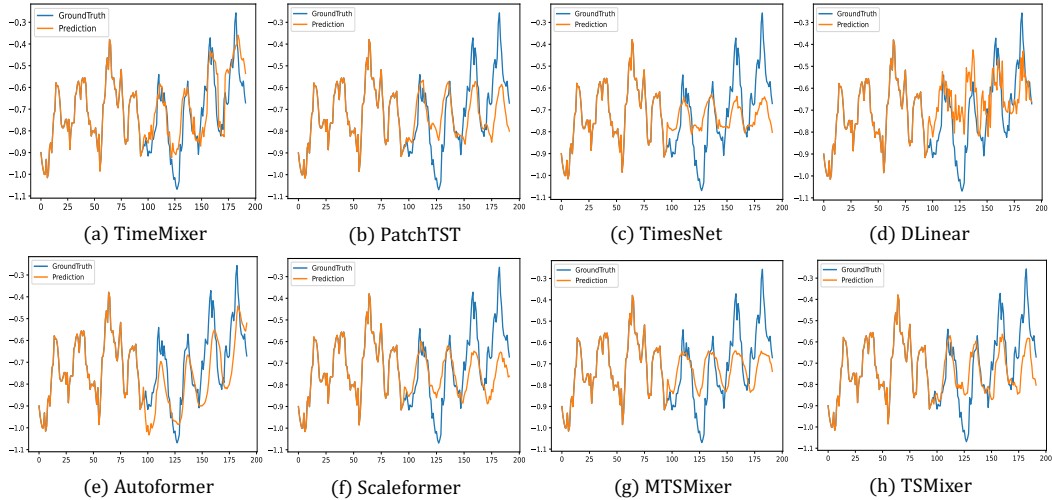

Figure 9: Prediction cases from ETTh1 by different models under the input-96-predict-96 settings. Blue lines are the ground truths and orange lines are the model predictions.

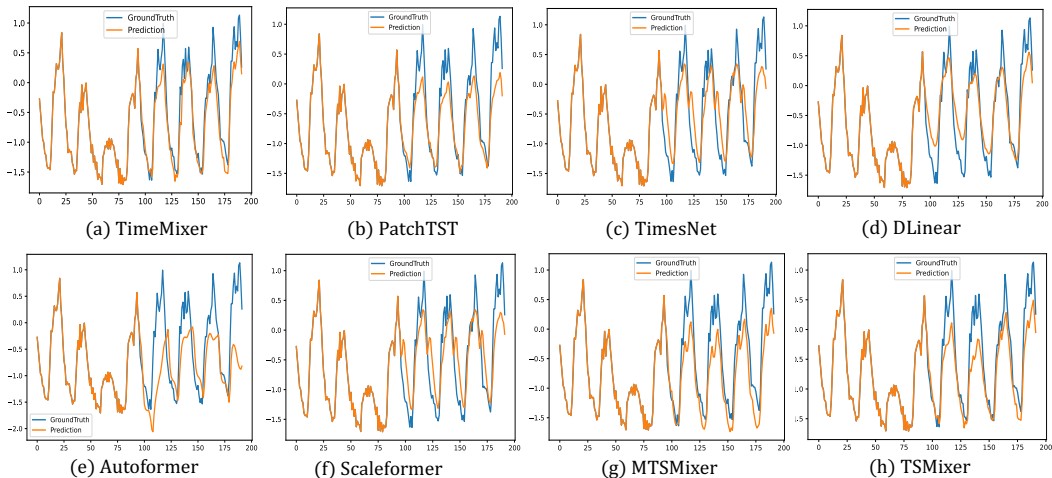

Figure 10: Prediction cases from Electricity by different models under input-96-predict-96 settings.

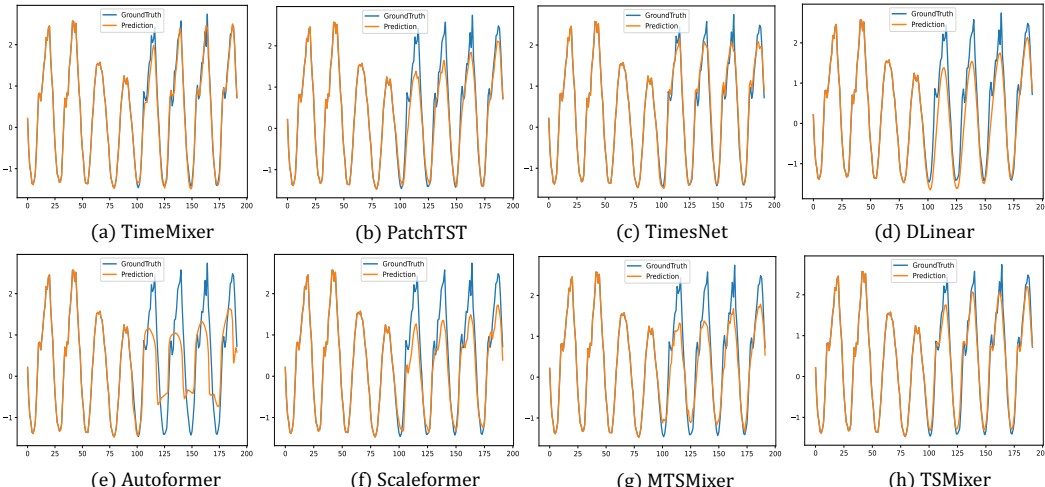

Figure 11: Prediction cases from Traffic by different models under the input-96-predict-96 settings.

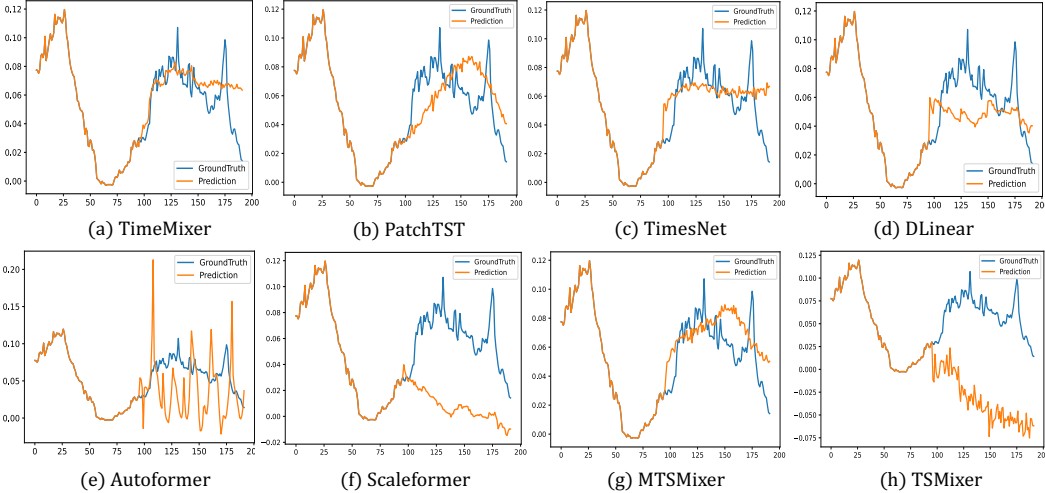

Figure 12: Prediction cases from Weather by different models under the input-96-predict-96 settings.

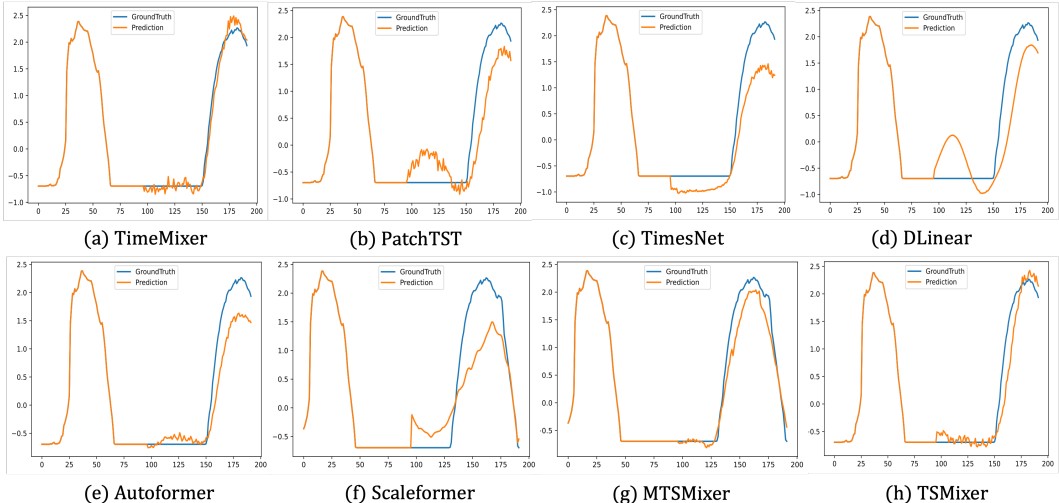

Figure 13: Showcases from Solar-Energy by different models under the input-96-predict-96 settings.

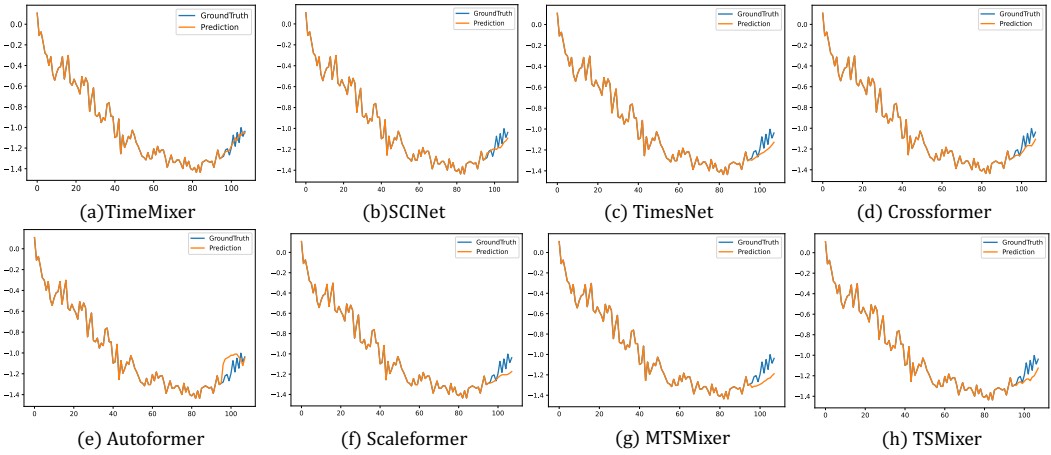

Figure 14: Showcases from PEMS03 by different models under the input-96-predict-12 settings.

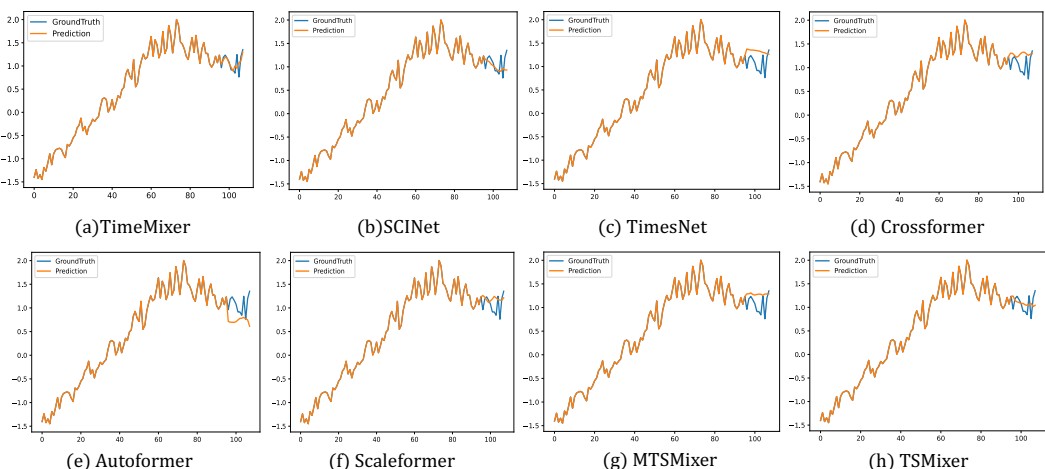

Figure 15: Showcases from PEMS04 by different models under the input-96-predict-12 settings.

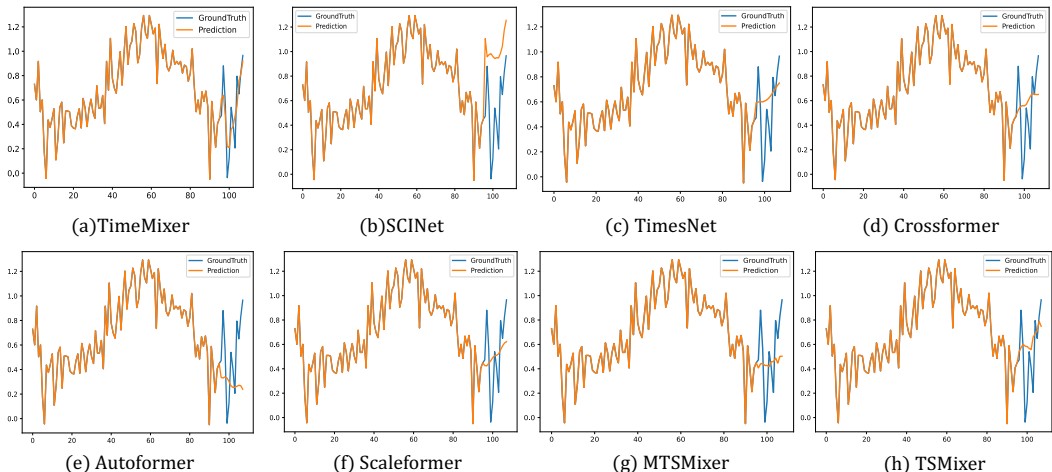

Figure 16: Showcases from PEMS07 by different models under the input-96-predict-12 settings.

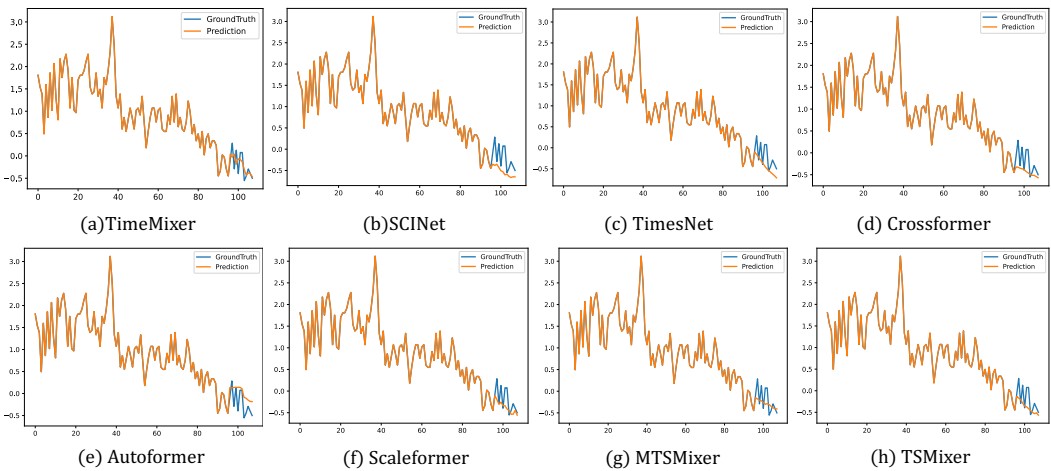

Figure 17: Showcases from PEMS08 by different models under the input-96-predict-12 settings.

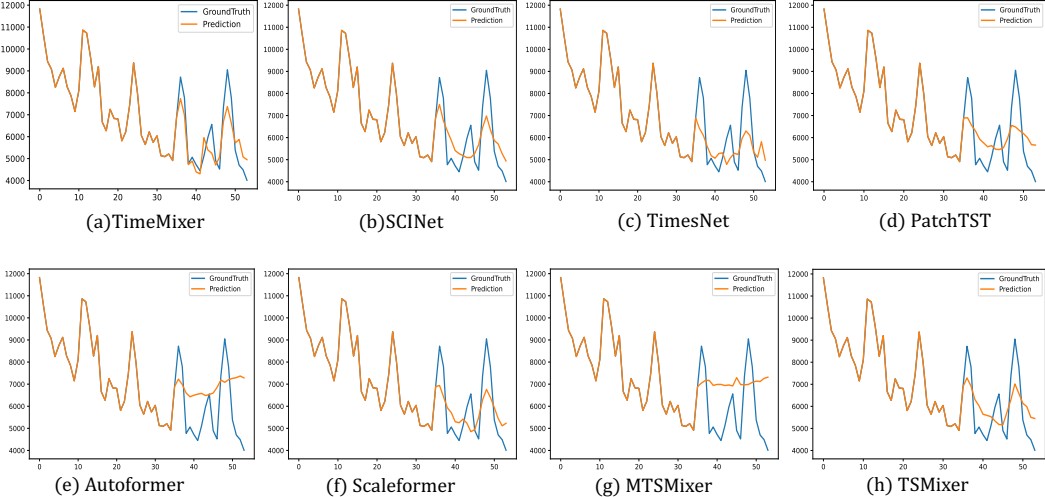

Figure 18: Showcases from the M4 dataset by different models under the input-36-predict-18 settings.

