# OpenReview forum: "TimeMixer: Decomposable Multiscale Mixing for Time Series Forecasting"
_ICLR.cc/2024/Conference — ICLR 2024 poster_

### Official Review · Reviewer_ixvB · 2023-10-26

**Soundness:** 3 good
**Presentation:** 4 excellent
**Contribution:** 3 good
**Rating:** 3
**Confidence:** 3

**Summary:**

To better forecast complicated time series that include distinct patterns in each scale, this paper proposes TimeMixer which is a fully MLP-based architecture. This model incorporates Past-Decomposable-Mixing(PDM) and Future-Multipredictor-Mixing (FMM). As for PDM, it first decomposes time series into seasonal and trend parts. Subsequently, PDM processes seasonal and trend in a fine-to-coarse and coarse-to-fine manner. After PDM processes input time series, FDM forecasts future observations with multiple predictors. TimeMixer equipped with PDM and FMM shows the best results in various datasets for short-term and long-term forecasting. Furthermore, it proves the efficacy of the decomposition, different-way processing, and multiple predictors.

**Strengths:**

**1. Easy to understand**
In simpler terms, the paper is written well. It explains its ideas clearly and in a way that anyone can understand. This makes it easier for readers to grasp the concepts and improves the paper's overall clarity.

**2. Good performance with simple models**
Empirically, TimeMixer exhibits remarkably low computational and memory costs because it consists of only simple linear models. However, it's important not to overlook its performance, which should not be underestimated.

**3. Good observations**
From the thought that fine-scale seasonality can provide coarse-scale one with detailed information while coarse-scale trend can provide fine-scale one with more denoised information, TimeMixer processes trend and seasonality in a different way. The experiment section proves the efficacy of this design and thought. I think that this can broadly affect how to utilize multiple scales in time series.

**4. Enough experiments**
In the experimental section, the paper presents numerous forecasting results, including outcomes for both long-term and short-term forecasting tasks.

**Weaknesses:**

In spite of the good aspects of this paper, I hesitate to give acceptance because of some minor but important concerns.

**1. Absence of some important baselines**
Because this paper is based on decomposition, I think the authors have to include methods based on decompositions [1] into baselines. Furthermore, (although they are not accepted to any conference, ) it is beneficial to include methods based on MLP-Mixer [2,3]. I think that this absence makes the second and fourth strengths fade.

**2. Insufficient experiment in ablation studies**
I think the biggest contribution of this paper, or the main reason for acceptance, is different-way processing (coarse-to-fine and fine-to-coarse). Although the thought leading to this asymmetric design is not quite intuitive, Table 5 provides empirical results. However, the used datasets are insufficient. Considering its importance, more datasets should be used so this design has to be tested more rigorously. Furthermore, because small $M=1$ is employed for short-term forecasting such as M4, M4 cannot prove the efficacy of the asymmetric design sufficiently. Finally, there isn't an ablation setting where seasonal is coarse-to-fine and the trend is fine-to-coarse. These insufficient ablation studies make the third strength vanish.

If these main concerns are resolved (include more related baselines and design more strict ablation studies to prove the efficacy of asymmetric design), I will consider raising my points.


[1] Shabani et al., Scaleformer: Iterative Multi-scale Refining Transformers for Time Series Forecasting, 2023, ICLR
[2] Li et al., MTS-Mixers: Multivariate Time Series Forecasting via Factorized Temporal and Channel Mixing, 2023
[3] Chen et al., TSMixer: An All-MLP Architecture for Time Series Forecasting, 2023

**Questions:**

1. Most datasets you used are multivariate datasets but I am curious about why didn't you design MLPs to capture inter-feature connections.

2. I think the equation (6) is not aligned with Figure 4. An average of Figure (b,c,d,e) ($\hat{x}=\frac{1}{M+1} \sum_{m=0}^{M}\hat{x}_m $) can produce figure 4. However, just summing Figure (b,c,d,e) doesn't result in Figure (a). When I checked your code, not average but the sum was implemented. Can you explain this situation?

3. In table 4, I am confused about the meaning of x mark in pasting mixing and future mixing columns. What does it mean in each column in detail? In other words, how did you implement the method with x mark in each column?

---

> ### Author Response · Authors · 2023-11-15
> **Response to Reviewer ixvB [Part 1]**
>
> Many thanks to Reviewer ixvB for providing the insightful review and comments.
>
> > **Q1:** Add Scaleformer, MTS-Mixers, TSMixer in comparison.
>
> In the original submission, we compared TimeMixer with 15 baselines. Following the reviewer's request, we have included Scaleformer, MTS-Mixers, and TSMixer into comparison.
>
> As we stated in the $\underline{\text{Section 4 of main text}}$, we experiment with these baselines with their official code but align the hyperparameters to ensure a fair comparison. All the results have been included in the $\underline{\text{Appendix H of revised paper}}$. Here are some key experiments.
>
> In addition to the quantitive results, showcase comparisons have also been added in $\underline{\text{Appendix J of revised paper}}$ for an intuitive comparison.
>
> | Averaged from 4 prediction lengths (MSE \| MAE) | Scaleformer    | MTS-Mixer      | TSMixer        | TimeMixer              |
> | ----------------------------------------------- | -------------- | -------------- | -------------- | ---------------------- |
> | Weather                                         | 0.416 \| 0.423 | 0.258 \| 0.286 | 0.253 \| 0.304 | **0.240** \| **0.271** |
> | Solar-Energy                                    | 0.323 \| 0.378 | 0.261 \| 0.300 | 0.280 \| 0.348 | **0.216** \| **0.280** |
> | Electricity                                     | 0.203 \| 0.315 | 0.201 \| 0.293 | 0.220 \| 0.327 | **0.182** \| **0.272** |
> | Traffic                                         | 0.578 \| 0.352 | 0.558 \| 0.375 | 0.546 \| 0.372 | **0.484** \| **0.297** |
> | ETTh1                                           | 0.495 \| 0.488 | 0.482 \| 0.481 | 0.466 \| 0.467 | **0.447** \| **0.440** |
> | ETTh2                                           | 0.451 \| 0.460 | 0.426 \| 0.435 | 0.394 \| 0.412 | **0.364** \| **0.395** |
> | ETTm1                                           | 0.438 \| 0.440 | 0.415 \| 0.419 | 0.408 \| 0.416 | **0.381** \| **0.395** |
> | ETTh2                                           | 0.303 \| 0.351 | 0.296 \| 0.342 | 0.290 \| 0.344 | **0.275** \| **0.323** |
>
> > **Q2:** Complete the ablation studies to prove the efficacy of asymmetric design.
>
> In the original submission, we conducted experiments on M4 and ETTh1, which are two representative datasets in long- and short-term forecasting. Thanks for the reviewer's suggestion, we have completed the ablations in all the benchmarks. Besides, we also add a new ablation setting, where we adopt the bottom-up mixing for trend and top-down mixing for the seasonal part.
>
> **Overall, we have conducted 10 types of ablations in all 18 benchmarks.** Here are part of the main results. The complete results have been included in $\underline{\text{Table 5 and Table 15,16,17 of revised paper}}$. In all benchmarks, our official design performs best consistently, which can verify the efficacy of our asymmetric design.
>
> It is also notable that completely reversing mixing directions for seasonal and trend parts leads to a serious performance drop. This may come from that the essential microscopic information in finer-scale seasons and macroscopic information in coarser-scale trends are ruined by unsuitable mixing approaches.
>
> | Ablations                                                    | ETTm1 (Predict-336) MSE \|MAE | M4 SMAPE \|MASE \|OWA             | PEMS04 MAE \|MAPE \|RMSE        |
> | ------------------------------------------------------------ | -------------------------------- | ------------------------------------ | ----------------------------------- |
> | Bottom-up mixing for trend and top-down mixing for seasonal part | 0.412 \| 0.429                   | 13.012 \| 1.657 \| 0.954             | 22.27 \| 15.14 \| 34.67             |
> | TimeMixer                                                    | **0.390** \| **0.404**           | **11.723** \| **1.559** \| **0.840** | **19.21** \| **12.53** \| **30.92** |

---

> > ### Author Response · Authors · 2023-11-15
> > **Response to Reviewer ixvB [Part 2]**
> >
> > > **Q3:** "Why didn't you design MLPs to capture inter-feature connections?"
> >
> > Thanks for the reviewer's suggestion. Just as we highlighted in our model name "TimeMixer", we only focus on the time dimension mixing in the paper.
> >
> > As the reviewer mentioned, the variate dimension mixing is definitely an important direction. However, they may also degenerate in the univariate forecasting tasks, such as M4. Thus, considering the scope of our paper, we would like to leave the exploration of the variate dimension mixing as the future work.
> >
> > > **Q4:**  "Equation (6) is not aligned with Figure 4. Can you explain this situation?"
> >
> > As we presented in Equation (6), TimeMixer summarizes the prediction results from multiscale mixed features.
> >
> > In Figure 4, we want to demonstrate that multiscale series hold complementary forecasting capabilities. The reasonable design to examine this hypothesis is to forecast the future solely based on one single scale feature. **Thus, we fix the PDM and retrain a new predictor for the feature at each scale with the ground truth future as supervision, that is why the predictions from multiple scales are all close to the real future.**
> >
> > Sorry for the missing descriptions. We have included the implementation detailed for Figure 4 in the $\underline{\text{Appendix A of revised paper}}$.
> >
> > > **Q5:** "How did you implement the method with 'x' mark in each column?"
> >
> > Sorry for these missing details.
> >
> > (1) For the "x" mark in decomposition, we just obtain a single feature for each scale. Thus, we can experiment with the bottom-up and top-down choices separately.
> >
> > (2) For the "x" mark in past mixing, we just leave the decomposed component with the "x" mark alone and only conduct mixing to the other component.
> >
> > (3) For the "x" mark in future mixing, we only regress the future from the finest scale feature $\mathbf{x}_{0}^L$.
> >
> > All the implementation details have been included in the $\underline{\text{Appendix G.1 of revised paper}}$.

---

> ### Author Response · Authors · 2023-11-18
> **Request of Reviewer's attention and feedback**
>
> Dear Reviewer,
>
> We kindly remind you that it has been 3 days since we posted our rebuttal. Please let us know if our response has addressed your concerns.
>
> Following your suggestion, we have answered your concerns and improved the paper in the following aspects:
>
> - We have **added 3 new baselines (Scaleformer, MTSMixer, TSMixer) in all 18 benchmarks** to demonstrate the advancement of TimeMixer. Both **quantitive results and showcases** are provided in the revised paper.
> - We can **complete 10 types of ablations in all 18 benchmarks**, which verifies the effectiveness of our design in TimeMixer. The implementation details of each ablation have also been included.
> - We have also provided the implementation details for the visualization experiments.
>
> In total, we have added more than 100 new experiments. **All of these results have been included in the $\underline{\text{revised paper}}$.**
>
> Thanks again for your valuable review. We are looking forward to your reply and are happy to answer any future questions.

---

> > ### Comment · Reviewer_ixvB · 2023-11-20
> > **Response to Author**
> >
> > Thank you for your effort to resolve my concerns, including abundant experiments.
> >
> > However, there are still some nonsensical parts to me. The followings are some concerns that I still maintain.
> >
> > **1. Concerns about your empirical evidence for the asymmetric design.**
> > As I said before, I think the only novel part of your method is the asymmetric design. However, your intuitions ("essential microscopic information in finer-scale seasons and macroscopic information in coarser-scale trends") for this design aren't very convincing. For example, microscopic information in finer-scale seasons can still contain a lot of noise, leading to useless information. Therefore, very rigorous tests are required. However, despite additional experiments, there are still non-sensical parts in your evaluation settings of ablation studies.
> >
> > - (1) Because small M is employed for short-term forecasting such as M4, M4 cannot prove the efficacy of the asymmetric design sufficiently.
> >  &rarr; Ablation studies (Table 5) must include short-term forecasting results with large M.
> > - (2) In long-term forecasting experiments, a relatively small input length, 96, is used. I think the effects of decomposition and making information coarse and fine might be significant in long input length. I think that with a small input length, it is hard to recognize the effect of your design. For this reason, the efficacy of your method against TSMixer, MTSMixer, and Scaleformer has to be proven in long input length.
> > &rarr; Ablation studies (Table 5) in long-term forecasting have to include the results with a large search space of the input length, such as in Table 14. Also, TSMixer, MTSMixer, and Scaleformer have to be compared to TimeMixer in Table 14.
> >
> > **2. Concerns about evaluations in Figure 4**
> > I understand your evaluation settings. However, I think this setting makes some changes in the original design and the changes are not trivial. With this significant change, I think the exact behavior of your method cannot be caught. Maintaining your original design, this figure becomes meaningful.
> >
> > **3. Additional question**
> > After identifying the exact design of your method, I am curious about the reason why the final future multi-perdictor mixing is designed with not averaging but summing. Considering the scales of input in each predictor, averaging is more natural. Are there any intuition, theoretical reasons, or empirical results for this design?

---

> > > ### Author Response · Authors · 2023-11-20
> > > **The Second Response to Reviewer ixvB [Part 1]**
> > >
> > > Many thanks for your prompt response and further clarifying your concerns, which is quite instructive for us to answer your questions in every detail.
> > >
> > > > **Q1: Concerns about your empirical evidence for the asymmetric design.**
> > >
> > > We are happy to add new experiments to address the reviewer's concern, including:
> > >
> > > （1）Ablation studies for short-term forecasting results with large M.
> > >
> > > （2）Ablation studies for long-term forecasting with large input length.
> > >
> > > （3）Compare with TSMixer, MTSMixer, and Scaleformer under searched hyperparameter.
> > >
> > > **But this requires more than 80 new experiment results. We just are working on it, which needs about 1 day for running. After we finish these experiments, we will respond to you as soon as possible.**
> > >
> > > In addition, we would like to highlight some points, that might be helpful to you in understanding our experiment settings:
> > >
> > > - M4 is a standard dataset, whose input length of each subset is fixed to a small number. Increasing M may result in a meaningless setting. **Thus, we will provide the ablations for the PEMS dataset with large M, which is also a short-term forecasting benchmark.**
> > > - In this paper, we have provided two types of experiments: unifying input length as 96 and searching input length. **Actually, all the previous papers only evaluate their models under one of the above two settings, such as TimesNet (ICLR 2023), and PatchTST (ICLR 2023).** Especially, Scaleformer and MTSMixer that you mentioned also only provide the input-96 results.
> > > - To make the paper structure clear, we place the unified input-96 setting in the main text and all the results with the searched hyperparameter in the appendix.  We have done our best to ensure a fair comparison and clear presentation. **But it seems like our effort also brings many additional concerns that we have to do double ablation and comparison, making this work with a double workload. To ensure the completeness of the main text, we plan to include the hyperparameter searching results only in the Appendix. Hope you can understand this arrangement.**
> > > - Given many previous papers only provide the input-96 setting experiments (e.g. TimesNet, Scaleformer, MTSMixer), **we believe that our previous rebuttal under the input-96 setting (more than 100 new results) is meaningful and convincing, including both quantitive results and showcases.** Note that these ablations are all under an aligned hyperparameter setting to ensure rigor.
> > >
> > > We also understand the point that you mentioned. Note that it is hard (perhaps not necessary) to try all the ablations under every hyperparameter combination during the limited rebuttal period, while we will try our best to provide full results under searched hyperparameters to ensure scientific rigor.

---

> > > ### Author Response · Authors · 2023-11-21
> > > **The Second Response to Reviewer ixvB [Part 3]**
> > >
> > > As per the reviewer's request, we have added **116 new experiments** to finish ablations of TimeMixer under new configurations and comparisons with TSMixer, MTSMixer, and Scaleformer under hyperparameter searching. **All the results and analysis have been included in $\underline{\text{Table 21, 22, 23, Appendix G.5, H (Page 20-23) of revised paper highlighted in purple}}$.** Here are part of the results and analysis.
> > >
> > > > **Q1-Part 1: "Ablation studies (Table 5) must include short-term forecasting results with large M."**
> > >
> > > In Table 5, we provide the short-term forecasting results of M4 and PEMS04. As we stated previously, M4 is a standard dataset, whose input length is fixed to a small value. Increasing M may result in a meaningless setting. Thus, we newly provide the ablations for the PEMS04 dataset with a large M. It is observed that under the larger M setting, our designs in TimeMixer are still valuable to the final performance.
> > >
> > > We also provide the relative promotion to quantify the performance gain of each design in TimeMixer, where we can observe the following items:
> > >
> > > - A proper mixing direction is essential to the final performance (case 5,6,7).
> > > - Under the M=3 setting, the relative promotion is generally more significant than the smaller M configuration, further demonstrating the effectiveness of our design in mixing direction under multiscale modeling framework. (*Many thanks to the reivewer's suggestion.*)
> > >
> > > The complete results and relative promotion analysis for **all 10 cases** can be found in the $\underline{\text{revised paper}}$.
> > >
> > > | Metric (MAE\|MAPE RMSE)              | PEMS04 with M=1 (Table 5 of main text) | Relative Promotion for M=1 in MAE | PEMS04 with M=3 (Table 21 of appendix) | Relative Promotion for M=3 in MAE |
> > > | ------------------------------------ | -------------------------------------- | --------------------------------- | -------------------------------------- | --------------------------------- |
> > > | 1 (Official design of TimeMixer)     | **19.21 \|12.53 \|30.92**              | -                                 | **18.10 \|11.73 \|28.51**              | -                                 |
> > > | 2 (Remove FMM)                       | 21.67 \|13.45 \|34.89                  | 11.4%                             | 21.49 \|13.12 \|33.48                  | 15.8%                             |
> > > | 3 (Remove seasonal mixing)           | 24.49 \|16.28 \|38.79                  | 21.6%                             | 23.68 \|16.01 \|37.42                  | 23.6%                             |
> > > | 4 (Remove trend mixing)              | 22.91 \|15.02 \|37.04                  | 16.2%                             | 22.44 \|14.81 \|36.54                  | 19.2%                             |
> > > | 5 (Change seasonal mixing direction) | 20.78 \|13.02 \|32.47                  | 7.6%                              | 20.41 \|13.08 \|31.92                  | 11.4%                             |
> > > | 6 (Change trend mixing direction)    | 21.09 \|13.78 \|33.11                  | 8.9%                              | 21.28 \|13.19 \|32.84                  | 15.0%                             |
> > > | 7 (Change all the mixing direction)  | 22.27 \|15.14 \|34.67                  | 13.7%                             | 22.16 \|14.60 \|35.42                  | 18.4%                             |

---

> > > ### Author Response · Authors · 2023-11-22
> > > **[We are anticipating your feedback] & [Summary of second-round revisions]**
> > >
> > > Dear Reviewer,
> > >
> > > Many thanks for your valuable suggestions, instructive responses, and detailed descriptions of your concerns, which have inspired us to improve our paper substantially.
> > >
> > > In the newly submitted revision, we have provided extensive experiments faithfully following your suggestions:
> > >
> > > - Complete ablations of Table 5 for long- and short-term forecasting under both unified and searched hyperparameters.
> > > - Comprehensive comparison with Scaleformer, MTSMixer and TSMixer under both unified and searched hyperparameters.
> > > - Add new results for Figure 4 and discuss the ensemble strategies of FMM.
> > >
> > > We hope that our efforts in the rebuttal period (**more than 200 new results, 9 new pages**) have addressed your concerns to your satisfaction. We eagerly await your reply and are happy to answer any further questions. **We kindly remind you that the reviewer-author discussion phase will end in 24 hours. After that, we may not have a chance to respond to your comments**.
> > >
> > > Sincere thanks for your dedication! We are looking forward to your reply.

---

> ### Author Response · Authors · 2023-11-20
> **The Second Response to Reviewer ixvB [Part 2]**
>
> > **Q2: Concerns about evaluations in Figure 4.**
>
> The original motivation of Figure 4 is to **"provide an intuitive understanding of the forecasting skills of multiscale series", which means that our experiment subject is the learned multiscale features, not predictors.** Thus, we think our previous fix-PDM-retrain-FMM design can verify this point.
>
> But as per your request, **we have also provided the case of directly visualizing the results of each predictor in $\underline{\text{Appendix A (Figure 7) of revised paper}}$ as a supplement, which is also intuitive in verifying the different forecasting capabilities of series in different scales.**
>
> But since we use the sum ensemble, the scale of each predictor output is around $\frac{1}{M+1}$ of real series. More discussions are included in the **Q3**.
>
> > **Q3: Are there any intuition, theoretical reasons, or empirical results for this design?**
>
> Firstly, we would like to highlight that in our design, the loss is calculated based on the ensembled results, not each predictor, that is $MSE(\mathbf{x},\hat{\mathbf{x}})$=$MSE(\mathbf{x},\sum_{m=0}^{M}\hat{\mathbf{x}}_{m})$.
>
> Thus, if you change the ensemble strategy as "average", the loss will be $MSE(\mathbf{x},\hat{\mathbf{x}})$=$MSE(\mathbf{x},\frac{1}{M+1}\sum_{m=0}^{M}\hat{\mathbf{x}}_{m})$. Obviously, the difference between "average" and "mean" is only a constant multiple.
>
> **Theoretical intuition:** It is common sense in the deep learning community that deep models can easily fit constant multiple. For example, if we replace the "sum" with "average", under the same supervision, the deep model can easily fit this change by learning the parameters of each predictor equal to the $\frac{1}{M+1}$ of the "sum" case, which means these two designs are equivalent in learning final prediction under the deep model aspect. And we do expect the reviewer can think about this design from the deep learning perspective.
>
> **Empirical results:** Following your suggestion, we also provide the empirical results here. We can find that the performances of these two strategies are very close.
>
> |                     | M4 (SMAPE \| MASE \| OWA) | PEMS04 (MAE \| MAPE \| RMSE) | ETTm1 (MSE \| MAE) |
> | ------------------- | ------------------------- | ---------------------------- | ------------------ |
> | TimeMixer (sum)     | 11.723\|1.559\|0.840      | 19.21\|12.53\|30.92          | 0.390\|0.404       |
> | TimeMixer (average) | 11.742\|1.573\|0.851      | 19.17\|12.45\|30.88          | 0.391\|0.407       |
> | Relative fluctuation on the first metric |  **0.16%**     | **0.20%**          |   **0.25%**     |
>
> These results have also been included in $\underline{\text{G.4 of revised paper}}$.
>
> Thanks again for your detailed and prompt response, which is very important to us. **All the results for Q2 and Q3 are inclued in $\underline{\text{revised paper highlighted in purple}}$. The new results for Q1 will come soon.**

---

> ### Author Response · Authors · 2023-11-21
> **The Second Response to Reviewer ixvB [Part 4]**
>
> > **Q1-Part 2: "Ablation studies (Table 5) in long-term forecasting have to include the results with a large search space of the input length, such as in Table 14."**
>
> In Table 5, we provide the long-term forecasting results of ETTm1 in the input-96-predict-336 setting. Following the reviewer's suggestion, we also conducted the ablations on input-336 (searched input length). Here are the results. We can find that under the input-336 configuration, the relative promotion is more significant than the smaller input setting.
>
> As for the "noise in finer-scale seasons" that the reviewer mentioned, we would like to recall that the **bottom-up mixing layer will downsample the seasonal information**, and the downsample operation can play a denoising role in principle. However since we attempt to propose a practical model, the theoretical proof is left as future work. In fact, it is hard to find strict proof for deep models. We can only provide the intuitive understanding and empirical results (extensive ablations that we have provided).
>
> | Metric (MSE\|MAE)                    | ETTm1 with input-96 (Table 5 of main text) | Relative Promotion for input-96 in MSE | ETTm1 with input-336 (Table 21 of appendix) | Relative Promotion for input-336 in MSE |
> | ------------------------------------ | ------------------------------------------ | -------------------------------------- | ------------------------------------------- | --------------------------------------- |
> | 1 (Official design of TimeMixer)     | **0.390 \| 0.404**                         | -                                      | **0.360 \| 0.381**                          | -                                       |
> | 2 (Remove FMM)                       | 0.402 \| 0.415                             | 2.9%                                   | 0.375 \| 0.398                              | 4.1%                                    |
> | 3 (Remove seasonal mixing)           | 0.411 \| 0.427                             | 5.1%                                   | 0.390 \| 0.415                              | 7.7%                                    |
> | 4 (Remove trend mixing)              | 0.405 \| 0.414                             | 3.7%                                   | 0.386 \| 0.410                              | 6.8%                                    |
> | 5 (Change seasonal mixing direction) | 0.392 \| 0.413                             | 0.5%                                   | 0.371 \| 0.389                              | 3.0%                                    |
> | 6 (Change trend mixing direction)    | 0.396 \| 0.415                             | 1.5%                                   | 0.370 \| 0.388                              | 2.7%                                    |
> | 7 (Change all the mixing direction)  | 0.412 \| 0.429                             | 5.3%                                   | 0.384 \| 0.409                              | 6.2%                                    |
>
> > **Q1-Part 3: "Also, TSMixer, MTSMixer, and Scaleformer have to be compared to TimeMixer in Table 14."**
>
> Following the reviewer's request, we reproduce these baselines with their official code and conduct a comprehensive hyperparameter search, where TimeMixer still performs best. Here are the results. (See $\underline{\text{Table 23 of revised paper}}$ for full results)
>
> | Averaged from 4 prediction lengths (MSE \| MAE) | Scaleformer    | MTS-Mixer      | TSMixer        | TimeMixer          |
> | ----------------------------------------------- | -------------- | -------------- | -------------- | ------------------ |
> | Weather                                         | 0.248 \| 0.304 | 0.254 \| 0.278 | 0.240 \| 0.269 | **0.222 \| 0.262** |
> | Solar-Energy                                    | 0.270 \| 0.333 | 0.230 \| 0.277 | 0.229 \| 0.283 | **0.192 \| 0.244** |
> | Electricity                                     | 0.191 \| 0.298 | 0.179 \| 0.286 | 0.167 \| 0.262 | **0.156 \| 0.246** |
> | Traffic                                         | 0.443 \| 0.307 | 0.539 \| 0.354 | 0.415 \| 0.290 | **0.387 \| 0.262** |
> | ETTh1                                           | 0.468 \| 0.466 | 0.461 \| 0.464 | 0.418 \| 0.432 | **0.411 \| 0.423** |
> | ETTh2                                           | 0.412 \| 0.432 | 0.397 \| 0.422 | 0.350 \| 0.399 | **0.316 \| 0.384** |
> | ETTm1                                           | 0.406 \| 0.421 | 0.393 \| 0.405 | 0.350 \| 0.377 | **0.348 \| 0.375** |
> | ETTh2                                           | 0.288 \| 0.336 | 0.284 \| 0.335 | 0.270 \| 0.326 | **0.256 \| 0.315** |
>
> During the rebuttal period, following the instruction from the reviewer, we have done our best to provide double experiment results for both settings for scientific rigor. We hope our effort in rebuttal (**more than 200 new results in total, 9 new pages, 7 days experiment with 32 A100 GPUs**) can resolve the reviewer's concerns.

---

> ### Author Response · Authors · 2023-11-23
> **The discussion period ending soon**
>
> Dear Reviewer,
>
> Thanks again for your valuable and constructive review, which helped us improve the paper in completing ablations and comparisons. All the updates have been included in the  $\underline{\text{revised paper highlighted in purple}}$.
>
> **We kindly remind you that the reviewer-author discussion phase will end in 3 hours. May we know if our response addresses your main concerns? After that, we will not have a chance to respond to your comments.**
>
> Sincere thanks for your dedication! We eagerly await your reply.

---

> ### Author Response · Authors · 2023-11-23
> **[The discussion period will end in one hour] & [Summary of rebuttal]**
>
> Dear Reviewer,
>
> Many thanks for your valuable review. With the deadline for the author-reviewer discussion phase drawing near (**less than 1 hour**), we wish to ensure that our response sufficiently addressed your concerns. After this hour, we may not have a chance to answer your question.
>
> For clearness, we summary our revision that we made towards your concerns here.
>
> **(1) New baselines**: we have added 3 baselines that you mentioned in all 18 benchmarks under both unified and search hyperparameter settings.
>
> **(2) New ablations:** we have completed 10 types of ablations in all 18 benchmarks under the unified hyperparameter setting and provided the requested ablations in two specific settings faithfully following the reviewer's request.
>
> **(3) New visualizations:** we have provided visualization for FMM in two types of configurations and clarified that ensemble strategies that the reviewer mentioned are equivalent in the deep model perspective.
>
> In total, we have provided extensive experiments (**more than 200 new results, 9 new pages**) to address every concern that you proposed. Hope our effort can address your concerns to your stastification and looking forward to your reply.

---

> ### Comment · Reviewer_ixvB · 2023-11-23
>
> First of all, I want to express my gratitude for your effort.
>
> I read all of the authors' responses and the discussions with other reviewers.
>
> However, there are three points where I am reluctant to raise the point.
>
> 1) Actually, I found that the performance scores of TSMixer in your paper are different from those of TSMixer officially reported in [1] whereas PachTST is the same in both cases. This inconsistency is quite critical in believing the scores in the paper.
>
> 2) Furthermore, when comparing the scores of TimeMixer to those of TSMixer reported in [1], I don't think there is a large margin between them. At this point, I don't know that the module the authors proposed is truly required to boost forecasting performance when considering that TimeMixer is similar to TSMixer without some methods.
>
> 3) Finally, in this case where performance improvement is mediocre, I believe that more ablation studies are required to emphasize the effectiveness of asymmetric design.
>
> Because of these points, I decide to uphold the score as before. (Still, I respect the different opinions of other reviewers.)
>
> [1] Chen et al., TSMixer: An All-MLP Architecture for Time Series Forecasting, 2023

---

> ### Author Response · Authors · 2023-11-23
> **Thanks for your response**
>
> Many thanks for your response.
>
> > Inconsistency with TSMixer offical results comes from the training epochs.
>
> As for the comparison with TSMixer, we would like to highlight that the difference is from the training epoch, where **we only employ 10 training epochs due to the time limitation in rebuttal, while TSMixer employs 100 epochs.** Note that training 10 epochs is a convention in this area.
>
> Beisdes, as we always emphasized, we have made a great effort to ensure fair comparison. The unified hyperparameter setting results are also important, given many works have followed this setting, such as Scaleformer, MTSMixer, TimesNet.
>
> > About ablations.
>
> Since many preivous papers only provide the input 96 settings and we have provided ablations in both input settings strictly following your suggestions, we do hope the reviewer can take the double workload into consideration.
>
> Overall, we have made our best to ensure fair compairson. Hope the reviewer can reconsider the score.

---

### Official Review · Reviewer_ansv · 2023-10-31

**Soundness:** 3 good
**Presentation:** 4 excellent
**Contribution:** 3 good
**Rating:** 8
**Confidence:** 3

**Summary:**

The authors tackled the challenge of temporal variations in time series forecasting, recognizing that real-world time series frequently present complex temporal fluctuations, which intensify the intricacy of forecasting. In response, they introduced TimeMixer: a model grounded in a multiscale mixing architecture, integrating both Past-Decomposable-Mixing and FutureMultipredictor-Mixing components. Central to their approach is the notion of capturing variations across different time scales. This design leverages disentangled variations while harnessing complementary forecasting strengths. Empirical evaluations revealed that TimeMixer consistently achieved state-of-the-art results for both long-term and short-term forecasting, with its MLP-centric architecture offering remarkable runtime efficiency.

**Strengths:**

1. **Solid Motivation**:
    - The motivation behind the paper is robust and well-justified.
    - The significance of **TIME SERIES FORECASTING** is inherently evident and requires no further validation.
    - Echoing the authors' sentiments, the multiscale analysis paradigm stands out as a classic yet crucial methodology to model the intricate temporal variations inherent in time series data.

2. **Coherent Conceptual Framework**:
    - The core idea of decomposing the signal into various scales and subsequently aligning trends across these scales is both intuitive and compelling.
    - This seemingly straightforward approach not only resonates with the fundamentals of time series analysis but has also demonstrated its efficacy in the authors' experiments.

3. **Rigorous Experimental Evaluation**:
    - The experiments conducted in the paper are methodologically sound, lending further credence to the proposed approach.
    - The ablation study is particularly convincing, illuminating the individual contributions of different components.
    - It's worth noting the diverse range of benchmarks utilized and the comparison with competitive models, which further underscores the robustness and generalizability of the proposed method.

4. **Clear Presentation**:
    - The paper is commendably articulated.
    - The authors present their ideas with a blend of intuitive explanations supplemented with coherent textual descriptions, making the content both accessible and insightful for readers.

**Weaknesses:**

1. **Analysis and Explanation of the Results**:
    - While the empirical experiments, inclusive of the ablation study, provide evidence of the effectiveness of TimeMixer, the underlying reasons for its superior performance remain somewhat opaque. Specifically, when juxtaposed against competing models, it's not lucidly expounded how TimeMixer excels in capturing temporal variations. A deeper dive into this comparative analysis would have been enlightening. Introducing spectral analysis or similar methodologies might offer theoretical insights that bridge this understanding gap.

2. **Alternative Decomposition Methods**:
    - The paper seems to sidestep the exploration of alternative decomposition techniques that are prevalent in the domain. For instance, methods such as the Discrete Fourier Transformer (DFT) have been widely recognized in time series analysis. How does the TimeMixer's Past Decomposable Mixing distinguish itself fundamentally from these frequency-based techniques? A comparative discourse delving into the nuances between the proposed method and established frequency-based approaches would enrich the paper's content and address potential queries from the readership.
    - Average pooling is only one of the pooling methods one could try. Is that possible that any alternatives could be better or how the authors could prove that average pooling is the optimal one.

**Questions:**

In relation to the second weakness mentioned, how does TimeMixer's Past Decomposable Mixing fundamentally differentiate itself from frequency-based methods like the Discrete Fourier Transformer (DFT)? I wonder if, while it might compute variations across domains, it might not efficiently capture temporal dynamics without specific design considerations. Furthermore, as I've highlighted, could there be alternatives to Average pooling that might perform better? How can the authors demonstrate that average pooling is indeed the optimal choice?

---

> ### Author Response · Authors · 2023-11-15
> **Response to Reviewer ansv [Part 1]**
>
> We would like to sincerely thank Reviewer ansv for providing the valuable feedback.
>
> > **Q1:** "A deeper dive into this comparative analysis would have been enlightening. Introducing spectral analysis or similar methodologies might offer theoretical insights that bridge this understanding gap."
>
> Thanks for the reviewer's valuable suggestion. We have provided the spectral analysis in the $\underline{\text{Appendix I of revised paper}}$. It is observed that TimeMixer outperforms other baselines significantly in capturing different frequencies.
>
> > **Q2-part 1:** "How does TimeMixer's Past Decomposable Mixing fundamentally differentiate itself from frequency-based methods like the Discrete Fourier Transformer (DFT)?"
>
> **(1) Ablations on decomposition methods**
>
> In this paper, we adopt the moving-average-based season-trend decomposition, which is widely used in previous work, such as Autoformer, FEDformer and DLinear. Following the reviewer's suggestion, we also try the DFT-based decomposition as a substitute. Here we present two types of experiments.
>
> - DFT-based high- and low-frequency decomposition: We treat the high-frequency part like the seasonal part in TimeMixer and the low-frequency part like the trend part.
> - DFT-based season-trend decomposition: We replace the moving average with DFT-based seasonal part extraction, which the most significant frequencies are extracted as season. Then the rest is trend.
>
> The results are shown as follows, which has also been included in $\underline{\text{Appendix G.2 of revised paper}}$.
>
> | Ablation on decomposition methods               | ETTm1 (Predict-336) MSE \| MAE | M4 SMAPE \| MASE \| OWA | PEMS04 MAE \| MAPE \| RMSE |
> | ----------------------------------------------- | ---------------------------------- | -------------------------- | ------------------------------- |
> | DFT-based high- and low-frequency decomposition | 0.392 \| 0.404                     | 12.054 \| 1.632 \| 0.862   | 19.83 \| 12.74 \| 31.48         |
> | DFT-based season-trend decomposition            | 0.383 \| 0.399                     | 11.673 \| 1.536 \| 0.824   | 18.91 \| 12.27 \| 29.47         |
> | moving-average-based Season-trend decomposition | 0.390 \| 0.404                     | 11.723 \| 1.559 \| 0.840   | 19.21 \| 12.53 \| 30.92         |
>
> We can have the following observations:
>
> - Replacing the season-trend decomposition with high-low-frequency decomposition does not bring performance gain. Since we only explore the proper mixing approach for the former decomposition in the paper, the bottom-up and top-down mixing strategies may be not suitable for high- and low-frequency parts. New visualizations like $\underline{\text{Figure 3 and 4 of original submission}}$ are expected to provide insights to the model design. Thus, we would like to leave the exploration as the future work.
>
> - Enhancing season-trend decomposition with DFT perfroms better than moving average. However, since moving average is quite simple and easy to implement with PyTorch, we eventually chose the moving-average-based season-trend decomposition in TimeMixer, which can also achieve a favorable balance between performance and efficiency.
>
> **(2) Discussion about season-trend decomposition and DFT-based methods**
>
> Firstly, we would like to highlight that **we propose TimeMixer towards a simple but effective method**, which we have already emphasized in the Introduction section. **In the spirit of building a practical model, proving that seasonal-trend decomposition is optimal or fundamentally different from other decomposition methods is out of the scope of our paper.**
>
> But thanks for the reviewer's suggestion. We would like to leave this discussion of the optimality and completeness of our model as future work, which has been added to the $\underline{\text{Appendix F of revised paper}}$.

---

> > ### Comment · Reviewer_ansv · 2023-11-21
> >
> > Thanks to the authors for addressing my questions and concerns. Although I still believe that emphasizing optimalism for this work is crucial in the end, I appreciate your efforts in conducting new experiments to explore the methods I suggested. I anticipate your future work.

---

> ### Author Response · Authors · 2023-11-15
> **Response to Reviewer ansv [Part 2]**
>
> > **Q2-part 2:** "Could there be alternatives to average pooling that might perform better? How can the authors demonstrate that average pooling is indeed the optimal choice?"
>
> As per the reviewer's request, we replaced the average pooling with 1D convolutions with stride as 2. Here are the results. we can find that the complicated 1D-convolution-based outperforms average pooling slightly. But considering both performance and efficiency, we eventually use average pooling in TimeMixer.
>
> | Ablation on downsampling methods | ETTm1 (Predict-336) MSE \|MAE | M4 SMAPE \|MASE \|OWA | PEMS04  MAE \|MAPE \|RMSE |
> | -------------------------------- | -------------------------------- | ------------------------ | ---------------------------- |
> | Moving average                   | 0.390 \|  0.404                  | 11.723 \| 1.559 \| 0.840 | 19.21 \| 12.53 \| 30.92      |
> | 1D convolutions with stride as 2 | 0.387 \|  0.401                  | 11.682 \| 1.542 \| 0.831 | 19.04 \| 12.17 \| 29.88      |
>
> Again, we just want to build a practical model. Although average pooling may be not optimal, it obviously an effective and easy-to-implement choice. The above ablation has also been included in $\underline{\text{Appendix G.3 of revised paper}}$.

---

### Official Review · Reviewer_P2SD · 2023-11-02

**Soundness:** 2 fair
**Presentation:** 3 good
**Contribution:** 2 fair
**Rating:** 6
**Confidence:** 4

**Summary:**

This work proposed TimeMixer, which employs a multiscale mixing architecture to address the complex temporal variations in time series forecasting. By utilizing Past-Decomposable-Mixing and Future-Multipredictor-Mixing blocks, TimeMixer leveraged disentangled variations and complementary forecasting capabilities. Additionally, thanks to its fully MLP-based architecture, TimeMixer demonstrated efficient runtime processing.

**Strengths:**

S1. The paper has a clear and easily understandable structure.

S2. The experiments are extensive, involving long time series forecasting without the use of highly noisy exchange-rate and illness datasets. Additionally, a new solar-energy dataset is introduced, and the provided code and configurations enhance the credibility of the experimental results.

**Weaknesses:**

W1. In general, upsampling results in more data points, while downsampling results in fewer data points (as illustrated in Figure 1, leftmost). Based on this, I believe the descriptions 'Up-mixing' and 'Down-Mixing' in Figure 2 by the authors may not be appropriate and should perhaps be reversed.

W2. The paper lacks significant innovation. (1) Decoupling of multiscale [1], seasonal-trend disentanglement [2] are common modules that have already been proposed.  (2) The so-called FMM module appears to be essentially a linear layer mapping the concatenated multiscale features back to the prediction length in the time dimension. In my view, the author's main contribution seems to be the integration of these two [1,2]  modules.

[1] MICN: Multi-scale Local and Global Context Modeling for Long-term Series Forecasting

[2] Autoformer: Decomposition Transformers with Auto-Correlation for Long-Term Series Forecasting

W3. The improvement in experimental performance is marginal. Table 10 (the best results among all baselines) reveals that the actual enhancement by TimeMixer is not substantial, especially in three larger datasets, 'traffic,' 'weather,' and 'Electricity,' where it only holds a slight advantage at the third decimal place.

**Questions:**

Please see my listed weaknesses.

**Details Of Ethics Concerns:**

N/A.

---

> ### Author Response · Authors · 2023-11-15
> **Response to Reviewer P2SD [Part 1]**
>
> We would like to sincerely thank Reviewer P2SD for providing a detailed review and insightful suggestions.
>
> > **Q1:** "The descriptions 'Up-Mixing' and 'Down-Mixing' in Figure 2 may not be appropriate and should perhaps be reversed."
>
> Sorry for these two inaccurate formalizations. Following your suggestion, we have rephrased 'Up-Mixing' and 'Down-Mixing' to 'Bottom-Up-Mixing' and 'Top-Down-Mixing'  in both figures and text of the $\underline{\text{reviserd paper}}$.
>
>
>
> > **Q2-part1:** "The paper lacks significant innovation. In my view, the author's main contribution seems to be the integration of these two modules from MICN and Autoformer."
>
> **(1) TimeMixer is clearly distinct from MICN and Autoformer.**
>
> Firstly, TimeMixer is a **MLP-based** model, while MICN is **convolution-based** and Autoformer is **Transformer-based**.
>
> Secondly, although multiscale analysis and decomposition have been used in previous models, we would like to highlight that **their usage in TimeMixer is distinct from MICN and Autoformer in both motivation and technical design,** which is summarized in the following tables:
>
> | Motivation comparison | Why use decomposition?                                       | Why use multiscale analysis?                                 |
> | --------------------- | ------------------------------------------------------------ | ------------------------------------------------------------ |
> | Autoformer            | Disentangle intricate variations                             | **/**                                                        |
> | MICN                  | Disentangle intricate variations                             | Aggregate **local information** in different size regions with multiple convolutional kernels |
> | **TimeMixer (Ours)**  | We find that **mixing directions should be different for seasonal and trend parts**. | **1. Disentangle intricate variations** **2. Utilize complementary forecasting capabilities in multiscale series** |
>
> | Design comparison    | How to use decomposition?                                    | How to use multiscale analysis?                              |
> | -------------------- | ------------------------------------------------------------ | ------------------------------------------------------------ |
> | Autoformer           | Decompose **one-scale input series and one-scale intermediate results** | **/**                                                        |
> | MICN                 | Decompose **one-scale input series**                         | **Only for the seasonal part**                               |
> | **TimeMixer (Ours)** | Decompose **multiscale deep features**                       | **1. For both seasonal and trend features** **2. Propose to use different mixing directions for multiscale seasonal and trend features** |
>
> We do appreciate that MICN and Autoformer introduce multiscale analysis and decomposition into deep time series forecasting. But obviously, these three models are quite different in both model architecture and usage of the aforementioned two modules.
>
> **(2) The design in adopting different mixing directions for decomposed parts should not be overlooked.**
>
> In the paper, **we propose to utilize top-down and bottom-up mixing directions for seasonal and trend parts respectively**, which is a key contribution in our model. This design is well supported by:
>
> - Motivation analysis in $\underline{\text{3rd paragraph in Introduction}}$ and $\underline{\text{Section 3.2}}$.
> - Ablations in $\underline{\text{Table 5}}$, which verifies that our proposed mixing approach performs best.
> - Visualizations in $\underline{\text{Figure 3}}$, which demonstrates that seasonal and trend parts should utilize different mixing directions.
>
> Thus, TimeMixer is far beyond simple integration of multiscale analysis and decomposition. To highlight this point, we have rephrased the listed contributions in the $\underline{\text{Introduction of revised paper}}$.

---

> ### Author Response · Authors · 2023-11-15
> **Response to Reviewer P2SD [Part 2]**
>
> > **Q2-part 2:** "The so-called FMM module appears to be essentially a linear layer mapping the concatenated multiscale features back to the prediction length in the time dimension."
>
> **(1) There might be some misunderstandings of FMM (Future-Multipredictor-Mixing).**
>
> As presented in $\underline{\text{Equation 6 of main text}}$, we adopt **different predictors** to regress the future from past features in different scales separately.
>
> Thus, we think the review might be misconceived in the following two points:
>
> - We do not concatenate multiscale features, but keep the past information in their original length and directly regress the future separately. *(This is also different from the multiscale design in MICN.)*
> - FMM is not a single linear layer mapping, but an ensemble of multiple predictors.
>
> For clearness, we have rephrased $\underline{\text{Section 3.3 in the revised paper}}$ to clarify the misleading parts.
>
> **(2) FMM can utilize complementary forecasting capabilities in multiscale series, not just align the time dimension.**
>
> As we presented in $\underline{\text{Figure 4 of original submission}}$, different scales present complementary forecasting capabilities, which is why we adopt multiple different predictors in FMM. To our best knowledge, this point is firstly explored by TimeMixer, which is also an important design in our paper.
>
> > **Q3:** "The improvement in experimental performance is marginal, especially in three larger datasets, 'Traffic', 'Weather,' and 'Electricity'."
>
> We believe that to measure the value of a deep model, we should consider $\underline{\text{performance, hyperparameter tuning cost and efficiency}}$ simultaneously.
>
> **(1) In this unified hyperparameter setting, TimeMixer surpasses other baselines significantly.**
>
> As shown in $\underline{\text{Table 2,3,4 of original submission}}$, TimeMixer is clearly better than the previous state-of-the-art PatchTST, SCINet, and TimesNet in both performance and efficiency. This advantage is meaningful since sometimes we do not have enough time or sources for hyperparameter searching in real-world applications.
>
> **(2) In the hyperparameter-search setting, TimeMixer is much more efficient than the second-best model PatchTST.**
>
> As pointed out by the reviewer, after a comprehensive hyperparameter-search, the relative improvement of TimeMixer against PatchTST is smaller than the unified hyperparameter setting. But we have to highlight that, TimeMixer still holds a significant advantage in efficiency, making it a valuable model to this community and applications.
>
> To make this clearer, we list some results from $\underline{\text{Table 8 and Table 14 of original submission}}$ as follows. As presented in $\underline{\text{Table 6 of original submission}}$, Solar-Energy is the largest dataset and Weather is the second. We can find that TimeMixer surpasses PatchTST with 25% and 7.8% relative promotion in these two large benchmarks even after the hyperparameter searching. Besides, TimeMixer is over 2 times faster than PatchTST. In addition, we also present the results of DLinear, which is comparable to TimeMixer in efficiency. But TimeMixer beats it with more than 10% performance gain.
>
> These analyses have also been included in the $\underline{\text{Appendix E of revised paper}}$.
>
> | Model     | Solar-Energy (averaged MSE) Largest dataset | Weather (averaged MSE) Second Large dataset | Electricity (averaged MSE) | Traffic (averaged MSE) | GPU memory (predict-384) | Running Time (predict-384) |
> | --------- | ----------------------------------------------- | ---------------------------------------------- | -------------------------- | ---------------------- | ------------------------ | -------------------------- |
> | PatchTST  | 0.256                                           | 0.241                                          | 0.159                      | 0.391                  | 2097 MiB                 | 0.019 s/iter               |
> | DLinear   | 0.329                                           | 0.246                                          | 0.166                      | 0.434                  | 1021 MiB                 | 0.003 s/iter               |
> | TimeMixer | **0.192**                                       | **0.222**                                      | **0.156**                  | **0.387**              | 1043 MiB                 | 0.007 s/iter               |

---

> ### Author Response · Authors · 2023-11-18
> **Request of Reviewer's attention and feedback**
>
> Dear Reviewer,
>
> Thanks for your valuable and constructive review, which has inspired us to improve our paper further substantially. This is a kind reminder that it has been 3 days since we posted our rebuttal. Please let us know if our response has addressed your concerns.
>
> Following your suggestions, we have provided the following revisions to our paper:
>
> - **Elaborate the difference between TimeMixer and MICN, Autoformer** in using decomposition and multiscale analysis.
> - **Revise Figure 2** and text of the main text to resolve inaccurate formalizations.
> - **Revise the listed contributions in the $\underline{\text{Introduction section}}$** to highlight our contributions in proposing new mixing directions for decomposed components and utilizing multiple predictors for multiscale series.
> - **Revise the $\underline{\text{Future-Mixing section}}$** to clarify our design in Future-Multipredictor-Mixing block.
> - Highlight that although under hyperparameter searching, the relative promotion of TimeMixer is smaller than the unified hyperparameter setting, **TimeMixer is 2x faster than the second-best model PatchTST.**
>
> In this paper, we propose the TimeMixer as a simple but effective model and provide extensive experiments, visualization, and ablations to support our insight. All the revisions are included in the $\underline{\text{revised paper}}$.
>
> Sincere thanks for your dedication! We are looking forward to your reply.

---

> ### Author Response · Authors · 2023-11-22
> **The discussion period ending soon**
>
> Dear Reviewer,
>
> Thanks again for your valuable and constructive review, which helps us revise our contribution and advantages to a clearer stage.
>
> **We kindly remind you that the reviewer-author discussion phase will end in 24 hours. After that, we may not have a chance to respond to your comments**.
>
> Besides, during the rebuttal period, we also completed the ablations in all 18 benchmarks (**more than 100 new experiments**), which may be helpful to you in further justifying our contribution in PDM and FMM. **All of these results have been included in the $\underline{\text{revised paper}}$.**
>
> Sincere thanks for your dedication! We are looking forward to your feedback.

---

> > ### Comment · Reviewer_P2SD · 2023-11-22
> > **Response to Author's Rebuttal**
> >
> > Thanks for the detailed rebuttal, which further clarify the value of the work and supplement richer experiments. Although I feel there is a little lack of innovation,  the experiments are solid enough and maybe this paper will inspire anyone else. The open-sourcing of the code will also benefit the time-series community. Therefore, I increase my score from 5 to 6.

---

> > > ### Author Response · Authors · 2023-11-22
> > > **Many thanks for your response and raising score**
> > >
> > > Sincerely thanks for your response. We propose TimeMixer towards a simple but effective model based on intuitive observations and supported by extensive evaluations. Following your valuable suggestions, we revised the paper in highlighting our key contributions and discussing the model performance more comprehensively.
> > >
> > > Thanks again for your dedication in reviewing our paper. It helps us a lot.

---

### Author Response · Authors · 2023-11-15
**Summary of Revisions**

We sincerely thank all the reviewers for their insightful reviews and valuable comments, which are instructive for us to improve our paper further.

This paper presents the TimeMixer as a simple but effective model for time series forecasting, which analyzes temporal variations in a novel decomposable-multiscale-mixing view, taking advantage of disentangled variations and complementary forecasting capabilities from multiscale series simultaneously. **Experimentally, TimeMixer surpasses 15 advanced baselines in 18 well-established benchmarks with competitive run-time efficiency, covering both long-term and short-term forecasting tasks.**

The reviewers generally held positive opinions of our paper, in that the proposed method is "**intuitive and compelling**" with "**solid motivation**" and "**good observations**"; this paper is "**well-written**", "**commendably articulated**" and "**in a clear and easily understandable structure**"; we have provided "**extensive and rigorous experiments**"; "**the ablation study is particularly convincing**"; "**the model performance should not be underestimated**".

The reviewers also raised insightful and constructive concerns. We made every effort to address all the concerns by providing detailed clarification and requested results. Here is the summary of the major revisions:

- **Elaborate differences among TimeMixer, MICN and Autoformer (Reviewer P2SD):** We clarify that TimeMixer is distinct from MICN and Autoformer in the usage of multiscale analysis and decomposition from both motivation and technical design aspects. Also, we rephrased the Introduction section to highlight our contributions in exploring the decomposable mixing direction and ensembling multiple predictors.
- **Discuss the performance gain under the hyperparameter-search setting (Reviewer P2SD):** We highlight that we should consider performance, hyperparameter tuning cost, and efficiency simultaneously. (1) TimeMixer outperforms other baselines significantly under unified hyperparameter. (2) Although the performance gain is smaller under hyperparameter searching, TimeMixer still outperforms the second-best model PatchTST significantly in Solar-Energy, the largest dataset, and is 2x faster than PatchTST.
- **Provide spectral analysis to results (Reviewer ansv):** Following the reviewer's suggestion, we add the spectrogram of prediction cases, where TimeMixer can capture different frequency parts precisely.
- **Add comparison with Scaleformer, MTSMixer, and TSMixer (Reviewer ixvB):**  Following the reviewer's suggestion, we have compared the above three baselines in all 18 benchmarks. Comparing to these new baselines, TimeMixer still performs best. The showcases of these newly added baselines are also provided.
- **Complete the ablations in all benchmarks (Reviewer ansv, ixvB):** As per the reviewer's request, we have completed 10 types of ablations in all 18 benchmarks, where our official design in TimeMixer performs best. The ablations on alternative decomposition and downsampling methods are also provided.

**After 5 full days of experiments (with 32 A100 GPUs), we have newly added more than 150 new experiment results to address the mentioned issues. All the revisions have been included in the $\underline{\text{revised paper highlighted in orange}}$.**

The valuable suggestions from reviewers are very helpful for us to revise the paper to a better shape. We'd be very happy to answer any further questions.

Looking forward to the reviewer's feedback.

---

### Meta-Review · Area_Chair_EtsQ · 2023-12-07

**Metareview:**

The proposed paper is targeting time series forecasting using two somewhat minor novelties; using MLP mixer instead of transformer blocks and simultaneous processing of bottom-up and top-down streams. Though these are minor, authors provide comprehensive empirical study clearly showing the value of the approach. The paper was reviewed by 3 reviewers and initially received diverging reviews. After the discussion, two of the reviewers suggested acceptance whereas one remained in reject decision. I carefully read the paper, reviews, authors' response and the discussion. I believe the paper has a merit to be published.

The main issue raised against acceptance is incomplete demonstration of the value. Although I initially agreed with the reviewer, authors additional experiments demonstrate the value to a large extent. More important authors proved their willingness to extend and complete the empirical study by extending them during the rebuttal. Hence, I believe this issue is largely resolved and will be completely resolved by the camera-ready deadline. I strongly urge authors to complete the empirical study and integrate it to the main test.

**Justification For Why Not Higher Score:**

The paper has very limited novelty. Although the approach is technically sound and the experiments are rigorous, it will still have limited impact due to limited novelty.

**Justification For Why Not Lower Score:**

The paper is introducing a technically sound method with extensive analysis. The results are also promising. Hence, it should be shared with the community.

---

### Decision · Program_Chairs · 2024-01-16

Accept (poster)